# Improved representation of the global dust cycle using observational constraints on dust properties and abundance

Jasper F. Kok[1], Adeyemi A. Adebiyi[1], Samuel Albani[2,3], Yves Balkanski[3], Ramiro Checa-Garcia[3], Mian Chin[4], Peter R. Colarco[4], Douglas S. Hamilton[5], Yue Huang[1], Akinori Ito[6], Martina Klose[7,§], Danny M. Leung[1], Longlei Li[5], Natalie M. Mahowald[5], Ron L. Miller[8], Vincenzo Obiso[7,8], Carlos Pérez García-Pando[7,9], Adriana Rocha-Lima[10,11], Jessica S. Wan[5], and Chloe A. Whicker[1]

[1]Department of Atmospheric and Oceanic Sciences, University of California, Los Angeles, CA 90095, USA
[2]Department of Environmental and Earth Sciences, University of Milano-Bicocca, Milano, Italy
[3]Laboratoire des Sciences du Climat et de l'Environnement, CEA-CNRS-UVSQ-UPSaclay, Gif-sur-Yvette, France
[4]Atmospheric Chemistry and Dynamics Laboratory, NASA Goddard Space Flight Center, Greenbelt, MD 20771, USA
[5]Department of Earth and Atmospheric Sciences, Cornell University, Ithaca, NY 14850, USA
[6]Yokohama Institute for Earth Sciences, JAMSTEC, Yokohama, Kanagawa 236-0001, Japan
[7]Barcelona Supercomputing Center (BSC), 08034 Barcelona, Spain
[8]NASA Goddard Institute for Space Studies, New York NY10025 USA
[9]ICREA, Catalan Institution for Research and Advanced Studies, 08010 Barcelona, Spain
[10]Physics Department, UMBC, Baltimore, Maryland, USA
[11]Joint Center Joint Center for Earth Systems Technology, UMBC, Baltimore, Maryland, USA
§Present address: Institute of Meteorology and Climate Research (IMK-TRO), Department Troposphere Research, Karlsruhe Institute of Technology (KIT), Karlsruhe, Germany

*Correspondence to:* Jasper F. Kok (jfkok@ucla.edu)

**Abstract.** Even though desert dust is the most abundant aerosol by mass in Earth's atmosphere, atmospheric models struggle to accurately represent its spatial and temporal distribution. These model errors are partially caused by fundamental difficulties in simulating dust emission in coarse-resolution models and in accurately representing dust microphysical properties. Here we mitigate these problems by developing a new methodology that yields an improved representation of the global dust cycle. We present an analytical framework that uses inverse modeling to integrate an ensemble of global model simulations with observational constraints on the dust size distribution, extinction efficiency, and regional dust aerosol optical depth. We then compare the inverse model results against independent measurements of dust surface concentration and deposition flux and find that errors are reduced by approximately a factor of two relative to current model simulations of the Northern Hemisphere dust cycle. The inverse model results show smaller improvements in the less dusty Southern Hemisphere, most likely because both the model simulations and the observational constraints used in the inverse model are less accurate. On a global basis, we find that the emission flux of dust with geometric diameter up to 20 μm (PM$_{20}$) is approximately 5,000 Tg/year, which is greater than most models account for. This larger PM$_{20}$ dust flux is needed to match observational constraints showing a large atmospheric loading of coarse dust. We obtain gridded data sets of dust emission, vertically integrated loading, dust aerosol optical depth, (surface) concentration, and wet and dry deposition fluxes that are resolved by season and particle size. As our results indicate that this data set is more accurate than current model simulations and the MERRA-2 dust reanalysis product, it can be used to improve quantifications of dust impacts on the Earth system.

## 1. Introduction

Desert dust produces a wide range of important impacts on the Earth system, including through interactions with
radiation, clouds, the cryosphere, biogeochemistry, atmospheric chemistry, and public health (Shao et al., 2011).
Despite the important role of dust in the Earth system, simulations of the global dust cycle suffer from several key
deficiencies. For instance, models show large differences relative to observations for critical aspects of the global
dust cycle, including dust size distribution, surface concentration, dust aerosol optical depth (DAOD), and
deposition flux (e.g., Huneeus et al., 2011; Albani et al., 2014; Ansmann et al., 2017; Adebiyi and Kok, 2020; Wu et
al., 2020). Moreover, models struggle to reproduce observed interannual and decadal changes in the global dust
cycle over the observational record (Mahowald et al., 2014; Ridley et al., 2014; Smith et al., 2017; Evan, 2018; Pu
and Ginoux, 2018), and it remains unclear whether atmospheric dust loading will increase or decrease in response to
future climate and land use changes (Stanelle et al., 2014; Kok et al., 2018).

One key reason that models struggle to accurately represent the global dust cycle and its sensitivity to climate and
land use changes is that dust emission is a complex process for which the relevant physical parameters vary over
short distances of about 1 m to several km (Okin, 2008; Bullard et al., 2011; Prigent et al., 2012; Shalom et al.,
2020). As such, large-scale models with typical spatial resolutions on the order of 100 km are fundamentally ill-
equipped to accurately simulate dust emission. Confounding the problem is the non-linear scaling of dust emissions
with near-surface wind speed above a threshold value (Gillette, 1979; Shao et al., 1993; Kok et al., 2012; Martin and
Kok, 2017). As such, dust emissions are especially sensitive to errors in simulating high-wind events (Cowie et al.,
2015; Roberts et al., 2017) and to variations in the soil properties that set the threshold wind speed. Despite some
recent progress, accounting for the effect of sub-grid-scale wind variability on dust emissions remains a substantial
challenge that causes the simulated global dust cycle to be sensitive to model resolution (Lunt and Valdes, 2002;
Cakmur et al., 2004; Comola et al., 2019), especially at low resolution (Ridley et al., 2013). Another substantial
challenge for models is the small-scale variability of vegetation (Raupach et al., 1993; Okin, 2008), surface
roughness (Menut et al., 2013), soil texture (Laurent et al., 2008; Martin and Kok, 2019), mineralogy (Perlwitz et al.,
2015a), and soil moisture (McKenna Neuman and Nickling, 1989; Fécan et al., 1999). These and other soil
properties control both the dust emission threshold, and the intensity of dust emissions once wind exceeds the
threshold (Gillette, 1979; Shao, 2001; Kok et al., 2014b). Models lack accurate high-resolution data sets of pertinent
soil properties, which also limits the use of dust emission parameterizations that incorporate the effect of these soil
properties (e.g., Darmenova et al., 2009). As a result of these fundamental challenges in accurately representing dust
emission, most models use both a source function map (Ginoux et al., 2001) and a global dust emission tuning
constant to produce a global dust cycle that is in reasonable agreement with measurements (Cakmur et al., 2006;
Huneeus et al., 2011; Albani et al., 2014; Wu et al., 2020).

A second key problem limiting the accuracy of model simulations of the global dust cycle is that models struggle to
adequately describe dust properties such as dust size, shape, mineralogy, and optical properties. All these dust
properties have recently been shown to be inaccurately represented in many models (Kok, 2011b; Perlwitz et al.,
2015b; Pérez Garcia-Pando et al., 2016; Ansmann et al., 2017; Di Biagio et al., 2017; Di Biagio et al., 2019; Adebiyi
and Kok, 2020; Huang et al., 2020). These model errors in dust properties occur because parameterizations are not
always kept consistent with up-to-date experimental and observational constraints. In addition, models need to use
fixed values for such physical variables and thus can only represent the uncertainties inherent in such constraints
through computationally expensive perturbed parameter ensembles (Bellouin et al., 2007; Lee et al., 2016).

The nature of these challenges in accurately representing the global dust cycle is such that they are difficult to
overcome from advances in modeling alone (e.g., Stevens, 2015; Kok et al., 2017; Adebiyi et al., 2020). We
therefore develop a new methodology to obtain an improved representation of the present-day global dust cycle. Our
approach builds on previous work that used a combination of observational and modeling results to constrain the
dust size distribution, extinction efficiency, and dust aerosol optical depth (Ridley et al., 2016; Kok et al., 2017;
Adebiyi and Kok, 2020; Adebiyi et al., 2020). We present an analytical framework that uses inverse modeling to
integrate these observational constraints on dust properties and abundance with an ensemble of global model
simulations. Our procedure determines the optimal emissions from different major source regions and particle size
ranges that result in the best match against these observational constraints on the dust size distribution, extinction
efficiency, and regional dust aerosol optical depth. Our methodology propagates uncertainties in these observational
constraints and due to the spread in simulations in the model ensemble. As such, our approach mitigates the
consequences of the fundamental difficulty that models have in representing the magnitude and spatiotemporal
variability of dust emission, and in representing the properties of dust and the uncertainties in those properties.

Moreover, whereas the assimilation of observations in reanalysis products creates inconsistencies between the different components of the dust cycle (i.e., emission, loading, and deposition are not internally consistent), our framework integrates observational constraints in a self-consistent manner.

We detail our approach in Sect. 2, after which we summarize independent measurements used to evaluate our representation of the global dust cycle in Sect. 3, and present results and discussion in Sections 4 and 5. We find that our procedure results in a substantially improved representation of the Northern Hemisphere global dust cycle and modest improvements for the Southern Hemisphere. We provide a data set representing the global dust cycle in the present climate (2004-2008) that is resolved by particle size and season. Because comparisons against independent measurements indicate that this data set is more accurate than obtained by an aerosol reanalysis product and by a

large number of climate and chemical transport model simulations, this data set can be used to obtain more accurate quantifications of the wide range of dust impacts on the Earth system.

## 2. Methods

We seek to obtain an improved representation of the global dust cycle by integrating observationally-informed constraints on dust properties and abundance with an ensemble of simulations of the spatial distribution of dust

emitted from different source regions. We achieved this with an analytical framework that uses optimal estimation to determine how many units of dust loading from different size ranges and main source regions are required to maximize agreement against observational constraints on the dust size distribution and dust aerosol optical depth near source regions (see Fig. 1). We then compare the results against independent measurements of dust surface concentration and deposition flux (Sect. 3.1). Although our methodology can be considered inverse modeling in that

it inverts observational constraints to force a model, the methodology used here differs substantially from standard inverse modeling studies used in atmospheric and oceanic sciences (e.g., Bennett, 2002; Dubovik et al., 2008; Escribano et al., 2016; Brasseur and Jacob, 2017; Chen et al., 2019) in that it uses a bootstrap procedure to integrate several different observational constraints on dust microphysical properties and abundance and to propagate and quantify uncertainties. We summarize the methodology in the next few paragraphs and then describe each step in

detail in the sections that follow.

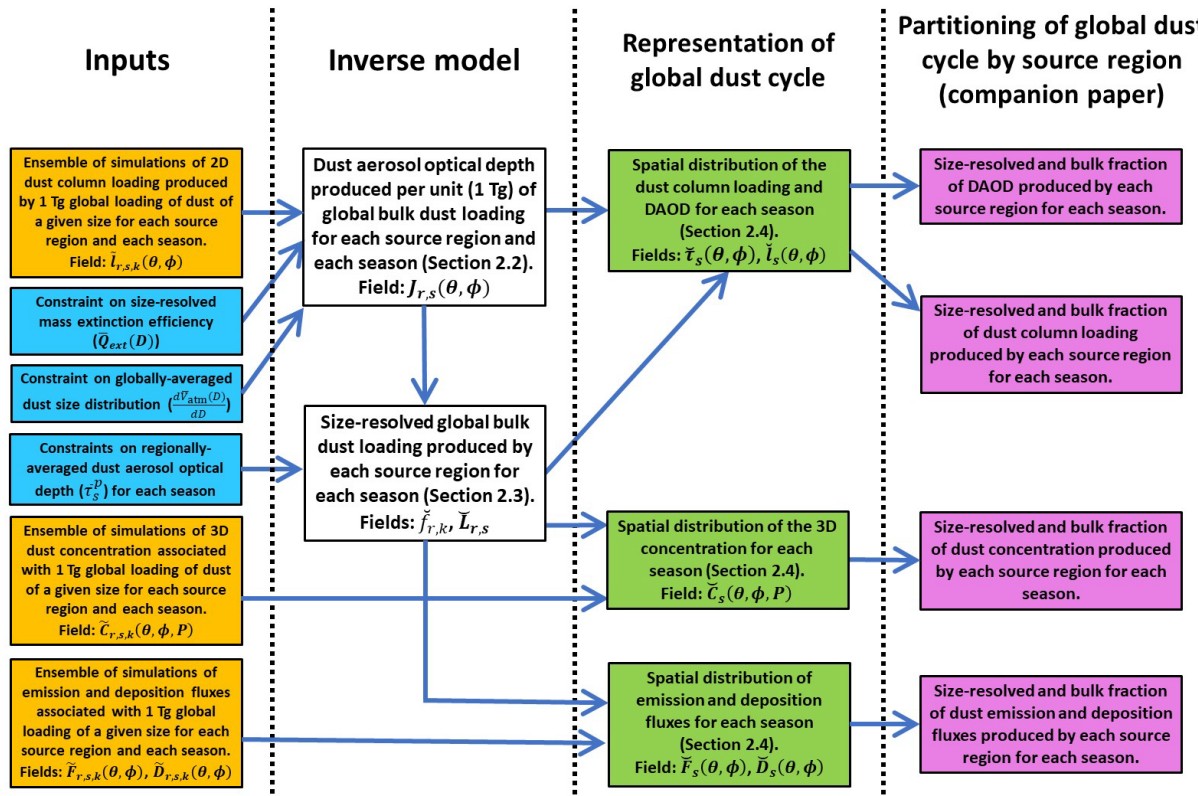

**Figure 1. Schematic of the methodology used to obtain an improved representation of the global dust cycle.** Yellow boxes denote inputs from an ensemble of global model simulations, blue boxes denote inputs from observational constraints on dust properties and abundance, and white boxes denote the inverse model. We report the resulting representation of the global dust cycle in the present paper (green boxes) and the partitioning of the global dust cycle by source region (magenta boxes) in our companion paper (Kok et al., 2021). The subscripts $r$, $s$, and $k$ respectively refer to the originating source region, the season, and a model's particle size bin. Other variables are defined in the main text and the glossary.

We first divided the world into nine major source regions (Fig. 2a), and obtained an ensemble of global model simulations of how a unit of dust mass loading (1 Tg) of different particle sizes from each of these source regions is distributed across the atmosphere (Sect. 2.1). We then used constraints on the globally averaged dust size distribution (Adebiyi and Kok, 2020) and the size-resolved dust extinction efficiency (Kok et al., 2017) to determine the column-integrated dust aerosol optical depth produced by a single unit of bulk dust loading (1 Tg) from each source region (Sect. 2.2). Then, we used an inverse model to determine the optimum number of units of loading that must be generated by each source region to best match joint observational-modeling constraints on the DAOD for fifteen regions (Fig. 2b) near major dust sources (Sect. 2.3). The calculations in Sections 2.2 and 2.3 are performed iteratively, because the fractional contribution to global dust loading from each source region affects the agreement against the constraint on the globally averaged dust size distribution. Since we have more regional DAOD constraints than we have source regions, the problem is over-constrained, allowing for lower uncertainties on our results.

We summed the optimal dust loadings of the nine source regions to obtain the main properties of the global dust cycle, resolved by particle size, season, and location. Specifically, we obtained the dust emission flux, loading, concentration, deposition flux, and DAOD (Sect. 2.4), which we added to the Dust Constraints from joint Experimental-Modeling-Observational Analysis (DustCOMM) dataset (Adebiyi et al., 2020). Throughout these calculations, we used a bootstrap procedure to propagate uncertainties in the observational constraints on dust properties and abundance, and uncertainties due to the spread in our ensemble of model simulations of the spatial distributions of a unit of dust loading, concentration, and deposition (Sect. 2.5).

Our methodology uses a large number of variables, which are all listed in the glossary for clarity. To further help distinguish between different variables, we denote input variables obtained directly from global model simulations with the accent "~" (yellow boxes in Fig. 1). These fields are seasonally averaged and either two-dimensional (2D; $\theta, \phi$) or three-dimensional (3D; $\theta, \phi, P$), where $\theta$, $\phi$, and $P$ respectively denote longitude, latitude, and the vertical pressure level (see Table 1). Moreover, all model fields are "normalized", meaning that they represent values produced per unit (1 Tg) of global loading of dust in a given particle size bin $k$ from a given source region $r$ and for a given season $s$ (seasons are taken as December-January-February (DJF), March-April-May (MAM), June-July-August (JJA), and September-October-November (SON)), We further use the accent "-" to denote an observational constraint on dust properties or dust abundance (blue boxes in Fig. 1). These include constraints on the globally averaged dust size distribution ($\frac{d\bar{V}_{\text{atm}}(D)}{dD}$), the size-resolved extinction efficiency ($\bar{Q}_{\text{ext}}(D)$), and the regional DAOD ($\bar{\tau}_s^p$). All these fields have a quantified uncertainty, which we propagated through our analysis using the bootstrap procedure discussed in Sect. 2.5. Finally, the accent "~" denotes a product that results from our analysis, such as the 3D dust concentration, resolved by particle size and season (white and green boxes in Fig. 1). Such variables are thus generated by combining normalized model simulations with observational constraints on the dust size distribution, size-resolved extinction efficiency, and the DAOD near source regions.

## 2.1.  Dividing the world into nine main source regions

The first step in our methodology is to divide the world into its major source regions. Most dust is emitted from the so-called "dust belt" of Northern Africa, the Middle East, Central Asia, and the Chinese and Mongolian deserts (Prospero et al., 2002). In addition, dust is emitted in smaller quantities from Australia, South Africa, and North and South America. Correspondingly, we divided the world into nine source regions that together account for the overwhelming majority (>99%) of desert dust emissions simulated in models (Fig. 2a). Our analysis includes both natural and anthropogenic (land use) emissions of dust in those source regions, because our analysis is based on observations that by nature integrate both (but see further discussion in Sect. 5.1). However, our analysis explicitly does not include high-latitude dust sources, which produce dust through different mechanisms and with different properties than desert dust, yet likely dominate the dust loading for some high latitude regions (Prospero et al., 2012; Bullard et al., 2016; Tobo et al., 2019; Bachelder et al., 2020). The nine source regions partially follow the definition in Mahowald (2007), with the main difference that we divided the North African source region, which accounts for

~half of global dust emissions (Wu et al., 2020), into western North Africa, eastern North Africa, and the Sahel. Similar dust source regions were also used in more recent studies (Ginoux et al., 2012; Di Biagio et al., 2017).

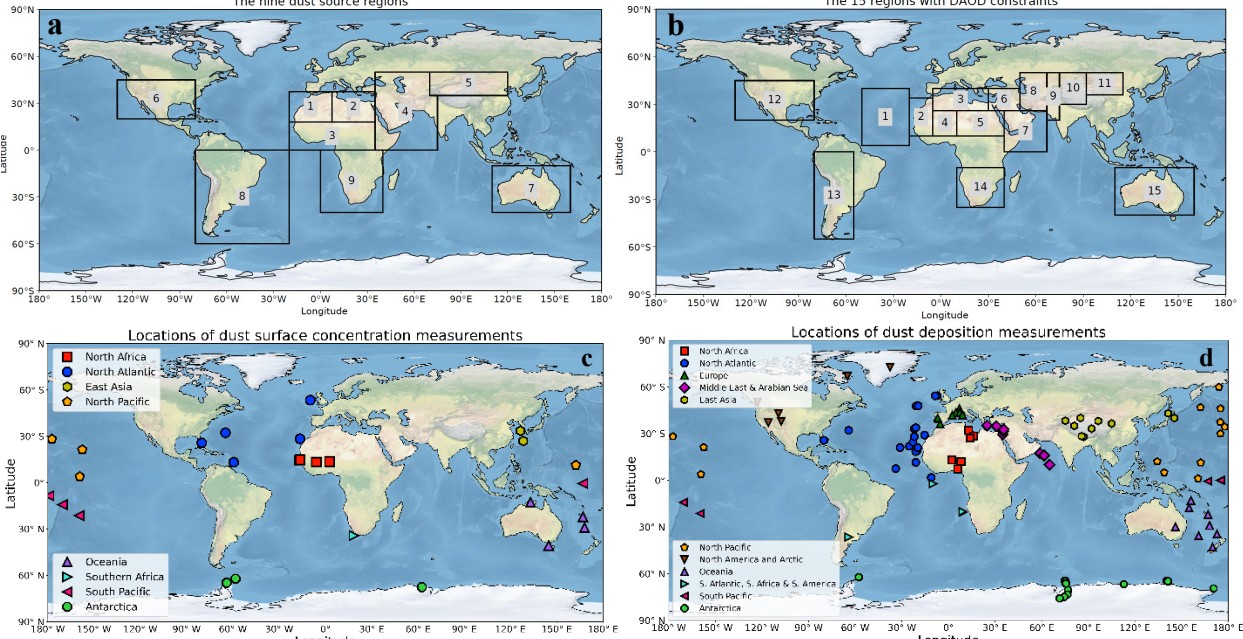

**Figure 2. Coordinates of (a) the nine main source regions, (b) the 15 observed regions with constraints on the regional dust aerosol optical depth (DAOD), (c) dust surface concentration measurements, and (d) deposition flux measurements used in this study.** The coordinates of the nine source regions are: (1) western North Africa (20° W – 7.5° E; 18° N – 37.5° N), (2) eastern North Africa (7.5E – 35E; 18N – 37.5N), (3) the Sahel (20W – 35E; 0N – 18N), (4) the Middle East & Central Asia (which includes the Horn of Africa; 35° E – 75° E for 0° N – 35° N, and 35° E – 70° E for 35° N – 50° N), (5) East Asia (70° E – 120° E; 35° N – 50° N), (6) North America (130° W – 80° W; 20° N – 45° N), (7) Australia (110° E – 160° E; 10° S – 40° S), (8) South America (80° W – 20° W; 0° S – 60° S), and (9) Southern Africa (0° E – 40° E; 0° S – 40° S). The coordinates and seasonal DAOD of the 15 observed regions are listed in Table 2. Symbols in (c) and (d) denote groupings of observations by different regions. Made with Natural Earth.

We use an ensemble of global chemical transport and climate models (see Table 1) to obtain simulations of emission, transport, and deposition of dust from each of the nine source regions. Specifically, we use simulations from the Community Earth System Model (CESM; Hurrell et al., 2013; Scanza et al., 2018), IMPACT (Ito et al., 2020), ModelE2.1 (Miller et al., 2006; Kelley et al., 2020), GEOS/GOCART (Rienecker et al., 2008; Colarco et al., 2010), MONARCH (Perez et al., 2011; Badia et al., 2017; Klose et al., in review), and INCA/IPSL-CM6 (Boucher et al., 2020). These six models were forced with three different reanalysis meteorology data sets (Table 1), which helped sample the uncertainty due to the exact reanalysis meteorology used, which past work indicates is substantial (Largeron et al., 2015; Smith et al., 2017; Evan, 2018). Most of the six models were ran for the years 2004-2008, or a subset thereof, to coincide with the analysis period of regional DAOD in Ridley et al. (2016), which provided most of observational DAOD constraints used in this study (see Table 1). Sensitivity tests indicated that using different years from each simulation resulted in differences of less than 10% in the inverse model results. Each model either ran a separate simulation for each source region, or used "tagged" dust tracers from each source region. The exact set-up of each model is described in the supplement.

Our inverse model uses several results derived from model simulations (Fig. 1). First, for each model we obtained the "normalized" seasonally averaged column loading $\tilde{l}_{r,s,k}(\theta,\phi)$, which is the spatial distribution of a unit (1 Tg) of loading originating from source region $r$, for season $s$ and particle size bin $k$. As such, the units of this field are m$^{-2}$ (Tg m$^{-2}$ loading per Tg of loading from source $r$), and we show annual averages of the normalized bulk dust loading for each model and source region in Fig. S1. Additionally, we obtained the "normalized" 3D concentration (m$^{-3}$) ($\tilde{C}_{r,k,s}(\theta,\phi,P)$, units of m$^{-3}$) and the 2D dust emission ($\tilde{F}_{r,k,s}(\theta,\phi)$, units of m$^{-2}$ yr$^{-1}$) and (dry and wet) deposition fluxes ($\tilde{D}_{r,k,s}(\theta,\phi)$, units of m$^{-2}$ yr$^{-1}$) that are associated with a unit of global dust loading for each source region, season, and particle size bin. All model fields were regridded using a modified Akima cubic Hermite

interpolation (Akima, 1970) to a common resolution of 1.9° latitude by 2.5° longitude and with 48 vertical levels (see Adebiyi et al. (2020) for further details). As explained further below, since our inverse model only uses normalized model fields per particle size, our results are independent of model tuning of global dust emissions or the simulated relative contributions of the major source regions defined here (Fig. 1). Our results are also not affected by model errors in representing dust mass extinction efficiency or the emitted dust size distribution.

We restricted our analysis to dust with diameter $D \leq D_{max} = 20$ μm because there are insufficient measurements to constrain the abundance of coarser dust particles in the atmosphere (Adebiyi and Kok, 2020). Note, however, that the few measurements that have been made of dust with $D > 20$ μm suggest that it is abundant over and near source regions such as North Africa and accounts for a non-negligible fraction of shortwave and longwave extinction (Ryder et al., 2019). As such, more measurements of "super coarse" ($D > 10$ μm) and "giant" ($D > 62.5$ μm) dust are needed, which would allow the analysis presented here to be extended to larger particle sizes in the future. Since some of the models in our ensemble do not account for dust with $D$ up to 20 μm, we use the procedure in Adebiyi et al. (2020; see their section 2.3.1) to extend these models to 20 μm. Specifically, we use the normalized $12 - 20$ μm particle size bin simulated by the GEOS/GOCART model to estimate what the CESM and GISS ModelE2.1 models would have simulated for an additional particle size bin extending to 20 μm (see additional details in the Supplement). We chose this bin specifically from the GEOS/GOCART model because it shows the best agreement against the observational constraint on regional DAOD (Fig. 3).

**Table 1.** Overview of global model set-ups used in this study.

| Model number | Model name | Spatial resolution** (longitude x latitude x level) | Dust particle size bin diameter ranges (μm) | Simulation period*** | Meteorological dataset used |
|---|---|---|---|---|---|
| 1 | CESM/CAM4 | 2.5° x 1.9° x 56 levels | 0.1-1; 1.0-2.5; 2.5-5; 5-10; 10-20* | 2004 - 2008 | ERA-Interim |
| 2 | IMPACT | 2.5° x 2.0° x 59 levels | 0.1-1.26; 1.26-2.5; 2.5-5; 5-20 | 2004 - 2005 | MERRA2 |
| 3 | GISS ModelE2.1 | 2.5° x 2.0° x 40 levels | 0.2-0.36; 0.36-0.6; 0.6-1.2; 1.2-2; 2-4; 4-8; 8-16; 16-20* | 2004 - 2008 | NCEP |
| 4 | GEOS/ GOCART | 1.25° x 1.0° x 72 levels | 0.2-2; 2-3.6; 3.6-6; 6-12; 12-20 | 2004 - 2008 | MERRA2 |
| 5 | MONARCH | 1.4° x 1.0° x 48 levels | 0.2-0.36; 0.36-0.6; 0.6-1.2; 1.2-2; 2-3.6; 3.6-6; 6-12; 12-20 | 2004 - 2008 | ERA-Interim |
| 6 | INCA | 2.5° x 1.27° x 79 levels | 0.2-2; 2-3.6; 3.6-6; 6-12; 12-20 | 2010 - 2014 | ERA-Interim |

*Denotes an additional bin added to the original model output in order to extend the particle diameter range to Dmax = 20 μm. This additional bin was derived from the GEOS/GOCART 12-20 μm particle size bin (see main text).
**All model fields were regridded to a common resolution of 2.5° longitude by 1.9° latitude.
***A multi-year mean for each season was used.

## 2.2. Constraining the spatially resolved DAOD corresponding to a unit (1 Tg) of bulk dust loading

We next implemented an inverse model to determine the optimal bulk dust loading that must be generated by each source region to produce the best match against constraints on regional DAOD. This inverse model thus requires the spatial pattern of DAOD produced per unit bulk dust loading from each source region, which is the Jacobian matrix of DAOD with respect to dust loading. We obtained this DAOD produced per unit (1 Tg) of bulk dust loading by combining the simulated distributions of a unit of size-resolved dust loading ($\tilde{l}_{r,s,k}(\theta, \phi)$) with constraints on the globally averaged dust size distribution and extinction efficiency (Kok et al., 2017; Adebiyi and Kok, 2020). The calculations of the Jacobian matrix (this section) and the optimal bulk loading per source region (next section) are performed iteratively, because each source region's fractional contribution to global dust loading from affects the agreement against the constraint on the globally averaged dust size distribution.

The DAOD produced per unit of bulk dust loading originating from source region $r$ in season $s$ is (Kok et al., 2017)

$$J_{r,s}(\theta, \phi) = \frac{\partial \breve{\tau}_{r,s}(\theta, \phi)}{\partial \breve{L}_{r,s}} = \sum_{k=1}^{N_{bins}} \bar{\epsilon}_k \breve{f}_{r,s,k} \tilde{l}_{r,s,k}(\theta, \phi), \tag{1}$$

where $\breve{L}_{r,s}$ is the globally integrated bulk dust loading generated by source region $r$ in season $s$, $\breve{\tau}_{r,s}(\theta,\phi)$ is the spatial distribution of DAOD due to dust from source region $r$ in season $s$, $J_{r,s}$ is the Jacobian matrix (Tg$^{-1}$) of $\breve{\tau}_{r,s}$ with respect to $\breve{L}_{r,s}$, $N_{\text{bins}}$ is the number of particle size bins in a global model simulation (or derived from the simulated modes), $\bar{\epsilon}_k$ is the size-dependent mass extinction efficiency (m$^2$/g) of particle size bin $k$ and is defined further below, $\tilde{l}_{r,s,k}(\theta,\phi)$ (m$^{-2}$) is the simulated seasonally averaged spatial distribution of a unit of dust loading from source region $r$ and particle bin $k$, and $\breve{f}_{r,s,k}$ (unitless) is the fractional contribution of dust loading in size bin $k$ to the seasonally averaged global dust loading generated by source region $r$ (i.e., $\sum_k \breve{f}_{r,s,k} = 1$). As such, Eq. (1) obtains the DAOD produced per unit dust loading from a given source region and season by adding up the normalized spatial distributions of the loading from each particle size bin, in proportion to each bin's contribution to the globally integrated loading produced by the source region, and then multiplying the size-resolved loading by the mass extinction efficiency (MEE) to obtain the DAOD.

To obtain the Jacobian matrix in Eq. (1) we need to obtain $\breve{f}_{r,s,k}$, each particle bin's fractional contribution to the globally integrated dust loading generated by source region $r$ in season $s$. Because models as a group underestimate the mass of particles with larger diameters ($D > {\sim}5$ μm; Kok et al., 2017), we adjust the model size distribution to match a constraint on the globally averaged dust size distribution derived from a combination of observations and models (Adebiyi and Kok, 2020). This procedure retains regional differences in the atmospheric dust size distribution that models simulated for the different source regions, while forcing the globally averaged dust size distribution that results from the summed contributions from all source regions to match the constraint on the globally averaged dust size distribution. That is,

$$\breve{f}_{r,s,k} = \frac{\alpha_k \tilde{f}_{r,s,k}}{\sum_{k=1}^{N_{\text{bins}}} \alpha_k \tilde{f}_{r,s,k}}, \tag{2}$$

where $\tilde{f}_{r,s,k}$ is the modeled mass fraction per particle size bin for a given source region $r$ and season $s$, and $\alpha_k$ is the global correction factor for particle size bin $k$, which is different for each model. We obtained $\alpha_k$ by setting the fraction of atmospheric dust in particle size bin $k$, summed over all source regions and seasons, equal to the constraint on the fractional contribution of particle size bin $k$ to the global dust loading from Adebiyi and Kok (2020). That is,

$$\alpha_k = \frac{\int_{D_{k-}}^{D_{k+}} \frac{d\bar{V}_{\text{atm}}(D)}{dD} dD}{\sum_r^{N_{\text{sreg}}} \sum_{s=1}^{N_{\text{s}}} \tilde{f}_{r,s,k} \breve{L}_{r,s} / \sum_{r=1}^{N_{\text{sreg}}} \sum_s^{N_{\text{s}}} \breve{L}_{r,s}}, \tag{3}$$

where $N_{\text{sreg}} = 9$ is the number of source regions (Fig. 2a) and $\frac{d\bar{V}_{\text{atm}}(D)}{dD}$ is a realization of the size-normalized (that is, $\int_0^{D_{\max}} \frac{d\bar{V}_{\text{atm}}}{dD} dD = 1$, where $D_{\max} = 20$ μm) globally averaged volume size distribution from Adebiyi and Kok (2020), which was obtained by combining dozens of *in situ* measurements of dust size distributions with an ensemble of climate model simulations. Further, $D_{k-}$ and $D_{k+}$ are respectively the lower and upper diameter limits of particle size bin $k$, and $\breve{L}_{r,s}$ is the globally integrated and seasonally averaged bulk dust loading per source region (as obtained from the analysis below). As such, the denominator in Eq. (3) denotes the simulated globally averaged mass fraction, whereas the numerator denotes the globally averaged mass fraction in particle size bin $k$ as constrained from *in situ* measurements and model simulations by Adebiyi and Kok (2020).

The final ingredient needed to use Eq. (1) to obtain the DAOD produced by a unit (1 Tg) bulk dust loading from a given source region and season is the MEE ($\bar{\epsilon}_k$). We do not use each model's assumed MEE because these tend to be substantially biased compared to measurements (Adebiyi et al., 2020). This bias is largely due to a neglect or underestimation of the asphericity of dust (Huang et al., 2020), which increases the surface-to-volume ratio and thereby enhances the MEE by ${\sim}40$ % (Kok et al., 2017). We thus follow Kok et al. (2017) in obtaining the MEE from constraints on the dust size distribution and the extinction efficiency of randomly-oriented (Ginoux, 2003; Bagheri and Bonadonna, 2016) aspherical dust. That is,

$$\bar{\epsilon}_k = \frac{3}{2\bar{\rho}_d} \frac{\int_{D_{k-}}^{D_{k+}} \frac{\bar{Q}_{\text{ext}}(D)}{D} \frac{d\bar{V}_{\text{atm}}(D)}{dD} dD}{\int_{D_{k-}}^{D_{k+}} \frac{d\bar{V}_{\text{atm}}(D)}{dD} dD}. \tag{4}$$

where $\bar{Q}_{\text{ext}}(D)$ is a realization of the globally averaged size-resolved extinction efficiency from the analysis of Kok et al. (2017), which is defined as the extinction cross-section divided by the projected area of a sphere with diameter $D$ ($\pi D^2 / 4$). The term $\frac{d\bar{V}_{\text{atm}}(D)}{dD}$ inside the integrals approximates the sub-bin distribution in particle size bin $k$ as the globally averaged dust volume size distribution. Further, $\bar{\rho}_d = (2.5 \pm 0.2) \times 10^3$ kg m$^{-3}$ is the globally averaged density of dust aerosols (Fratini et al., 2007; Reid et al., 2008; Kaaden et al., 2009; Sow et al., 2009). This observationally

constrained density of dust is lower than the 2600 to 2650 kg m$^{-3}$ used in many models (Tegen et al., 2002; Ginoux et al., 2004), most likely because dust aerosols are aggregates with void space that lowers their density below that of individual mineral particles.

### 2.3.    Constraining the bulk dust loading generated by each source region

The above procedure combined model simulations of the 2D spatial variability of size-resolved dust loading with
constraints on dust size distribution and MEE. This procedure yielded the spatial distribution of DAOD that is produced by a unit (1 Tg) of dust loading from a given source region and season. Next, we use an inverse modeling approach to determine the number of Tg of loading needed from each source region to produce optimal agreement against constraints on the seasonal DAOD over areas proximal to major dust source regions.

We use joint observational-modeling constraints on regional DAOD at 550 nm from Ridley et al. (2016). This study
used three different satellite AOD retrievals – from the Multi-angle Imaging Radiometer (MISR) and the Moderate Resolution Imaging Spectroradiometer (MODIS) onboard the Terra and Aqua satellites - and bias-corrected those satellite data using more accurate ground-based aerosol optical depth measurements from AERONET. Ridley et al. (2016) then used an ensemble of global model simulations to obtain the fraction of AOD that is due to dust in 15 regions for which AOD is dominated by dust. Ridley et al. (2016) thus leveraged the strengths of these different
tools by combining the accuracy of ground-based measurements with the global coverage of satellite retrievals and the ability of models to distinguish between different aerosol species. Furthermore, by averaging the resulting DAOD over large areas and long time periods (2004-2008 for each season), this study minimized representation errors that can affect model comparisons to data (Schutgens et al., 2017). An additional strength of the Ridley et al. (2016) analysis is that it transparently propagates a range of uncertainties that are both observationally and modeling
based, and which we in turn propagate into our own analysis (see Sect. 2.5). We also consider the Ridley et al. dataset more accurate than aerosol reanalysis products that assimilate similar AOD observations. This is because the Ridley et al. product includes a transparent quantification of errors that we propagated into the representation of the global dust cycle here, and because the partitioning of assimilated AOD into different aerosol species in reanalysis products depends on the underlying aerosol models and is thus susceptible to the large biases in the prognostic
aerosol schemes of these models (e.g., Adebiyi et al., 2020; Gliss et al., 2021). Nonetheless, the Ridley et al. (2016) data are subject to some important limitations discussed further in Sect. 5.1.

Although we consider the Ridley et al. (2016) constraints on DAOD to be more accurate than constraints from individual satellite products, AERONET data, or aerosol reanalysis products, this study's results for the Southern Hemisphere (SH) are susceptible to substantial biases. This is because dust makes up a substantially lower fraction
of total AOD in the SH than for the main Northern Hemisphere (NH) source regions (e.g., Fig. S2 in Kok et al., 2014a). Therefore, we did not use the Ridley et al. (2016) results for the SH and instead used the seasonally averaged DAOD estimated by Adebiyi et al. (2020) over the three SH regions. These DAOD constraints are based on an ensemble of four aerosol reanalysis products, namely the Modern-Era Retrospective analysis for Research and Applications, Version 2 (MERRA-2; Gelaro et al., 2017), the Navy Aerosol Analysis and Prediction System
(NAAPS; Lynch et al., 2016), the Japanese Reanalysis for Aerosol (JRAero; Yumimoto et al., 2017), and the Copernicus Atmosphere Monitoring Service (CAMS) interim Reanalysis (CAMSiRA; Flemming et al., 2017). The resulting regional DAOD product also includes an error estimation based partially on the spread in DAOD in the four reanalysis products. In addition, we added a region over North America, for which Ridley et al. (2016) did not obtain results, and for which we also use the reanalysis-based results of Adebiyi et al. (2020). In total, we thus have
constraints with error estimates on the seasonal and area-averaged DAOD over 15 regions (see Fig. 2b and Table 2).

**Table 2. Constraints on seasonal dust aerosol optical depth (DAOD) at 550 nm averaged over 15 regions.** Regional DAOD constraints for regions 1-11 are from Ridley et al. (2016) and were obtained using data from AERONET, MODIS, MISR, and a model ensemble. Regional DAOD constraints for regions 12-15 are from Adebiyi et al. (2020) and were obtained from an ensemble of aerosol reanalysis products. All constraints use data for the years 2004-2008.

| Region number $p$ | Region name | Region coordinates | DJF | MAM | JJA | SON |
|---|---|---|---|---|---|---|
| 1 | Mid-Atlantic | 20 - 50° W; 4 – 40° N | 0.064 ± 0.013 | 0.106 ± 0.008 | 0.143 ± 0.005 | 0.084 ± 0.006 |
| 2 | African West Coast | 20 – 5° W; 10 – 34° N | 0.180 ± 0.010 | 0.250 ± 0.019 | 0.365 ± 0.016 | 0.233 ± 0.022 |
| 3 | Northern Africa | 5° W – 30° E; 26 – 40° N | 0.118 ± 0.011 | 0.219 ± 0.010 | 0.207 ± 0.016 | 0.151 ± 0.016 |
| 4 | Mali/Niger | 5° W – 10° E; 10 – 26° N | 0.257 ± 0.019 | 0.441 ± 0.022 | 0.462 ± 0.044 | 0.277 ± 0.023 |
| 5 | Bodele/Sudan | 10 – 40° E; 10 – 26° N | 0.191 ± 0.006 | 0.339 ± 0.023 | 0.310 ± 0.018 | 0.212 ± 0.021 |
| 6 | Northern Middle East | 30 – 50° E; 26 – 40° N | 0.112 ± 0.011 | 0.223 ± 0.011 | 0.164 ± 0.015 | 0.113 ± 0.019 |
| 7 | Southern Middle East | 40 – 67.5° E; 0 – 26° N | 0.123 ± 0.018 | 0.204 ± 0.021 | 0.330 ± 0.044 | 0.150 ± 0.020 |
| 8 | Kyzyl Kum | 50 – 67.5° E; 26 – 50° N | 0.115 ± 0.017 | 0.176 ± 0.026 | 0.154 ± 0.034 | 0.101 ± 0.018 |
| 9 | Thar | 67.5 – 75° E; 20 – 50° N | 0.130 ± 0.029 | 0.238 ± 0.033 | 0.319 ± 0.029 | 0.135 ± 0.037 |
| 10 | Taklamakan | 75 – 92.5° E; 30 – 50° N | 0.119 ± 0.013 | 0.275 ± 0.027 | 0.171 ± 0.026 | 0.104 ± 0.011 |
| 11 | Gobi | 92.5 – 115° E; 36 – 50° N | 0.093 ± 0.022 | 0.192 ± 0.022 | 0.102 ± 0.035 | 0.047 ± 0.021 |
| 12 | North America | 80 – 130° W; 20 – 45° N | 0.010 ± 0.005 | 0.029 ± 0.011 | 0.028 ± 0.010 | 0.012 ± 0.006 |
| 13 | South America | 80 – 55° W; 0 – 55° S | 0.019 ± 0.011 | 0.013 ± 0.007 | 0.010 ± 0.006 | 0.016 ± 0.009 |
| 14 | Southern Africa | 10 – 40° E; 10 – 35° S | 0.016 ± 0.007 | 0.011 ± 0.005 | 0.013 ± 0.005 | 0.016 ± 0.007 |
| 15 | Australia | 110 – 160° E; 10 – 40° S | 0.025 ± 0.013 | 0.013 ± 0.006 | 0.010 ± 0.005 | 0.023 ± 0.011 |


We then used an inverse modeling approach to determine the optimal combination of dust loadings from the nine source regions (denoted with subscript $r$) that minimizes the disagreement against the DAOD constraint of these 15 observed regions (denoted with subscript $p$) for each season. We thus need to account for the contribution of each of the nine source regions (Fig. 2a) to the DAOD in each of these 15 observed regions. The seasonally averaged
DAOD over the observed region $p$ is

$$\bar{\tau}_s^p = \sum_{r=1}^{N_{\text{sreg}}} J_{r,s}^p \breve{L}_{r,s}, \tag{5}$$

where $\bar{\tau}_s^p$ is the DAOD averaged over observed region $p$ and season $s$, and $J_{r,s}^p$ (Tg$^{-1}$) is the Jacobian matrix of $\breve{\tau}_{r,s}^p$ with respect to $\breve{L}_{r,s}$, where $\breve{\tau}_{r,s}^p$ denotes the area-averaged and seasonally averaged DAOD over observed region $p$ that is produced by dust from source region $r$. The Jacobian matrix $J_{r,s}^p$ is the area-weighted DAOD over observed region $p$ that is produced per unit of bulk dust loading originating from source region $r$ in season $s$. We obtain $J_{r,s}^p$ by
integrating Eq.
(1) over $A_p$, the area of the observed region $p$ (Table 2),

$$J_{r,s}^p = \frac{\partial \breve{\tau}_{r,s}^p}{\partial \breve{L}_{r,s}} = \frac{\int_{A_p} \sum_k^{N_{\text{bins}}} \bar{\epsilon}_k \breve{f}_{r,k} \breve{l}_{r,s,k}(\theta,\phi) dA}{\int_{A_p} dA}. \tag{6}$$

The seasonally averaged globally integrated dust loading generated by each source region ($\breve{L}_{r,s}$) is thus determined from the number of units of dust loading from each source region $r$ that results in best agreement against the constraint on DAOD ($\bar{\tau}_s^p$) over the 15 observed regions. Eq. (5) thus represents a system of equations for each
simulation in our global model ensemble, which we can write in explicit matrix form for clarity,

$$\begin{bmatrix} \bar{\tau}_s^1 & \bar{\tau}_s^2 & \cdots & \bar{\tau}_s^{N_{\tau,\mathrm{reg}}} \end{bmatrix} = \begin{bmatrix} \breve{L}_{1,s} & \breve{L}_{2,s} & \cdots & \breve{L}_{N_{\mathrm{sreg}},s} \end{bmatrix} \begin{bmatrix} J_{1,s}^1 & J_{1,s}^2 & \cdots & J_{1,s}^{N_{\tau,\mathrm{reg}}} \\ J_{2,s}^1 & J_{2,s}^2 & \cdots & J_{2,s}^{N_{\tau,\mathrm{reg}}} \\ \vdots & \vdots & \ddots & \vdots \\ J_{N_{\mathrm{sreg}},s}^1 & J_{N_{\mathrm{sreg}},s}^2 & \cdots & J_{N_{\mathrm{sreg}},s}^{N_{\tau,\mathrm{reg}}} \end{bmatrix}. \tag{7}$$

We used Eq. (7) to obtain the seasonally averaged global dust loading generated by each source region. Specifically, for each season $s$ we used the simplex search optimization method (Lagarias et al., 1998) to determine the nine values of $\breve{L}_{r,s}$ that minimize the cost function of the summed squared deviation ($\chi_\tau^2$) between the 15 DAOD constraints and the corresponding regional DAOD calculated from Eq. (7). That is (e.g., Cakmur et al., 2006),

$$\chi_\tau^2 = \sum_{p=1}^{N_{\tau,\mathrm{reg}}} \sum_r^{N_{\mathrm{sreg}}} \left( \breve{L}_{r,s} J_{r,s}^p - \bar{\tau}_s^p \right)^2, \tag{8}$$

where $N_{\tau,\mathrm{reg}} = 15$ and $N_{\tau,\mathrm{reg}} = 9$. Because the variables in Eqs. (1)-(8) are interdependent, we iterated these equations until convergence was achieved.

### 2.4.    Obtaining constraints on DAOD, emission, loading, deposition, and concentration

After constraining the seasonal dust loading $\breve{L}_{r,s}$ generated by each source region, we now obtain the 2D DAOD and the size-resolved dust loading, emission and deposition fluxes, and 3D concentration. We do so by using the fact that other dust cycle components (DAOD, concentration, deposition) scale linearly with dust loading because our model simulations are driven by reanalysis products (Table 1), such that dust does not impact the meteorology. Each dust field can therefore be obtained by multiplying the simulated normalized dust field (e.g., seasonal dust concentration per unit dust loading) by the number of units of dust loading per source region and season ($\breve{L}_{r,s}$).

The 2D DAOD is then:

$$\breve{\tau}_s(\theta, \phi) = \sum_{r=1}^{N_{\mathrm{sreg}}} \breve{L}_{r,s} J_{r,s}(\theta, \phi). \tag{9}$$

The size-resolved and bulk dust loadings are respectively:

$$\breve{l}_{s,k}(\theta, \phi) = \sum_{r=1}^{N_{\mathrm{sreg}}} \breve{f}_{r,k} \breve{L}_{r,s} \tilde{l}_{r,s,k}(\theta, \phi), \text{ and} \tag{10}$$

$$\breve{l}_s(\theta, \phi) = \sum_{k=1}^{N_{\mathrm{bins}}} \sum_{r=1}^{N_{\mathrm{sreg}}} \breve{f}_{r,k} \breve{L}_{r,s} \tilde{l}_{r,s,k}(\theta, \phi). \tag{11}$$

Similarly, the 3D size-resolved and bulk concentrations produced by each source region are:

$$\breve{C}_{s,k}(\theta, \phi, P) = \sum_{r=1}^{N_{\mathrm{sreg}}} \breve{f}_{r,k} \breve{L}_{r,s} \tilde{C}_{r,s,k}(\theta, \phi, P), \text{ and} \tag{12}$$

$$\breve{C}_s(\theta, \phi, P) = \sum_{k=1}^{N_{\mathrm{bins}}} \sum_r^{N_{\mathrm{sreg}}} \breve{f}_{r,k} \breve{L}_{r,s} \tilde{C}_{r,s,k}(\theta, \phi, P), \tag{13}$$

where $P$ is the vertical pressure level. And the size-resolved and bulk emission fluxes are:

$$\breve{F}_{s,k}(\theta, \phi) = \sum_{r=1}^{N_{\mathrm{sreg}}} \breve{f}_{r,k} \breve{L}_{r,s} \tilde{F}_{r,s,k}(\theta, \phi), \text{ and} \tag{14}$$

$$\breve{F}_s(\theta, \phi) = \sum_{k=1}^{N_{\mathrm{bins}}} \sum_r^{N_{\mathrm{sreg}}} \breve{f}_{r,k} \breve{L}_{r,s} \tilde{F}_{r,s,k}(\theta, \phi). \tag{15}$$

Finally, the size-resolved and bulk deposition fluxes are:

$$\breve{D}_{s,k}(\theta, \phi) = \sum_{r=1}^{N_{\mathrm{sreg}}} \breve{f}_{r,k} \breve{L}_{r,s} \tilde{D}_{r,s,k}(\theta, \phi), \text{ and} \tag{16}$$

$$\breve{D}_s(\theta, \phi) = \sum_{k=1}^{N_{\mathrm{bins}}} \sum_r^{N_{\mathrm{sreg}}} \breve{f}_{r,k} \breve{L}_{r,s} \tilde{D}_{r,s,k}(\theta, \phi). \tag{17}$$

See the glossary for further descriptions of each variable. In our companion paper (Kok et al., 2021), we further
partition these fields into the originating source region.

## 2.5. Improved model and inverse model results with uncertainty

The results represented by Eqs. (9)-(17) require realizations of the various inputs (Fig. 1), which include both model fields and constraints on dust properties and abundance. Because each of these inputs is uncertain, and as such is represented by a probability distribution, we obtained two products that sample different aspects of this uncertainty of the inputs, namely "improved model" results and "inverse model" results.

First, we obtained "improved model" results by sampling over different realizations of observational constraints on dust properties and abundance but using the output of only a single model. That is, we solved Eqs. (1)-(17) a large number of times (100; limited by computational resources), and for each iteration we drew a random realization of each of the observational constraints but used simulation results from a single model. This procedure thus includes a random drawing of realizations of the globally averaged dust size distribution ($\frac{d\bar{V}_{atm}(D)}{dD}$), the extinction efficiency ($\bar{Q}_{ext}(D)$), the particle density ($\bar{\rho}_d$), and the observed regional DAOD ($\bar{\tau}_s^p$). As such, the improved model results represent output from a single model (see Table 1) for which DAOD is calculated from loading using the observational constraint on extinction efficiency (Eq. (4)), and for which the contributions from different source regions and particle bins are added in such a way to simultaneously match observational constraints on the dust size distribution (Eq. (2)) and DAOD (Eq. (8)).

Second, we obtained our main product, namely the "inverse model" product that represents the optimal representation of the global dust cycle. We obtained this product by similarly sampling over different realizations of the input fields, but now including a random drawing of one of the six global model simulations in each of the bootstrap iterations. This additional step propagates uncertainty in model predictions of the normalized size-resolved dust loading, concentration, and deposition fields into our results (Eqs. (9)-(17)). Because different models use different particle size bins (Table 1), we convert the size-resolved results from each bootstrap iteration to common particles size bins of 0.2-0.5, 0.5-1, 1-2.5, 2.5-5, 5-10, and 10-20 μm. We do so by assuming that sub-bin distributions follow the constraint on the globally averaged dust loading (Fig. 1). This assumption will introduce some further error in size-resolved results. For both the inverse model and improved model products, we retained only those bootstrap iterations that produced a root-mean-squared error of less than 0.05 relative to the DAOD constraints; this quality control retained approximately three quarter of the iterations.

In drawing the realizations of seasonally averaged observed DAOD ($\bar{\tau}_s^p$), we need to account for correlations of errors between different seasons and regions. Specifically, some of the errors in the calculation of the DAOD in Ridley et al. (2016) and Adebiyi et al. (2020) are systematic, such as errors in satellite retrieval algorithms and systematic model errors in simulations of (dust and non-dust) aerosols. These errors are thus at least partially correlated between seasons and regions, although we cannot establish the exact degree of correlation. We can thus roughly divide the errors into three different categories: errors that are completely random between seasons and regions, systematic errors that are correlated between different seasons for the same region, and systematic errors that are correlated across regions for a given season. The sum of the squared contributions of these three errors equals the square of the total error $\bar{\sigma}_s^p$ reported in Table 2. Since we cannot determine what the relative contribution of each of these three types of errors is, we assume that the contribution of each of these three errors is equal. Although the uncertainty in our results as quantified from the bootstrap procedure increases if a larger fraction of the DAOD error is assumed to be systematic, the median results presented in Sect. 4 are not sensitive to the partitioning of this error. The details of the mathematical treatment for calculating these errors are provided in the Supplement.

The bootstrap procedure used in the inverse model product propagates all the quantified random and systematic errors present in the inputs. Nonetheless, it cannot account for systematic biases in these inputs, such as the tendency of models to underestimate coarse dust lifetime (Ansmann et al., 2017; van der Does et al., 2018; Adebiyi et al., 2020). As such, the obtained uncertainty ranges should be interpreted as a lower bound on the actual uncertainty.

## 3. Comparison of inverse model results against independent measurements and model simulations

We evaluate the results of the inverse model described in the previous section using independent measurements of dust surface concentration and deposition fluxes (Sect. 3.1). We also compare the inverse model results against the ensemble of AeroCom Phase 1 global dust cycle simulations (Huneeus et al., 2011) and the MERRA-2 dust product (Sect. 3.2).

### 3.1. Independent dust measurements used to evaluate the inverse model

We use two sets of independent measurements to evaluate the ability of the inverse model to reproduce the global dust cycle. The first data set is a compilation of dust surface concentration measurements. Of the 27 total stations in this compilation, 22 are measurements of the bulk dust surface concentration taken in the North Atlantic from the Atmosphere–Ocean Chemistry Experiment (AEROCE; Arimoto et al., 1995) and taken in the Pacific Ocean from the sea–air exchange program (SEAREX; Prospero et al., 1989) for observation periods noted in Table 2 of Wu et

al. (2020). These data were obtained by drawing large volumes of air through a filter. To reduce the effects of anthropogenic aerosols, measurements were only taken when the wind was onshore and in excess of 1 m/s (Prospero et al., 1989). The mineral dust fraction of the collected particulates was determined either by burning the sample and assuming the ash residue to represent the mineral dust fraction, or from their Al content (assumed to be 8% for mineral dust, corresponding to the Al abundance in Earth's crust) (Prospero, 1999). Note that since these

measurements were taken during the period 1981-2000, the dust surface concentration "climatology" obtained from these measurements is for a different time period than that of the model simulations used in the inverse model (Table 1).

Since most of the AEROCE and SEAREX stations are located far downwind of source regions, we also added a data set of dust surface concentration from the Sahelian Dust Transect that was deployed in 2006 as part of the African

Monsoon Multidisciplinary Analysis (AMMA; Lebel et al., 2010; Marticorena et al., 2010). This data set contains measurements over 5-10 years of the surface concentration of aerosols with aerodynamic diameter $\leq 10$ μm ($PM_{10,aer}$) at four stations in the western Sahel (M'Bour, Bambey, Cinzana, and Banizoumbou; see http://www.lisa.u-pec.fr/SDT/). As with the AEROCE and SEAREX data sets, only measurements were used for which the wind direction was predominantly coming from dust-dominated regions. As such, these measurements have at least two

systematic errors: (i) the AMMA data reported the concentration of all particulate matter, so taking these measurements as being of dust concentration overestimates the true dust concentration, and (ii) measurements taken when wind was not coming from a dust-dominated region were omitted, which could also cause an overestimation of the dust concentration. To mitigate the effect of this second error, we only use seasonally averaged dust concentrations for which >70% of data was retained. This resulted in the omission of the Winter and Spring seasons

at the Bambey station.

Following Huneeus et al. (2011) and Wu et al. (2020), we additionally added surface concentration measurements of $PM_{10,aer}$ dust from a long-term (May 1995-December 1996) filter-based deployment in Jabiru, in northern Australia (Vanderzalm et al., 2003). However, unlike Huneeus et al. (2011) and Wu et al. (2020), we do not use data obtained in Rokumechi (Zimbabwe), which used a similar methodology, because most of the dust at this Southern Africa site

originated locally from within and near the national park where the station was located (p. 2649 in Nyanganyura et al., 2007).

To use the measurements of $PM_{10,aer}$ dust in Jabiru and the Sahel, we obtained the $PM_{10,aer}$ dust concentration for those models with size-resolved surface concentrations, namely the inverse model and each model in our ensemble. We did so by first obtaining the geometric diameter that corresponds to an aerodynamic diameter of 10 μm, which is

$D_{PM10aer} = c_{aer} \cdot 10$ μm $= 6.8$ μm. This uses the conversion factor $c_{aer} = 0.68$ from Huang et al. (2021), who accounted for the effects of particle shape (Huang et al., 2020) and density to link the aerodynamic and geometric diameters. For each model, we then summed the contributions from particle bins with diameters smaller than $D_{PM10aer}$ and used a correction factor $c_{PM10,aer}$ for particle size bins that straddle $D_{PM10aer}$. This correction factor uses the result from Adebiyi and Kok (2020) that the globally averaged dust size distribution ($\frac{d\bar{V}_{atm}(D)}{d\ln D}$) is approximately

constant in the range of 5 - 20 μm, such that the fractional contribution to the $PM_{10,aer}$ concentration of a bin that straddles $D_{PM10aer}$ can be approximated as

$$c_{PM10,aer} = \frac{\ln(D_{PM10aer}/D_{k-})}{\ln(D_{k+}/D_{k-})}, \tag{18}$$

where $D_{k-}$ and $D_{k+}$ are respectively the lower and upper limits of the particle size bin that straddles 10 μm aerodynamic diameter ($D = 6.8$ μm).

The second independent data set that we used to evaluate the inverse model results is a compilation (110 stations) of

the deposition flux of dust with geometric diameter $\leq 10$ μm ($PM_{10}$) from Albani et al. (2014). This study merged

data from previous data sets (Ginoux et al., 2001; Tegen et al., 2002; Lawrence and Neff, 2009; Mahowald et al., 2009) and adjusted these data to cover the $0.1 - 10$ μm geometric diameter range. We obtained the $PM_{10}$ deposition flux for the inverse model, the MERRA-2 data, and for each model in our ensemble following the approach above for the $PM_{10,aer}$ concentration data. Note that we cannot correct the concentration and deposition flux of the AeroCom Phase I models (next section) to the $PM_{10,aer}$ and $PM_{10}$ size ranges because of a lack of size-resolved simulation data. We thus used the bulk concentration and deposition fluxes as many of these models simulated the $PM_{10}$ size range (see Table 3 in Huneeus et al., 2011).

To assess the consistency of the inverse model results with both the independent data sets, we calculated the error-weighted mean-squared difference between the inverse model results and the observations. This statistic is known as the reduced chi squared statistic, and equals (Bevington and Robinson, 2003)

$$\chi_\nu^2 = \sum_i^{N_i} \frac{(M_i - O_i)^2}{\sigma_i^2 + \sigma_m^2}, \tag{19}$$

where the index $i$ sums over the $N_i$ measurements in the data set, $O_i$ is the $i$th measurement in the data set, $M_i$ is the inverse model result for the location and season of the $i$th measurement (if applicable), $\sigma_m$ is the calculated error in the inverse model result from the bootstrap procedure (see Section 2.5), and $\sigma_i$ is the error in the measurement. For a model that matches measurements within the experimental error, $\chi_\nu^2 \approx 1$ (Bevington and Robinson, 2003). Values of $\chi_\nu^2$ that are $<< 1$ indicate an overestimate of model or experimental error, whereas values of $\chi_\nu^2 >> 1$ indicate either an underestimate of errors or substantial biases in the model or experimental data.

We estimated the experimental errors on the surface concentration measurements by propagating the standard error in monthly averaged surface concentration measurements into seasonal and annual averages. Note that these errors do not include representation errors, which could be important (Schutgens et al., 2017). The errors on deposition data are more difficult to estimate, as these are not usually reported and because deposition fluxes can show large spatial and temporal variability (Avila et al., 1997), leading to larger representation errors. We estimated the relative error on deposition data measurements from the spread in measurements at similar locations. For the cluster of data in Southern Europe (eastern Spain, Southern France, Northern Italy; (e.g., Avila et al., 1997; Bonnet and Guieu, 2006), the standard deviation is about an order of magnitude, and for clusters of data north of Cape Verde (e.g., Jickells et al., 1996; Bory and Newton, 2000) and northwest of Tenerife (e.g., Honjo and Manganini, 1993; Kuss and Kremling, 1999) the standard deviation is about a quarter of an order of magnitude. We therefore take the relative error in deposition data as half an order of magnitude. This error is large compared to the inverse model error of approximately a quarter of an order of magnitude for deposition fluxes in the NH.

### 3.2.   Comparison of inverse model results against AeroCom models and MERRA-2

In order to compare the inverse model's representation of the global dust cycle against climate and chemical transport model simulations, we used the results of an ensemble of simulations for which the prognostic dust cycles were analyzed in detail, namely the AeroCom Phase I simulations of the dust cycle in the year 2000 (Huneeus et al., 2011). As such, the AeroCom simulations were obtained for a year closer to the time period in which most concentration and deposition measurements were taken (see above). We do not use newer AeroCom Phase II and Phase III simulations because only the dust component of Phase I models has been analyzed in detail. We furthermore also do not use recently analyzed dust cycle results from CMIP5 models (Pu and Ginoux, 2018; Wu et al., 2020) because less than half of CMIP5 models with prognostic dust cycles reported total deposition fluxes, which are needed for the analyses against measurements (see previous section). In addition, many CMIP5 models did not include a prognostic dust cycle and instead read in pre-calculated dust emissions (Lamarque et al., 2010). But note that CMIP5 model errors against measurements are similar to those for AeroCom models and those for our model ensemble (e.g., compare Figs. 8 and 9 in Wu et al. (2020) against Figs. S9, S10, S12, and S13).

We analyzed the AeroCom Phase 1 model results to obtain the seasonally and annually averaged DAOD at 550 nm, the dust surface concentration, and the annually averaged total (wet and dry) deposition fluxes for comparisons against measurements and the inverse model results. We also obtained the globally integrated annually averaged dust emission flux, dust loading, and DAOD. We obtained these variables for each of the 13 AeroCom simulations available from the online AeroCom database (see https://aerocom.met.no/; this repository does not contain the 14[th] model simulation analyzed in Huneeus et al. (2011), from the ECMWF model, which is thus omitted here).

We also analyzed the MERRA-2 dust product (Gelaro et al., 2017) in order to compare the inverse model's representation of the global dust cycle against a leading aerosol reanalysis product. We obtained the same variables from the MERRA-2 data as from the AeroCom data, except that we analyzed the MERRA-2 data for the years 2004-2008 to coincide with the regional DAOD constraints (Table 2).

We quantified the agreement of the various models against measurements using Taylor diagrams (Taylor, 2001) and by the correlation coefficients, bias, and root-mean-squared errors (RMSE). Because the surface concentration and deposition flux measurements span several orders of magnitude, their RMSEs are calculated in log-space. We furthermore quantified overall model agreement against measurements by calculating the normalized error $\Phi_m$ against the available data for each hemisphere:

$$\Phi_m^{\text{NH}} = \frac{1}{3}\left( \frac{S_{\tau,m}^{\text{NH}}}{\sum_n^{N_{\text{model}}} \frac{S_{\tau,n}^{\text{NH}}}{N_{\text{model}}}} + \frac{S_{\text{conc},m}^{\text{NH}}}{\sum_n^{N_{\text{model}}} \frac{S_{\text{conc},n}^{\text{NH}}}{N_{\text{model}}}} + \frac{S_{\text{dep},m}^{\text{NH}}}{\sum_n^{N_{\text{model}}} \frac{S_{\text{dep},n}^{\text{NH}}}{N_{\text{model}}}} \right) \tag{20}$$

$$\Phi_m^{\text{SH}} = \frac{1}{2}\left( \frac{S_{\text{conc},m}^{\text{SH}}}{\sum_n^{N_{\text{model}}} \frac{S_{\text{conc},n}^{\text{SH}}}{N_{\text{model}}}} + \frac{S_{\text{dep},m}^{\text{SH}}}{\sum_n^{N_{\text{model}}} \frac{S_{\text{dep},n}^{\text{SH}}}{N_{\text{model}}}} \right) \tag{21}$$

where $n$ and $m$ index the different models, which include the inverse model, MERRA-2, the 6 model ensemble members, and the 13 AeroCom models, such that $N_{\text{model}} = 21$. Further, $S$ denotes the RMSE of a model simulation with the DAOD (subscript $\tau$), surface concentration (subscript conc), and deposition flux (subscript dep) data sets on the annual timescale. These data are split into data sets for the Northern Hemisphere (superscript NH) and Southern Hemisphere (superscript SH). For the SH, there are no accurate observational constraints on DAOD available (see Sect. 2.3) so we calculate the error relative to only the surface concentration and deposition flux data sets. Note that $\Phi_m$ is defined such that $\Phi_m = 1$ implies that a model is average among the 21 models in reproducing the global dust cycle. The lower $\Phi_m$ is, the more accurately it reproduces measurements and observations of the various aspects of the global dust cycle.

## 4. Results

We first evaluate our methodology by verifying that the inverse model obtains improved agreement against the observed regional DAOD used in the inverse model (Sect. 4.1). We then obtain the predictions of the inverse model for the main properties of the global dust cycle, namely DAOD, dust emission, dust column loading, dust surface concentration, and dust deposition flux (Sect. 4.2). Subsequently, we evaluate whether the integration of observational constraints on dust properties and abundance indeed yields an improved representation of the global dust cycle by comparing our results against independent measurements and observations in the NH (Sect. 4.3.1) and the SH (Sect. 4.3.2).

### 4.1. Evaluation of inverse model results against observed regional DAOD

To verify the viability of our methodology, we first compare the inverse model's DAOD against the observationally constrained seasonal DAOD of 15 regions (Table 2). As is expected from the inverse modeling methodology, the error is substantially reduced compared to the unmodified ensemble of simulations for all seasons (Figs. 3a-d). This decrease in error is particularly pronounced over North Africa, which we characterized using three different source regions (western North Africa, eastern North Africa, and the Sahel; Fig. 2a), and which shows a decrease in the RMSE of a factor of approximately three to five, depending on the season. Note that the DAOD in the mid-Atlantic region is nonetheless systematically underestimated by both the models in our ensemble and the inverse model. This is a common problem in models that is likely in part due to too fast removal in models (Ridley et al., 2012; Yu et al., 2019). The RMSE over the relatively minor dust source regions of North America, Australia, South America, and Southern Africa is similarly reduced by about a factor of five. For the East Asia and Middle East & Central Asia regions, the decrease in RMSE is about a factor of one-and-a-half to two. This relatively smaller decrease in the

RMSE likely occurs because we used only one source region each for both these relatively extensive source regions. Consequently, our procedure is unable to eliminate some biases of the model ensemble in these regions, such as an underestimation of DAOD in the Thar desert, which could be due to model underestimations of emissions in this region (Shindell et al., 2013). Future work could thus improve upon our results by using more source regions to better constrain the contributions of the Middle East and Asian source regions to the global dust cycle.

Overall, our procedure achieves a substantial reduction of the total DAOD error summed over the fifteen regions, reducing the RMSE by over a factor of two, from 0.092 to 0.041. This reduction in error is expected, as our methodology minimized the error against these regional DAOD data. Moreover, we find that the reduced chi squared statistic, which is of order 1 for a model that captures observations within the uncertainties (Bevington and Robinson, 2003), is indeed less than 1 for all seasons except boreal Spring. This implies that our methodology results are in good agreement with the observational DAOD constraints. Further, the ability of the inverse model to reproduce the spatial pattern of DAOD on both seasonal (Fig. 3e) and annual (Fig. 3f) timescales is substantially improved relative to both the six models in the model ensemble and the AeroCom Phase I models, and is similar to that of the MERRA-2 dust product. This is noteworthy as many of the satellite and ground-based AOD observations upon which the observational DAOD is based have been used to inform the dust schemes in the ensemble models (Cakmur et al., 2006; Kok et al., 2014a) and have been assimilated by the MERRA-2 dust product (Buchard et al., 2017; Gelaro et al., 2017; Randles et al., 2017).

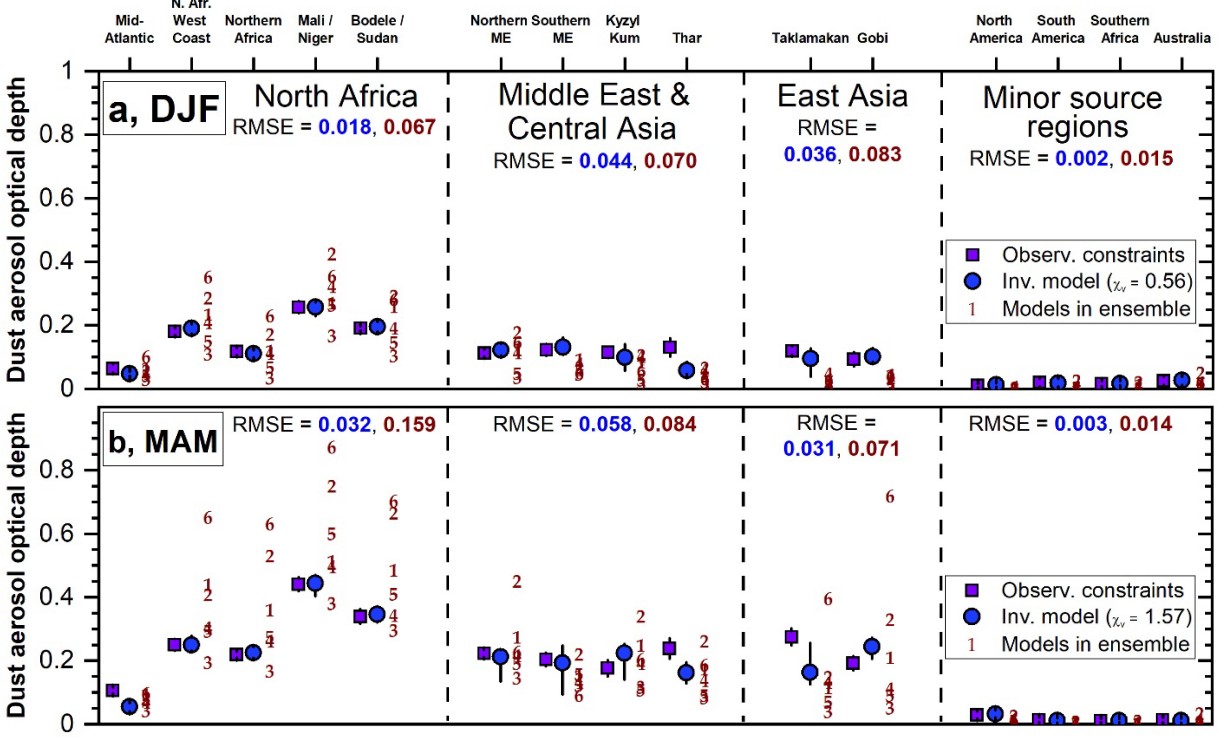

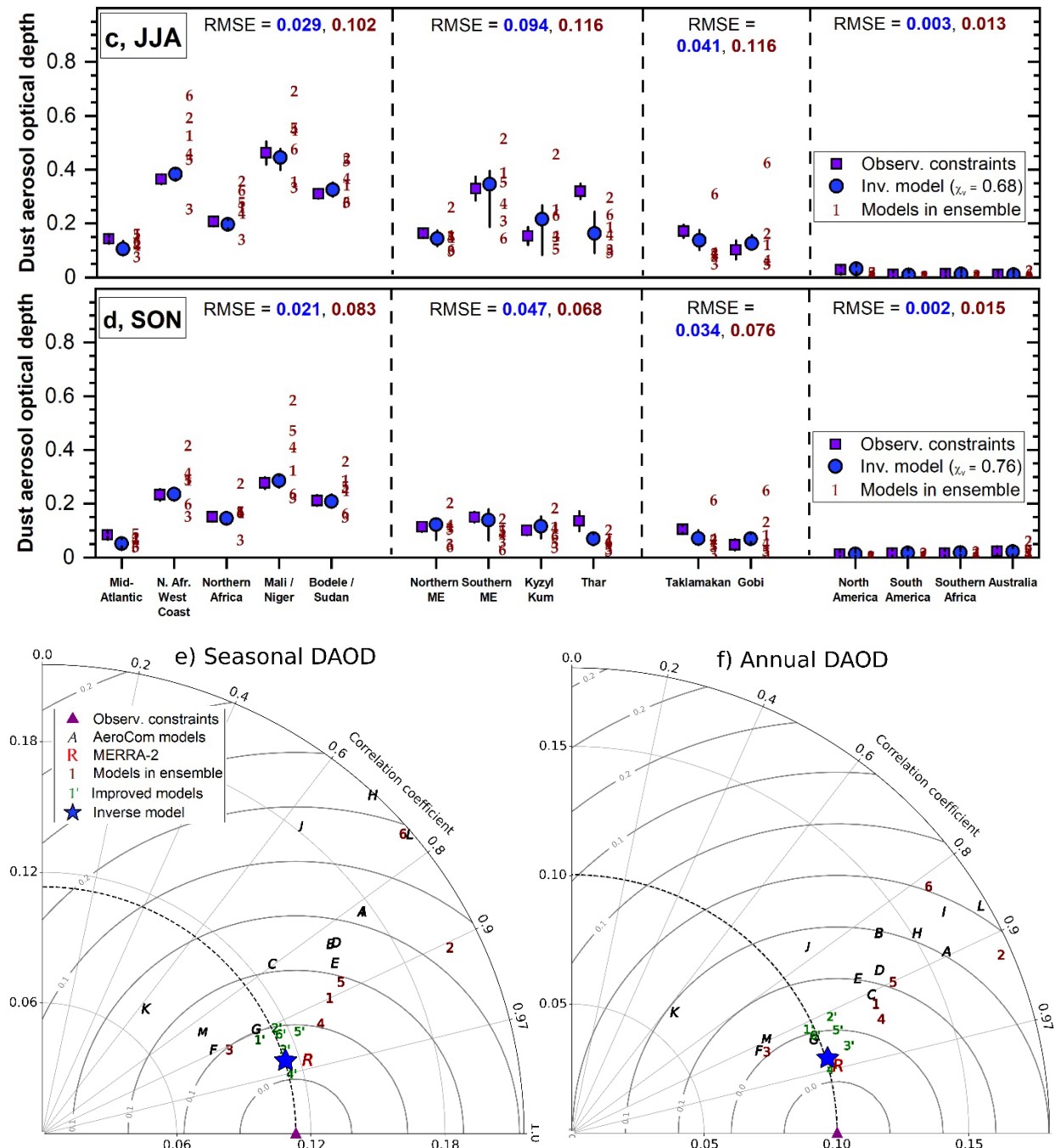

**Figure 3. Assessment of the effectiveness of the inverse model in reducing errors against observationally-informed constraints on regional dust aerosol optical depth (DAOD).** (a-d) Comparisons of the 15 observational constraints on regional DAOD (purple squares) against the inverse model results (blue circles) and the models in our ensemble (brown numbers; 1 = CESM, 2 = IMPACT, 3 = GISS ModelE2.1, 4 = GEOS/GOCART, 5 = MONARCH, 6 = INCA) for each of the four seasons. Results are grouped by the major source region nearest to each of the observed regions. Also listed are the root-mean-squared errors for each regional group for both the inverse model and model ensemble results, and the reduced chi squared metric ($\chi_v$) for the comparisons of the inverse model results against all 15 DAOD constraints. Error bars denote one standard error. (e) Taylor diagram summarizing the statistics of the comparison against the seasonally averaged regional DAOD constraints for the different models (Taylor, 2001). The different symbols represent the measurements (purple triangle), the 13 AeroCom models (black letters; A = CAM, B = GISS ModelE, C = GOCART, D = SPRINTARS, E = MATCH, F = MOZGN, G = UMI, H = LOA, I = UIO_CTM, J = LSCE, K = ECHAM5, L = MIRAGE, M = TM5), the MERRA-2 dust product (red "R"), the six models

in the model ensemble (brown numbers, as for panels a-d), the six improved model results (green numbers with a prime), and the inverse model results (blue star). The horizontal axis shows the standard deviation of the data set or model prediction, the curved axis shows the correlation, and the grey half-circles denote the centered root-mean-squared difference between the observations and the model predictions. As such, the distance between a model and the observations is a measure of the model's ability to reproduce the spatiotemporal variability in the observations; Taylor diagrams do not capture biases between model predictions and observations. (f) Same as panel (e), except showing a comparison against the annually averaged regional DAOD constraints.

## 4.2. Inverse modeling results of key aspects of the global dust cycle

We present inverse model results for the dust emission rate, DAOD, column-integrated dust loading, dust surface concentration, and dust deposition flux (Table 3, Fig. 4) and compare these inverse model results against independent measurements in Sect. 4.3. We also provide median estimates with uncertainty of the main size-resolved properties of the global dust cycle (Fig. 5).

Our results indicate that the global emission rate and loading of dust with geometric diameter $D \leq 20$ μm ($PM_{20}$) are larger than most models account for. AeroCom models reported an ensemble median global dust emission rate of 1.6 $\times 10^3$ Tg/year (one standard error range: 1.0-3.2 $\times 10^3$ Tg/year) and CMIP5 models reported a value of 2.7 (1.7-3.7) $\times 10^3$ Tg/year; both these ensembles included a mix of models simulating dust up to diameters of 10 μm or more (see Fig. S7). Our results indicate that the global emission rate of $PM_{20}$ dust is 4.6 (3.4-9.1) $\times 10^3$ Tg/year. This larger global dust emission rate is primarily due to two reasons. First, our methodology accounts for dust up to a geometric diameter of 20 μm, which is a larger size range than accounted for in many AeroCom and CMIP5 models (Huneeus et al., 2011; Wu et al., 2020; Fig. S7) and thus results in a larger bulk dust emission flux. Accounting for this larger size range is desirable because observations indicate that ~30% of $PM_{20}$ dust loading consists of super coarse dust ($D > 10$ μm) (Ryder et al., 2019; Adebiyi and Kok, 2020; Fig. 5b). Because super coarse dust has a shorter lifetime [1.0 (0.4-1.8) days; Fig. 5d] than finer dust, we find that super coarse dust accounts for ~65% of the total $PM_{20}$ dust emission flux, which corresponds to 2.9 (1.8-6.5) $\times 10^3$ Tg/year (Fig. 5a). This ~65% relative contribution of the $10 \leq D \leq 20$ μm size range is substantially larger than that inferred from size-resolved measurements of the emitted dust flux (Huang et al., 2021). In order to match *in situ* atmospheric dust size distributions, current models thus need to emit more super coarse dust than determined from measurements of the emitted dust flux, which further supports the inference from multiple previous investigations that super coarse dust deposits too quickly in atmospheric models (Maring et al., 2003; Ansmann et al., 2017; Weinzierl et al., 2017; van der Does et al., 2018). The small mid-visible (550 nm) MEE of super coarse dust [0.13 (0.12-0.15) $m^2$/g; Fig. 5e] causes it to account for only a small fraction [7.2 (5.7-9.3) %] of the total shortwave (SW) DAOD of 0.028 (0.024-0.030) (Fig. 5f and Table 3). However, dust with $10 \leq D \leq 20$ μm is nonetheless radiatively important because it accounts for a larger fraction of dust absorption of SW radiation (Tegen and Lacis, 1996; Samset et al., 2018), and because it produces ~20% of the global dust longwave (LW) DAOD of 0.014 ± 0.003 (Fig. 5h).

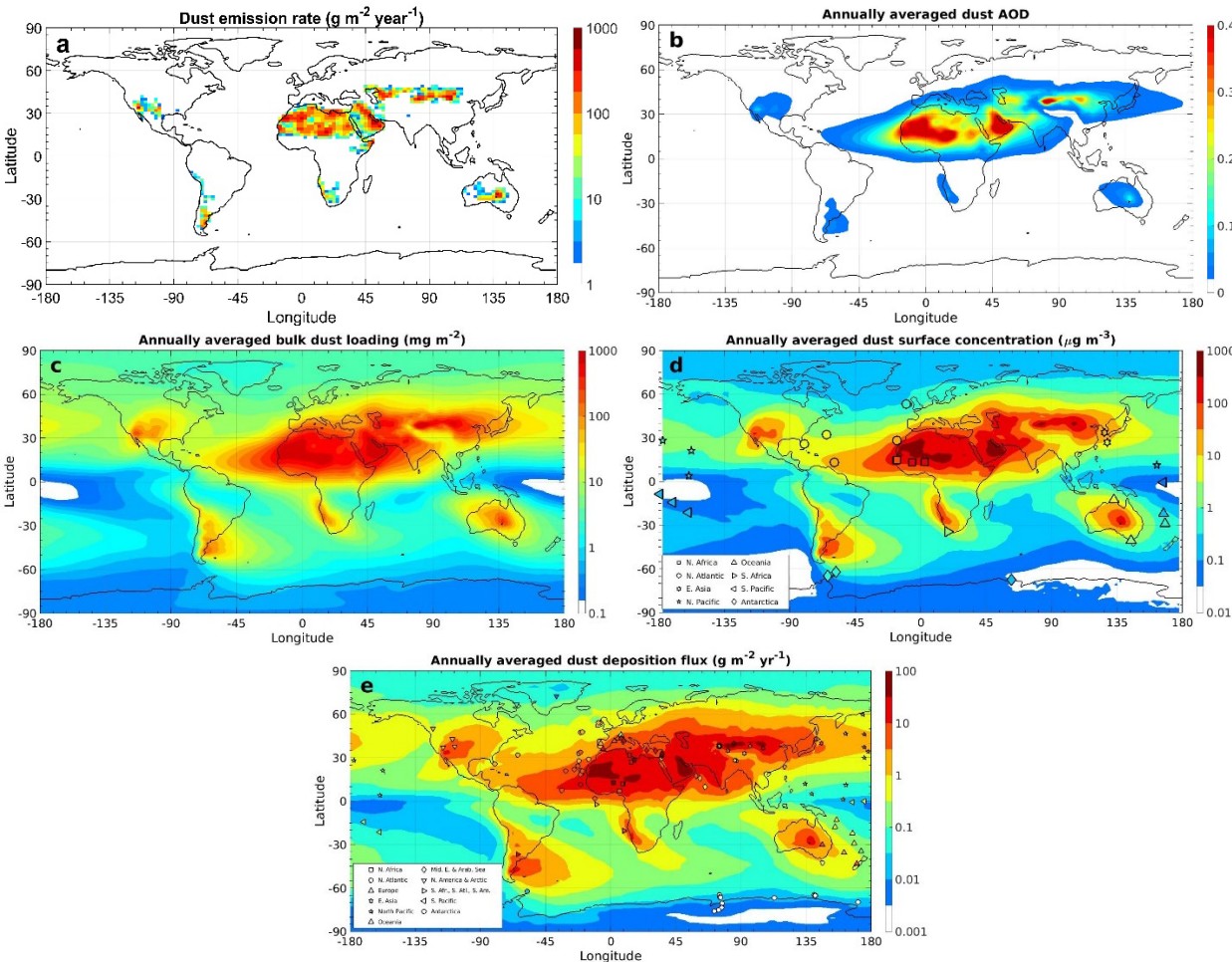

**Figure 4. Predictions of key aspects of the global dust cycle.** Shown are inverse model results for (a) annual dust emission rate, (b) annual dust AOD, (c) column-integrated dust loading, (d) dust surface concentration, and (e) dust deposition flux. Panels (a) – (d) show results for $PM_{20}$ dust, whereas panel (e) shows results for $PM_{10}$ dust, for optimal comparison against the measurement compilation of $PM_{10}$ dust deposition fluxes (Albani et al., 2014). Seasonally resolved predictions for each of these variables are shown in Figs. S2-S6. The symbols in (d) and (e) show the locations and values of the independent surface concentration and deposition flux measurements used for evaluation of the inverse model in Sect. 4.3 (also see Figs. 2c, d).

The second reason that $PM_{20}$ emission fluxes are larger than accounted for in most models is that observations have shown that many models have a bias towards fine dust (Kok, 2011b; Ansmann et al., 2017; Adebiyi and Kok, 2020). Indeed, models that do include dust up to 20 μm geometric diameter tend to underestimate the global $PM_{20}$ dust emission rate relative to our results (Fig. S7). Because coarse dust has a shorter lifetime and a lower MEE (Figs. 5e, f), correcting this fine-dust bias requires a substantially larger total emission flux to match DAOD constraints. Many of the models in our ensemble partially addressed the fine bias by using the brittle fragmentation theory parameterization for the emitted dust flux, which is substantially coarser than other emitted dust size distributions (Kok, 2011b). This causes our model ensemble to show a larger emission flux (3.5 (2.7-5.2) × 10³ Tg/year) than AeroCom models (1.6 (1.0-3.2) × 10³ Tg/year), although this increase is also due to these more recent models simulating dust out to larger particle diameters (Fig. S7). More recent work has used dozens of *in situ* measurements to show that the fine-dust bias in models is even more substantial than previously reported, and specifically that the atmospheric loading of coarse dust with $D > 5$ μm is several times greater than accounted for in most models (Adebiyi and Kok, 2020). Generating this even greater loading of coarse dust thus requires a correspondingly larger emission flux (Table 3; Fig. 5a). Emission fluxes would be even larger if the maximum size range was extended further to include dust with $D > 20$ μm, which measurements indicate is abundant close to source regions and might be important for interactions with longwave radiation (Ryder et al., 2013; Ryder et al., 2019; Figs. 5g, h). As previously reported by Adebiyi and Kok (2020), accounting for the substantial atmospheric loading of coarse dust






with $5 \leq D \leq 20$ μm also drives a larger total dust loading, increasing from 20 (12 - 24) Tg obtained by AeroCom
models and 17 (14-36) Tg obtained by CMIP5 models to 26 (22-30) Tg obtained here (Table 3). Since models
indicate that the atmospheric loading of non-dust aerosols is around 10 Tg (Textor et al., 2006; Gliss et al., 2021),
dust is likely by far the most dominant aerosol species by mass, accounting for approximately three quarters of the
atmosphere's total particulate matter loading.

The constraints on the global dust cycle obtained here are strongest on the DAOD, because our inverse model
minimizes error with respect to observed regional DAOD (Sect. 4.1). The inverse model then relies on observational
constraints on the globally averaged dust size distribution and extinction efficiency to link the DAOD to loading per
source region (Sections 2.2 and 2.3), which adds further uncertainty to our inverse model results. Constraints on dust
emission and deposition fluxes are still more uncertain, because these further depend on results from the ensemble
of models, such as the spatial pattern of emission within individual source regions, transport, and the size-resolved
dust lifetime. The lifetime of coarse dust shows especially large variability between models, which adds
substantially to the uncertainty in $PM_{20}$ emission and deposition fluxes because coarse dust dominates these fluxes
(Figs. 5a, b). Consequently, the relative uncertainties in global emission and deposition fluxes are several times
larger than the relative uncertainty in DAOD (Table 3).

**Table 3. Globally integrated annual dust emission rate, loading, DAOD, and mass extinction efficiency.** Listed are median
values, with one standard error ranges listed in parentheses. Also shown are AeroCom Phase I results, which were taken from
Table 3 in Huneeus et al. (2011), and the one standard error range was obtained by eliminating the two highest and lowest values.
This leaves the 10 central values of the 14 model results, which corresponds to the central 71% of model results. The CMIP5
results for the global dust emission rate and loading were obtained from the analysis of CMIP5 models with prognostic dust
cycles by Wu et al. (2020; see their Table 3), who did not analyze DAOD and mass extinction efficiency. For the CMIP5
ensemble we similarly eliminated the four extreme values, leaving the 11 central values of the 15 model results, which
corresponds to the central 73% of model results. For our own model ensemble, we eliminated the two extreme values, leaving the
four central values of the six model results, which corresponds to the central 67% of model results. Inverse model results are
listed for both PM10 and PM20 dust, whereas the size range accounted for by AeroCom and CMIP5 models differs for each
model (see Huneeus et al. (2011), Wu et al. (2020), and Fig. S7). DAOD and MEE were taken at 550 nm.

| Source | Annual dust emission and deposition rate ($\times 10^3$ Tg/year) | Dust loading (Tg) | DAOD | Mass extinction efficiency (m²/g) |
|---|---|---|---|---|
| AeroCom ensemble | 1.6 (1.0-3.2) | 20 (9-26) | 0.029 (0.021-0.035) | 0.65 (0.56-0.96) |
| CMIP5 ensemble | 2.7 (1.7-3.7) | 17 (14-36) | N/A | N/A |
| Model ensemble | 3.5 (2.7-5.2) | 31 (28-35) | 0.028 (0.025-0.031) | 0.44 (0.40–0.51) |
| Inverse model $PM_{2.5}$ | 0.22 (0.19 – 0.27) | 4.4 (3.8-5.0) | 0.014 (0.012-0.016) | 1.63 (1.50-1.80) |
| Inverse model $PM_{10}$ | 1.8 (1.2-2.9) | 18 (16-21) | 0.025 (0.022-0.028) | 0.70 (0.63-0.79) |
| Inverse model $PM_{20}$ | 4.7 (3.3-9.0) | 26 (22-31) | 0.028 (0.024-0.030) | 0.54 (0.46-0.62) |


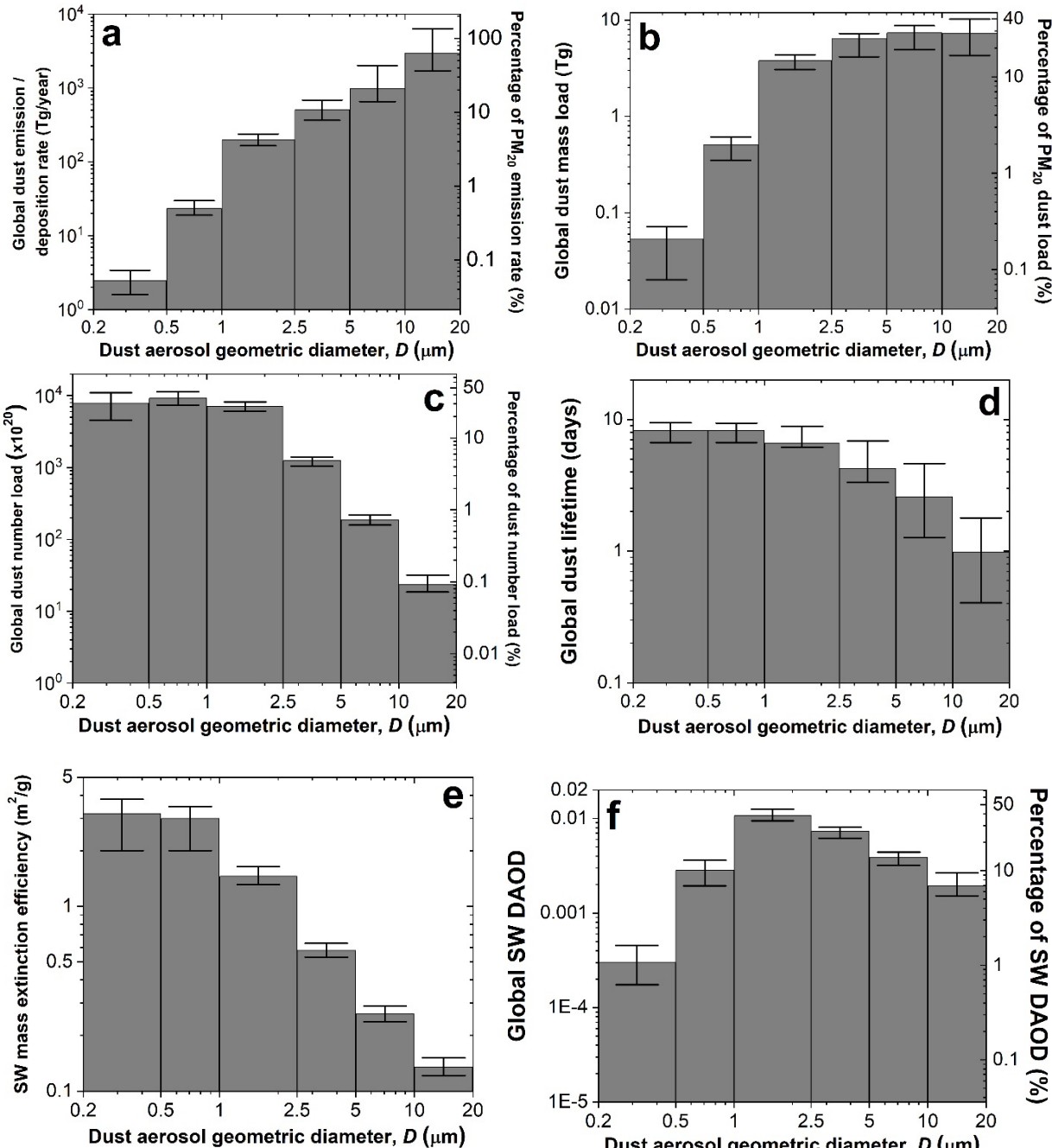


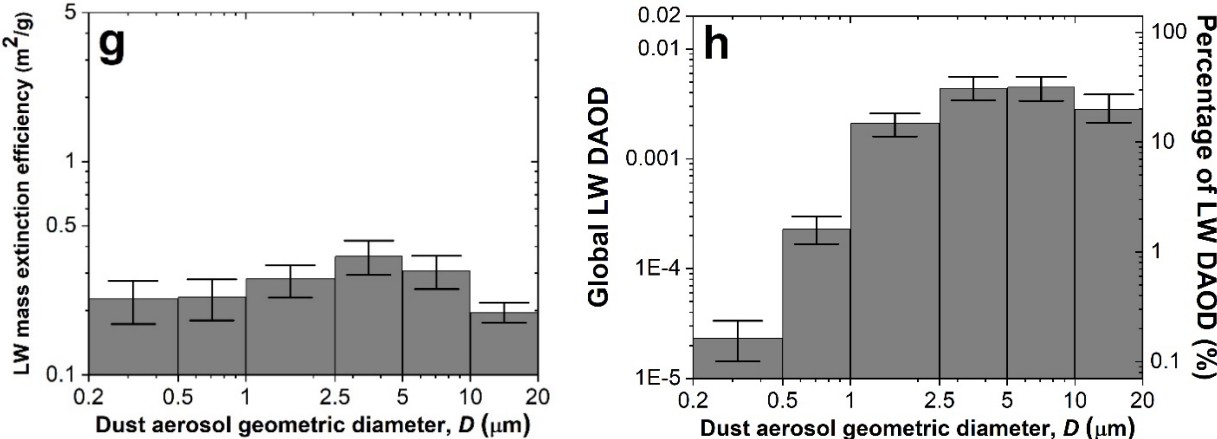

**Figure 5. Size-resolved properties of the global dust cycle.** Shown are the size-resolved (a) global dust emission rate (which equals the global dust deposition rate), (b) global dust loading in terms of mass per size bin, (c) global dust loading in terms of number of particles per size bin, (d) global dust lifetime, (e) dust mass extinction efficiency at 550 nm, (f) global DAOD at 550 nm, (g) dust mass extinction efficiency at 10 μm, and (h) global DAOD at 10 μm. The right axis of panels (a), (b), (c), (f), and (h) shows the fraction of each dust cycle property that is accounted for by each size bin, which was obtained by dividing the simulated quantity in each bin by the median total for all bins. For panels (e) and (f), we used the constraint on extinction efficiency at 550 nm from Kok et al. (2017); for panels (g) and (h), we obtained a constraint on extinction efficiency at 10 μm following the methodology of Kok et al. (2017), using probability distributions of dust shape descriptors obtained by Huang et al. (2020), and setting the real index of refraction to $1.70 \pm 0.20$ and the logarithm of the imaginary index to $-0.40 \pm 0.11$, both based on a compilation of measurements by Di Biagio et al. (2017). Error bars denote one standard error.

### 4.3. Performance of inverse model results against independent measurements

After obtaining inverse model results for key aspects of the global dust cycle, we next evaluate the accuracy of this representation of the global dust cycle using independent measurements of dust surface concentration and dust deposition fluxes (see Sect. 3.1). We divide these results into comparisons for the NH (Sect. 4.3.1) and the SH (Sect. 4.3.2). We do this because we have observationally informed constraints on DAOD for eleven NH regions, and therefore expect the inverse model results to show relatively good agreement against independent measurements in the NH. In contrast, we do not have observationally constrained DAOD for the SH; instead, the inverse model used an ensemble of reanalysis products, whose ensemble members might have similar biases as they assimilate similar remote sensing data sets. As such, we expect the inverse model results to show less agreement against independent measurements in the SH.

### 4.3.1. Performance of the inverse model results against independent measurements in the Northern Hemisphere

The inverse model results accurately reproduce the seasonal variation in surface dust at individual sites in the NH, capturing all the measurements within the uncertainties (Fig. 6). The inverse model results show an average correlation coefficient of $r = 0.90$ with the seasonally averaged measurements at the different sites, which exceeds the average correlation coefficient of models in our ensemble ($r = 0.85$), in the AeroCom ensemble ($r = 0.61$), and the MERRA-2 dust product ($r = 0.86$). The inverse model results also accurately reproduce the spatial variation in dust surface concentration among different locations, as shown by scatter plots comparing predicted and observed surface concentrations on seasonal (Fig. 7a) and annual (Fig. 7b) timescales. These plots also show that the inverse model reproduces concentration measurements on both seasonal and annual timescales well within the uncertainties, with values of the reduced chi square statistic ($\chi_v^2$; see Section 3.1) of 0.65 on the seasonal time scale and 0.18 on the annual time scale.

This strong agreement between the inverse model results and dust surface concentration is a notable improvement over any of the six models in our model ensemble, any of the 13 AeroCom Phase 1 models, and the MERRA-2 dust product. The strong performance of the inverse model is due to its improved ability to capture spatial variability in seasonal and annual dust concentration, as quantified by Taylor diagrams in Figures 7d and 7e, and because the

inverse model results show almost no bias against seasonally and annually averaged concentration measurements (Figs. 8a, b). This lack of bias in capturing the mean dust aerosol state also represents a substantial improvement over models, which show biases of up to approximately ±0.3 in logarithmic space, corresponding to a bias of up to a factor of ~2 in linear space. The inverse model's reduction in bias and improved representation of spatiotemporal variability of dust surface concentration combine to produce RMSEs (in log space) of only ~0.22 (~65% relative error) against seasonally averaged and ~0.12 (~30% relative error) against annually averaged dust surface concentration measurements (Figs. 8c, d). Compared to individual models and MERRA-2, this represents a reduction by a factor of ~1.5-5 in error in log space and a reduction by a factor of ~2-10 in relative error.

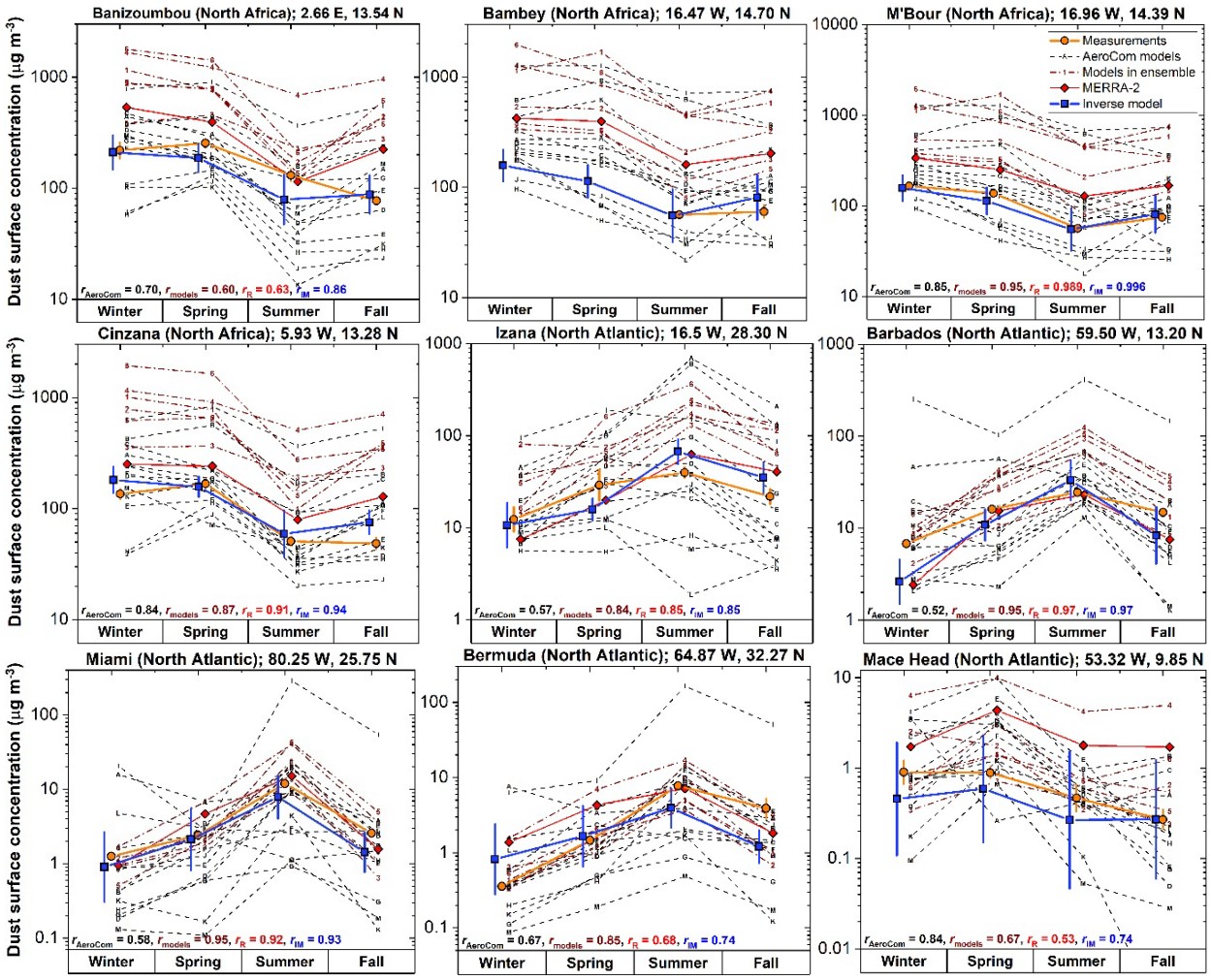

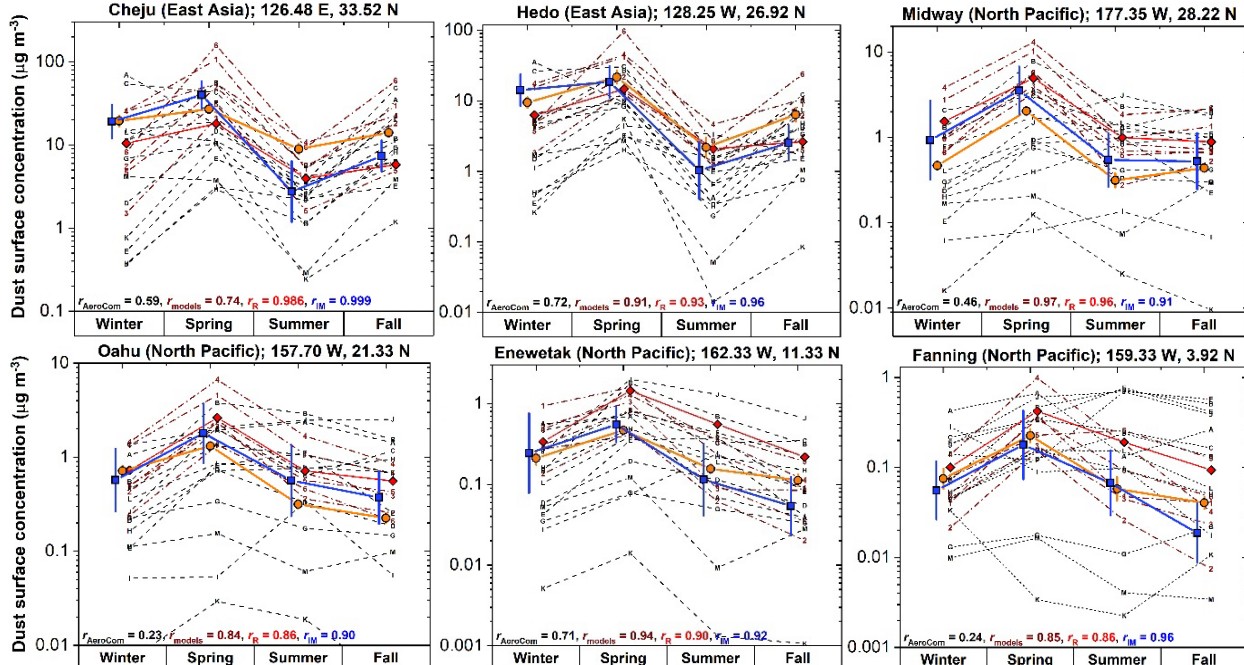

**Figure 6. Comparison of measured and modeled seasonally averaged dust surface concentrations at 15 Northern Hemisphere stations.** The inverse model results (blue line and squares) capture the measured seasonal variability (orange line and circles) at all stations, with lower error (see Fig. 8c) and on average higher correlation coefficients than MERRA-2 (red line and diamonds), models in the AeroCom ensemble (black dotted lines and letters), and (unmodified) models in our ensemble (brown dashed lines and numbers). Also shown are the mean correlation coefficients between measurements and the different AeroCom models ($r_{AeroCom}$) and the different models in our ensemble ($r_{models}$), and the correlation coefficients for MERRA-2 ($r_R$) and the inverse model results ($r_{IM}$). Uncertainty ranges on measurements and the inverse model results represent one standard error on the climatological seasonally averaged surface concentration. The legend for individual models is given in Fig. 3, and x-values are offset slightly for clarity.

We find that the inverse model results also show good agreement against the compilation of NH deposition flux measurements (Fig. 7c). The scatter between measurements and model predictions of deposition fluxes is about an order of magnitude larger than for the comparison against surface concentration measurements. This is partially driven by substantial model errors in deposition (Ginoux, 2003; Huneeus et al., 2011; Yu et al., 2019; Huang et al., 2020), and partially driven by the large experimental (e.g., Edwards and Sedwick, 2001) and representation errors (Schutgens et al., 2017) indicated by the large spread between measurements in similar locations (Figs. 4d, 7c; Section 3.1). Nonetheless, the inverse model reproduces the deposition measurements within these uncertainties, as quantified by the reduced chi squared value of 1.13. The inverse model also reproduces the spatial pattern of deposition flux better than most models (Fig. 7f). Additionally, whereas models in our ensemble and the AeroCom models show biases against deposition flux measurements of up to approximately ±0.5 in logarithmic space, which corresponds to a bias of up to a factor of ~3 in linear space, the inverse model results show a bias close to zero (Fig. 8a, b). Overall, the inverse model results show an RMSE of ~0.58, which matches that of the best performing models, and is less by ~5-25% relative to other models.

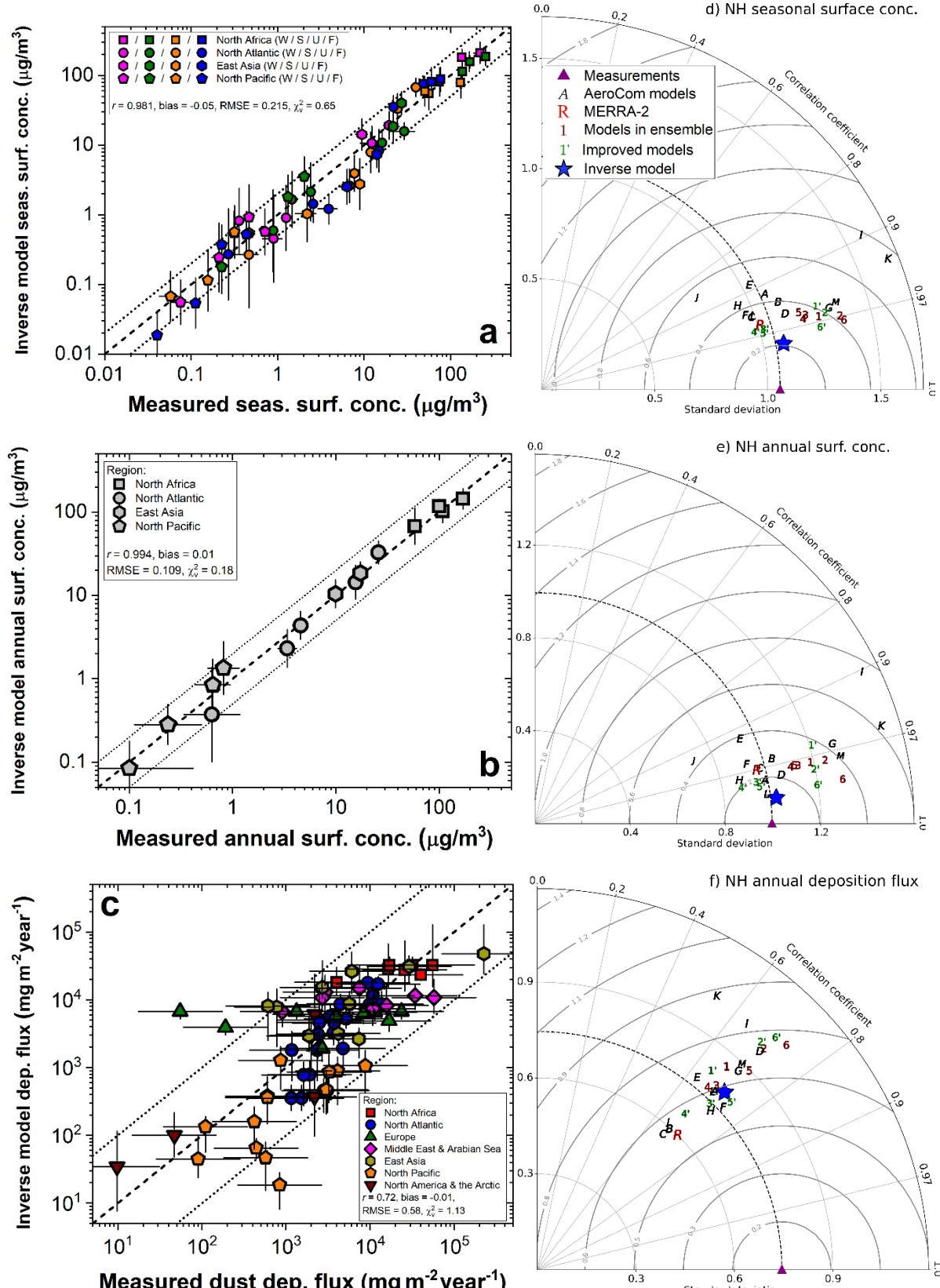

**Figure 7. Evaluation of the inverse model results against independent measurements of surface concentration and deposition flux in the Northern Hemisphere.** Shown are comparisons of inverse model results against (a) seasonally averaged (Winter, Spring, Summer, and Fall are respectively denoted by magenta, green, orange, and blue) and (b) annually averaged dust surface concentration measurements at 15 NH stations, and against (c) a compilation of 77 measurements of the dust deposition flux. Results are grouped by regions as shown in Fig. 2. Statistics of the comparisons are noted in the figures and are calculated in log-space because the measurements span several orders of magnitude. Uncertainties on inverse model results and measurements represent one standard error and are calculated as described in Sects. 2.5 and 3.1, respectively. Also shown are Taylor diagrams summarizing the statistics of the ability of the different models to reproduce the spatial variability in the measured fields of (d) seasonal and (e) annual surface concentration, and (f) dust deposition flux (Taylor diagrams do not capture biases between model predictions and observations). The different symbols represent the measurements (purple triangle), the 13 AeroCom models (black letters), MERRA-2 (red "R"), the six models in the model ensemble (brown numbers), the six improved models (green numbers with prime), and the inverse model results (large blue star). An exact legend for the different models is provided in Fig. 3.

We further explore the merit of our inverse modeling approach by analyzing the "improved model" results (Sect. 2.5), which represent output from each of the individual model ensemble members that was corrected using observational constraints on dust properties and abundance (Sect. 2). For each of the six ensemble members we find that the inverse modeling procedure reduces errors against both NH dust surface concentration and deposition flux measurements, with reductions ranging from a few percent to well over a factor of two (Figs. 8c, d). As with the inverse model results, for most models this is due to both an improvement in the representation of the spatiotemporal variability of dust surface concentration and deposition flux (Figs. 7d-f) and a reduction in the bias against both sets of measurements (Figs. 8a, b).

The comparison against independent measurements thus indicates that the inverse model results represent the NH dust cycle more accurately than both MERRA-2 and a large number of climate and chemical transport models. This is quantified in Fig. 8e, which shows the normalized model error for the various models and model ensembles. We find that the inverse model results show a normalized error of 0.49, which is well below that of the mean of models in our ensemble (1.08) and the AeroCom ensemble (1.22), and also below the MERRA-2 normalized error (0.62). Moreover, we find that the average normalized error of improved models is substantially less (0.72) than for the unmodified models in our ensemble. These results indicate that our approach of integrating observational constraints on dust properties and abundance is effective in improving model accuracy.

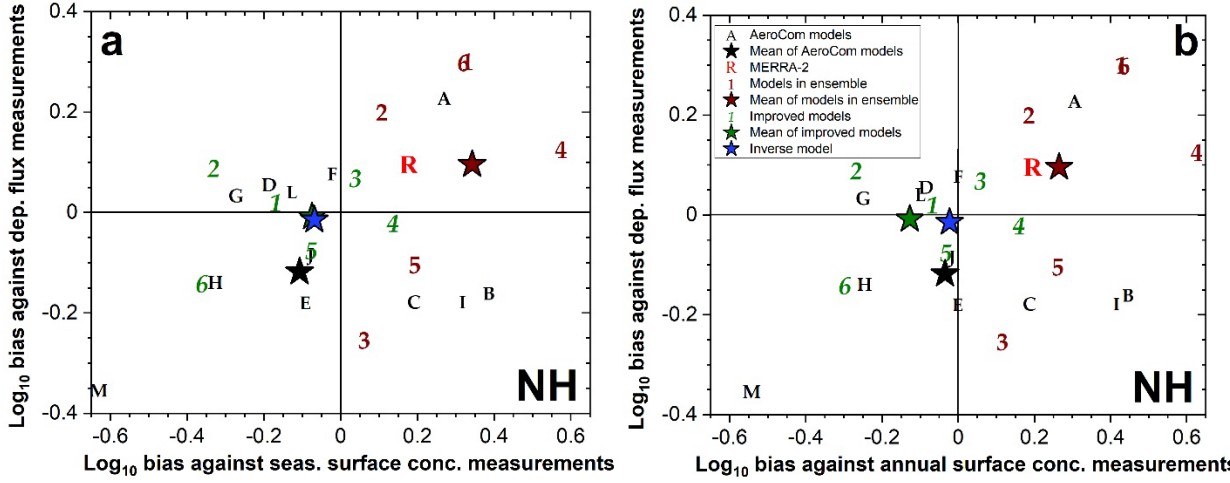

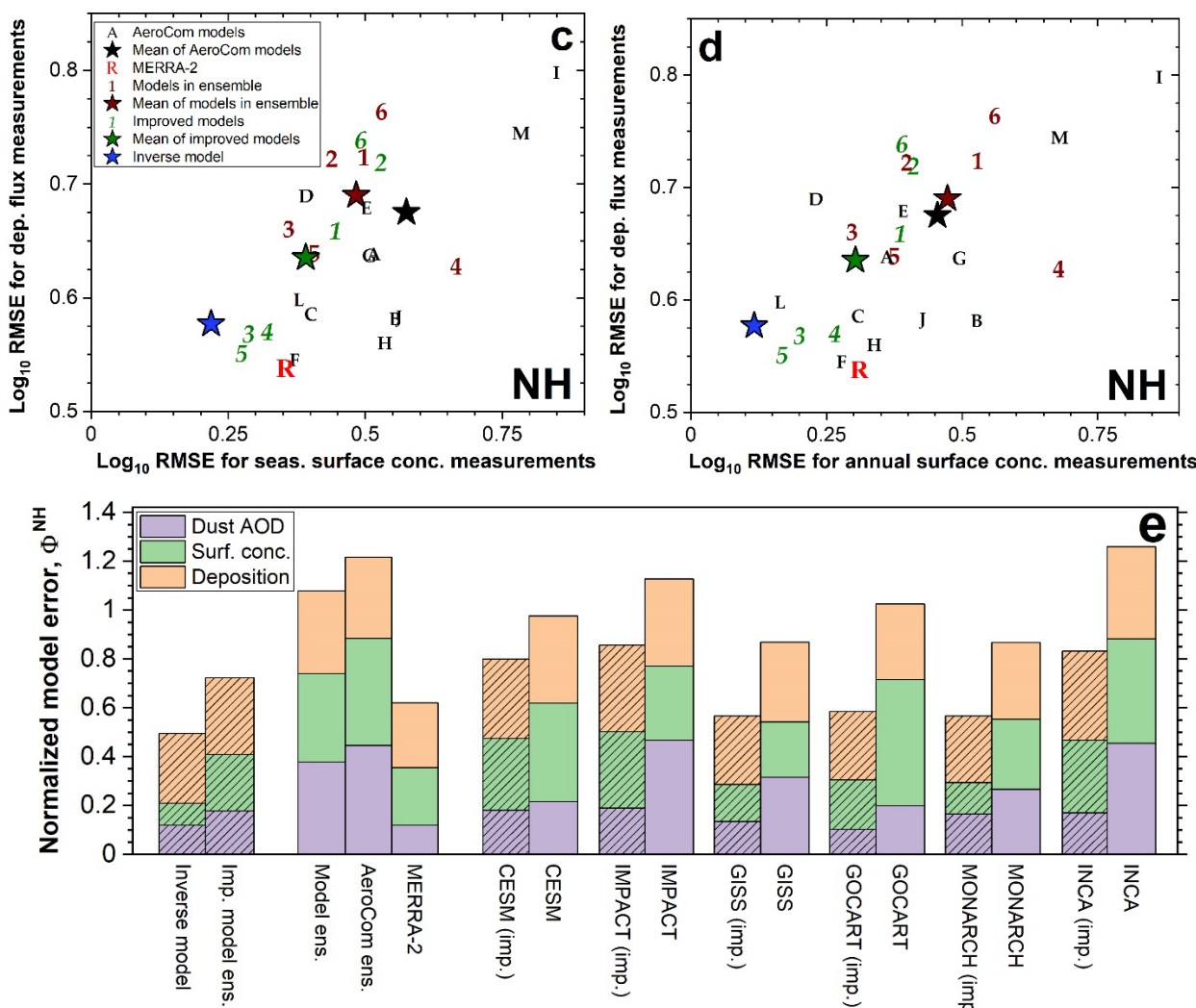

**Figure 8. Evaluation of whether integrating observational constraints on dust properties and abundance produces an improved representation of the Northern Hemisphere dust cycle.** Shown are the biases (top panels) and root-mean-squared errors (RMSEs; middle panels) in logarithmic space with respect to measurements of (a and c) seasonally averaged dust surface concentration and dust deposition flux and of (b and d) annually averaged dust surface concentration and deposition flux. Symbols in panels (a)-(d) denote results for the individual models in the AeroCom ensemble (black letters), MERRA-2 (red "R"), models in our ensemble (brown numbers), and the improved models (green italicized numbers). The exact legend for the different models is given in Figure 3, and stars denote the mean bias and RMSE for AeroCom models (black star), models in our ensemble (brown star), the improved models (green star), and the inverse model results (blue star). Panel (e) shows normalized model errors (see Sect. 3.2) relative to the DAOD (purple bars), surface concentration (green bars), and deposition flux (orange bars) data sets. Shown are results for the inverse model; the average of models in the AeroCom ensemble, our model ensemble, and our ensemble of improved models; MERRA-2; and for the individual models in our ensemble before and after applying observational constraints (see Sect. 2.5). Hatched bars denote results of the inverse model and improved models obtained through our methodology. The reductions in bias, RMSE, and normalized error for the inverse model and improved models relative to the individual models and MERRA-2 imply that the integrational of observational constraints on dust properties and abundance improve the representation of the NH dust cycle.

### 4.3.2.  Performance of inverse model results against independent measurements in the Southern Hemisphere

After analyzing the performance of the inverse model results in the Northern Hemisphere, we next analyze the performance of the inverse model results in the Southern Hemisphere. We expect less agreement against

independent measurements than in the NH because the SH DAOD constraints are of substantially lower quality (see Sect. 2.3).

The agreement of the inverse model results against independent data in the SH varies substantially between stations
and regions. The inverse model has difficulty reproducing the seasonality in surface concentration at many SH stations (Fig. 9), which could indicate that long-range transport is not well captured as most stations are remote from the main dust source regions (Fig. 2c). The inverse model results do produce good quantitative agreement against dust surface concentration measurements close to the Australian and Southern African source regions yet somewhat underestimate deposition fluxes in those regions (Figs. 9, 10a-c). Furthermore, the inverse model results
underestimate both the dust surface concentration and the deposition flux in the South Pacific, suggesting an underestimate of dust transport to this region. For Antarctica, the results are contradictory in that the inverse model results underestimate measurements of dust surface concentrations yet overestimate measurements of dust deposition fluxes. Overall, the inverse model might slightly underestimate errors of dust fields in the SH, as indicated by reduced chi squared values that are somewhat larger than 1 ($\chi_\nu^2$ = 1.32, 1.40, and 2.24 for of seasonal
surface concentration, annual surface concentration, and deposition flux, respectively). This possible underestimation of error might be due to systematic biases in the constraints on DAOD in the SH, as discussed further in Section 5.1.

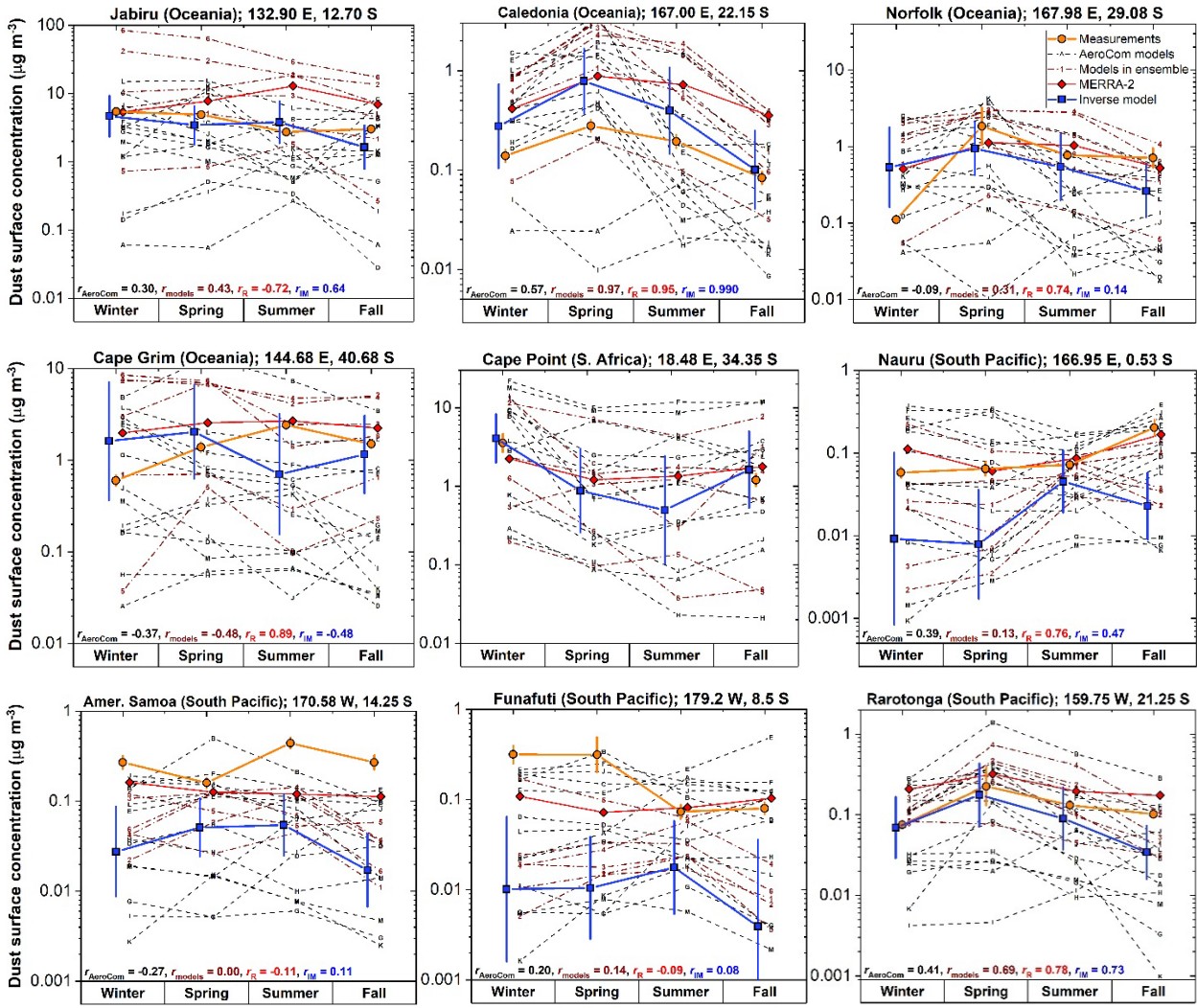

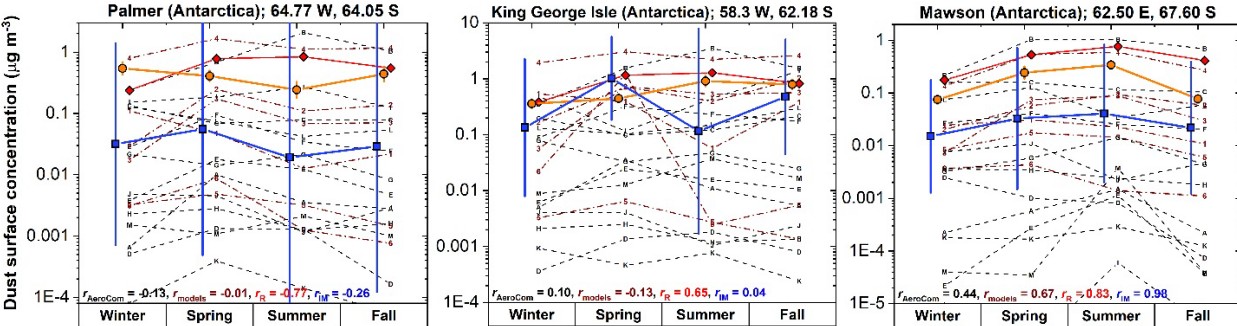

**Figure 9. Comparison of measured and modeled seasonally averaged dust surface concentrations at 12 Southern Hemisphere stations.** Shown are measurements (orange line and circles) and results from models in the AeroCom ensemble (black dotted lines and symbols) and our ensemble (brown dashed lines and symbols), and results from MERRA-2 (red line and diamonds) and the inverse model (blue line and squares). Also shown are the mean correlation coefficients between measurements and the different AeroCom models ($r_{AeroCom}$) and the different models in our ensemble ($r_{models}$), and the correlation coefficients for MERRA-2 ($r_R$) and the inverse model results ($r_{IM}$). Uncertainty ranges on the inverse model results and measurements represent one standard error on the climatological seasonally averaged surface concentration. The legend for individual models is given in Fig. 3, and x-values are offset slightly for clarity.

The underestimation of dust surface concentration but overestimation of deposition fluxes in Antarctica is puzzling (Figs. 10a-c). Indeed, many individual models show similar results (Figs. S11-S13; also see Huneeus et al. (2011), Wu et al. (2020), and (Checa-Garcia et al., 2020)). One possible explanation is large model errors in the conversion of dust concentrations to deposition fluxes, which is known to be one of the most uncertain aspects of global dust cycle simulations (Huneeus et al., 2011). This is particularly the case for regions dominated by wet deposition, which is a challenge for models to simulate accurately, in part because it depends on modeled precipitation, which itself can have large uncertainties (Huneeus et al., 2011; Mahowald et al., 2011a). Additionally, the inverse model and most individual models do not include high latitude dust emissions, which could cause additional errors for comparisons against measurements in Antarctica (Bullard et al., 2016). Another possibility is that measurements do not accurately represent either the dust surface concentration or the deposition fluxes. In particular, all but one of the Antarctic dust fluxes are derived from measurements of total-dissolvable iron in snow and ice, for which the conversion to the deposited dust flux involves many uncertainties (Edwards and Sedwick, 2001; Mahowald et al., 2009), and it is possible that this methodology systematically underestimates dust deposition fluxes (Huneeus et al., 2011). Another factor that could cause disagreement between the inverse model results and measurements might be a mismatch in timescales. The inverse model results characterize the dust cycle for the years 2004-2008, whereas the concentration data were taken for different dates in the period 1981–2000 (Prospero et al., 1989; Arimoto et al., 1995) and the deposition flux measurements were taken one to several decades earlier (Edwards et al., 2006; McConnell et al., 2007). This mismatch in time periods could cause modeled deposition fluxes to exceed measured fluxes as several studies have reported increases in dust emissions from South America and in dust deposition at Antarctica over the past century or so (McConnell et al., 2007; Gasso and Torres, 2019; Laluraj et al., 2020). Furthermore, there is substantial interannual variability in dust concentration that could affect the mismatch in time between models and measurements, especially for less dusty regions such as in the SH (Smith et al., 2017). Comparisons against measurements in previous studies have suffered from similar mismatches in time periods (Huneeus et al., 2011; Albani et al., 2014; Colarco et al., 2014; Kok et al., 2014a).

The ability of the inverse model to reproduce the spatial distribution of surface concentration and deposition measurements is thus less good in the SH than in the NH. However, despite the decreased agreement against independent measurements, the inverse model performs better than most of the individual models in our ensemble and in the AeroCom ensemble (Figs. 9, 10d-f, 11). The inverse model, the individual models, and the MERRA-2 results all show biases against SH surface concentration and deposition flux measurements that are substantially larger than the biases against NH measurements (Fig. 11a, b). Interestingly, the different models show a positive correlation between bias against surface concentration data and bias against deposition flux measurements, with both biases being negative for twelve of the models. This indicates that systematic under- or overestimation of SH dust are key contributors to errors against measurements, with additional errors due to difficulties in reproducing the spatial pattern of dust surface concentration and deposition fluxes (Fig. 10d-f). Consequently, almost all models show a substantially larger root mean-squared error relative to measurements for the SH than for the NH (Figs. 11c,

d). These results indicate substantial model errors in the magnitude and spatial pattern of SH dust emissions, dust transport, and/or dust deposition, and underscore the difficulties models have in capturing the SH dust cycle.

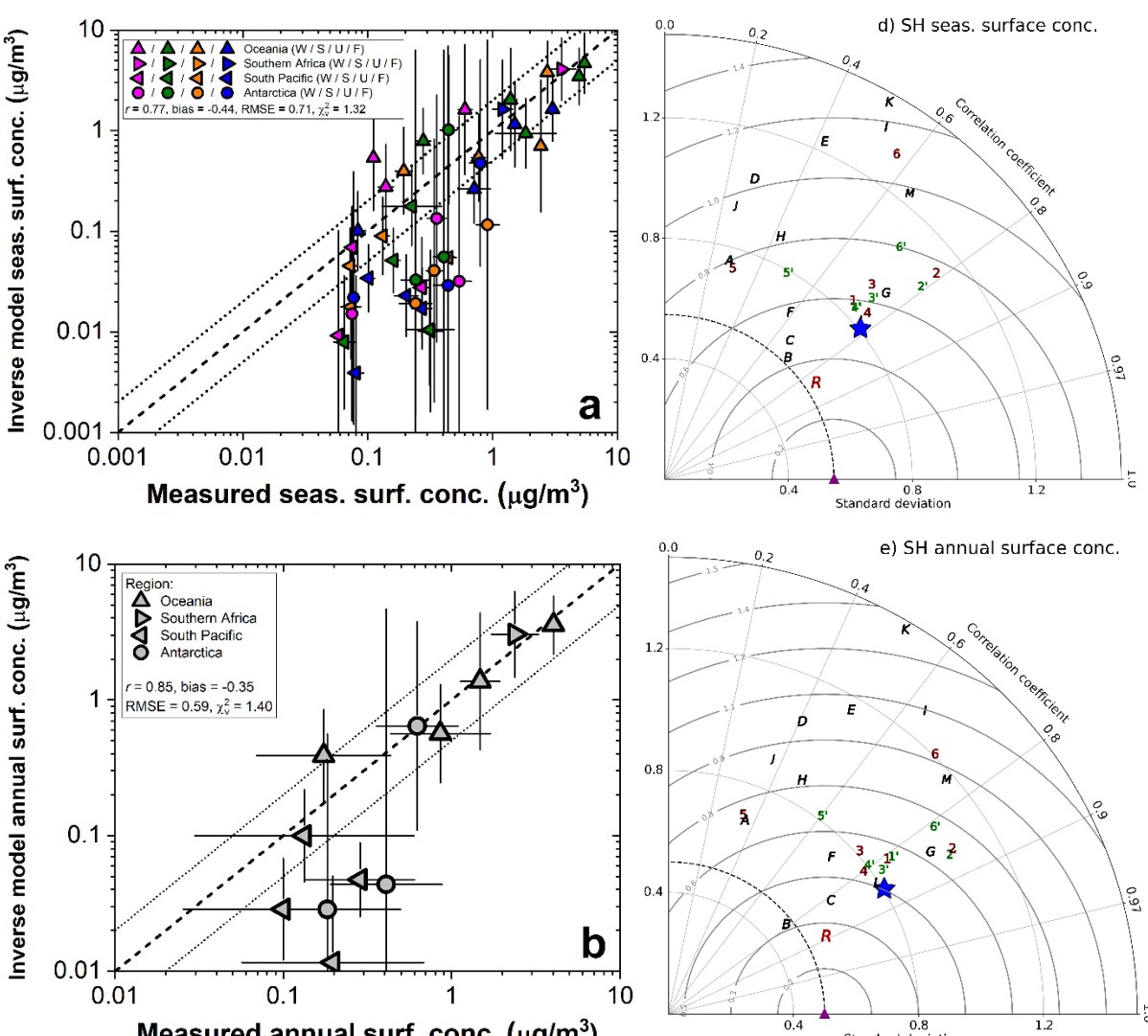

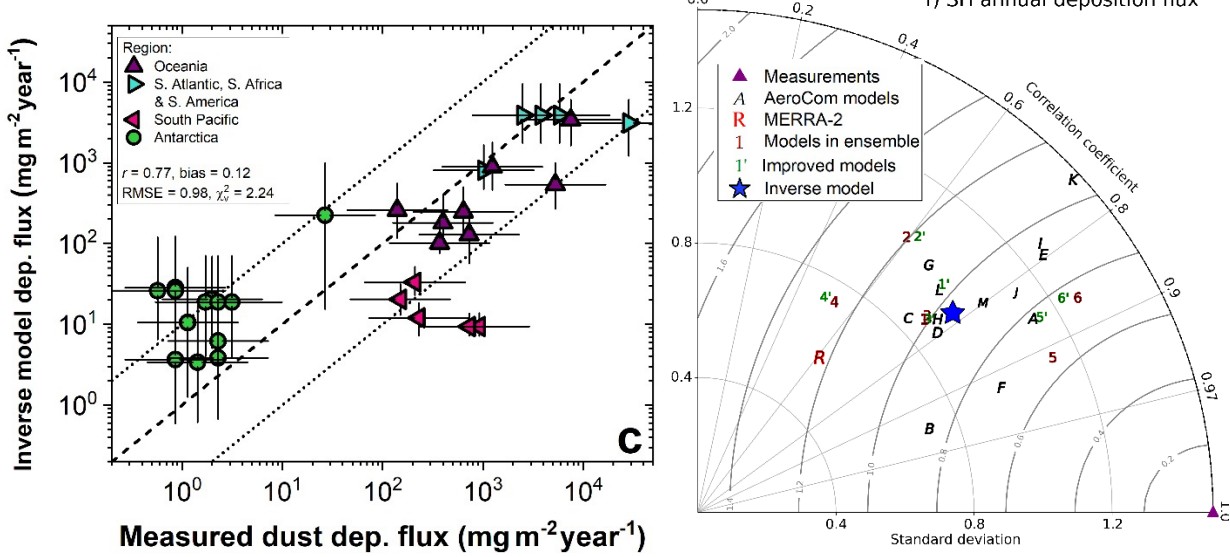


**Figure 10. Evaluation of the inverse model results against independent measurements of surface concentration and deposition flux in the Southern Hemisphere.** Shown are comparisons of the inverse model results against (a) seasonally averaged (austral Winter, Spring, Summer, and Fall are respectively denoted by magenta, green, orange, and blue) and (b) annually averaged dust surface concentration measurements at 12 SH stations, and against (c) a compilation of 33 measurements

of the dust deposition flux. Results are grouped by regions as shown in Figs. 2c and 2d. Statistics of the comparisons are noted in the figures and are calculated in log-space. Uncertainties on inverse model results and measurements represent one standard error and are calculated as described in Sects. 2.5 and 3.1, respectively. Also shown are Taylor diagrams for the (d) seasonal and (e) annual surface concentration, and (f) dust deposition flux. The different symbols represent the measurements (purple triangle), the 13 AeroCom models (black letters), MERRA-2 (red "R"), the six models in the model ensemble (brown numbers), the six

improved models (green numbers with a prime), and the inverse model results (large blue star). An exact legend for the different models is provided in Fig. 3.

Overall, the integration of observational constraints on dust properties and abundance seems to produce a modest improvement in the representation of the SH dust cycle. This is quantified in Fig. 11e, which shows that the normalized model error of the inverse model results is 0.78, which is below that of the mean of models in our model

ensemble (0.92) and the AeroCom ensemble (1.06), and below the normalized error of the MERRA-2 dust product (0.81). However, whereas the "improved model" results show clear reductions in bias, RMSE, and normalized error in the NH, they show no clear improvements in the SH (Fig. 11).

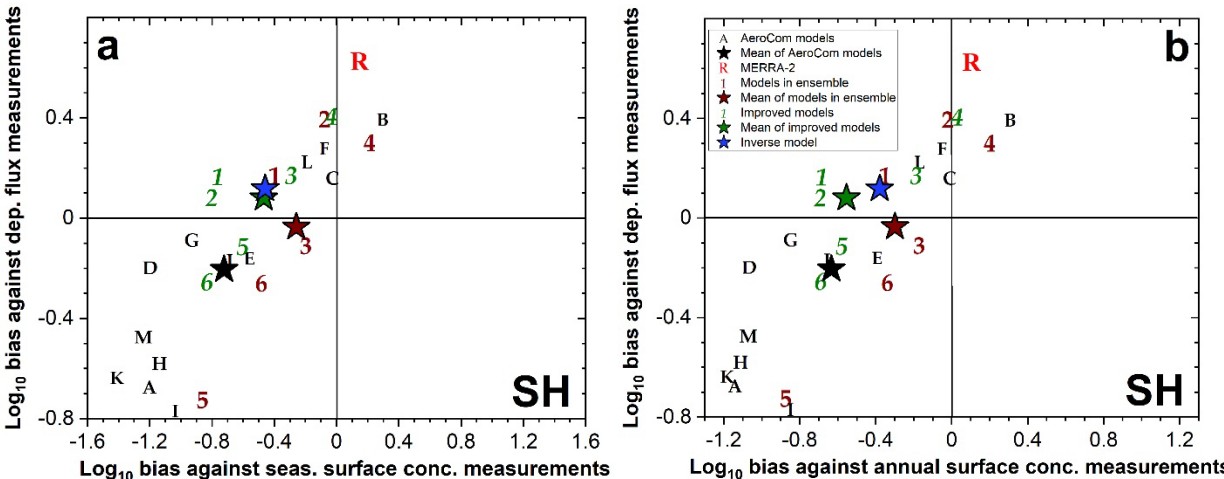

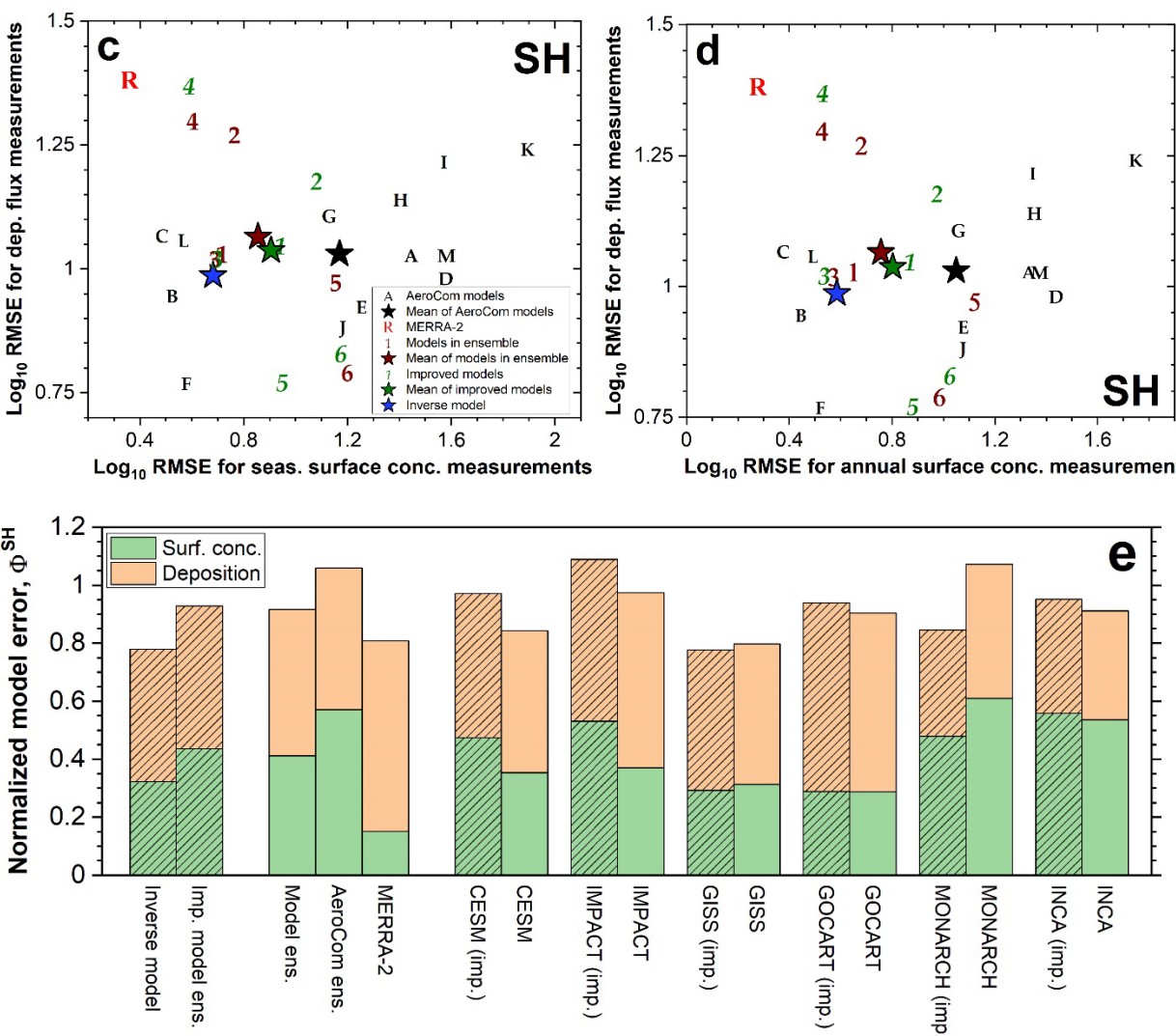


**Figure 11. Evaluation of whether integrating observational constraints on dust properties and abundance produces an improved representation of the Southern Hemisphere dust cycle.** Shown are the biases (top panels) and root-mean-squared errors (RMSEs; middle panels) in logarithmic space with respect to measurements of (a and c) seasonally averaged dust surface concentration and dust deposition flux and of (b and d) annually averaged dust surface concentration and deposition flux.

Symbols in panels (a)-(d) denote results for the individual models in the AeroCom ensemble (black letters), MERRA-2 (red "R"), models in our ensemble (brown numbers), and the improved models (green italicized numbers). The exact legend for the different models is given in Figure 3, and stars denote the mean bias and RMSE for AeroCom models (black star), models in our ensemble (brown star), the improved models (green star), and the inverse model results (blue star). Panel (e) shows normalized model errors (see Sect. 3.2) relative to the surface concentration (green bars) and deposition flux (orange bars) data sets. Shown are

results for the inverse model; the average of models in the AeroCom ensemble, our model ensemble, and our ensemble of improved models; MERRA-2; and for the individual models in our ensemble before and after applying observational constraints (see Sect. 2.5). Hatched bars denote results of the inverse model and improved models obtained through our methodology.

## 5. Discussion

Our results show that our framework for integrating observational constraints on dust properties and abundance

yields an improved representation of the global dust cycle. Relative to the model ensemble, the inverse model results show a reduction of errors against NH dust cycle measurements of over a factor of two (Fig. 8e) and modest improvements for the SH (Fig. 11e). Moreover, we have obtained a data set of the global dust cycle that is resolved

by particle size and season and that is more accurate than the MERRA-2 dust product and any of a large number of model simulations .

Below, we first discuss the main limitations of our methodology and results (Sect. 5.1). We then discuss how our results can be used to guide improvements in the representation of the global dust cycle in climate and chemical transport models (Sect. 5.2), after which we discuss the utility of the data set presented here in constraining dust impacts on the Earth system (Sect. 5.3).

## 5.1. Limitations of the methodology

Our results are subject to a few important limitations. First, although our methodology integrates observational constraints, it still relies on global model simulations to compute a number of key variables, including the spatial pattern and timing of dust emissions within each source region, the vertical distribution of dust, and the deposition flux of dust. All three of these processes are known to be subject to important model errors (e.g., Ginoux, 2003; Huneeus et al., 2011; Kim et al., 2014; Kok et al., 2014a; Evan, 2018; O'Sullivan et al., 2020). As discussed in Sect.

1, accurately simulating the magnitude and spatiotemporal variability of dust emissions represents a fundamental challenge for models. To mitigate this problem, many models prescribe prolific dust sources where geomorphologic processes concentrate fine soil particles as a result of fluvial erosion (Ginoux et al., 2001; Prospero et al., 2002; Tegen et al., 2002; Zender et al., 2003; Koven and Fung, 2008). However, these representations are highly uncertain, as indicated by large differences in the spatial patterns of emissions (Cakmur et al., 2006; Kok et al., 2014a; Wu et

al., 2020). In addition to these challenges with simulating dust emissions, many models also underestimate the height at which dust is transported (Yu et al., 2010; Johnson et al., 2012; Kim et al., 2014). Furthermore, excessive diffusion of coarse dust due to numerical sedimentation schemes causes additional problems in many models (Ginoux, 2003; Eastham and Jacob, 2017; Zhuang et al., 2018), and might be partially responsible for a general underestimation of long-range transport of coarse dust relative to measurements and satellite observations (Maring

et al., 2003; Ridley et al., 2014; Ansmann et al., 2017; Gasteiger et al., 2017; van der Does et al., 2018; Yu et al., 2019). Because of these various uncertainties in model representations of dust processes, our constraints on dust AOD and loading are strongest, and constraints on dust emission, deposition, and 3D concentration have greater uncertainty (Table 3). Furthermore, although uncertainties in the products obtained here include the error due to the spread in the results of the models in our ensemble, they do not account for systematic biases between the model

ensemble and the real world, which might be substantial in light of the problems in model simulations highlighted above. In addition, some of the other inputs to our methodology, such as the globally averaged dust size distribution (Adebiyi and Kok, 2020), would also be affected by possible biases in model results, such as in deposition. One consequence of our incomplete understanding of dust processes is that observational constraints will remain valuable even as model resolution is increased.

A second limitation of our methodology is that the quality of the inverse model depends on the accuracy of the observational constraints on the globally averaged dust size distribution (Adebiyi and Kok, 2020), extinction efficiency (Kok et al., 2017), and the regional DAOD constraints obtained in Ridley et al. (2016) and Adebiyi et al. (2020). As such, the results presented here are subject to the limitations of those studies. These limitations are described in detail in the corresponding papers and include possible biases due to errors in the dust extinction

efficiency due to the assumed tri-axial ellipsoid shape being an imperfect approximation to the highly heterogeneous shape and roughness of real dust particles (Lindqvist et al., 2014; Kok et al., 2017), errors in the remotely sensed optical depth retrieval algorithms for aspherical dust particles (Hsu et al., 2004; Kalashnikova et al., 2005; Dubovik et al., 2006), errors in the cloud-screening algorithms used in satellite and ground-based remote sensing products, errors due to a scarcity of AERONET "ground-truth" data in dust-dominated regions, and systematic differences

between clear-sky and all-sky AOD, although studies indicate such a systematic difference is small for dusty regions (Kim et al., 2014; Ridley et al., 2016; Adebiyi and Kok, 2020). The uncertainty due to many (not all) of these errors have been quantified in the relevant papers, and these errors have thus been propagated into the results in the present study. An additional key limitation is that the Ridley et al. (2016) DAOD constraint uses model simulations of the AOD due to other aerosol species to separate dust AOD from non-dust AOD in dusty regions. As such, consistent

biases in model simulations of non-dust AOD would have affected the inferred dust AOD. For instance, a systematic underestimation of biomass burning AOD across models (Reddington et al., 2016; van der Werf et al., 2017) would cause the underestimated biomass burning AOD to instead be assigned to dust, thereby causing an overestimate of dust AOD. This source of error might be particularly important in regions with substantial non-dust aerosol loadings, such as in much of Asia and in the Sahel during the biomass burning season (Yu et al., 2019).

Furthermore, the regional DAOD constraints from Adebiyi et al. (2020) for the lesser source regions of Australia, North America, South America, and South Africa are based on an ensemble of aerosol reanalysis products. These products assimilate remotely sensed AOD and partly rely on prognostic aerosol models to partition this AOD to the different aerosol species (e.g., Randles et al., 2017). Considering the large uncertainties in dust models (Huneeus et al., 2011; Checa-Garcia et al., 2020; Wu et al., 2020), these products could thus be substantially biased in regions for

which dust does not dominate AOD.

Another limitation of our results is that the representation of the modern-day global dust cycle is based mostly on model data and regional DAOD constraints for the period 2004-2008. As such, changes in the dust cycle before or after that period are not reflected in our results. For instance, satellite measurements have shown an increase in dust loading in the Middle East (Hsu et al., 2012; Kumar et al., 2019). Further, we assume that dust contributes to loading

and deposition in the same season that it is emitted, which is not always true and could generate small inconsistencies. We also use observational constraints on DAOD only at the mid-visible (550 nm), which is most sensitive to dust with a diameter between ~1-5 μm (Fig. 5f). Dust particles outside of this size range are thus partially constrained by correcting model simulations to match the globally averaged dust size distribution inferred in Adebiyi and Kok (2020) and might thus have larger errors that dust with diameters around 1-5 μm. Another

important limitation is that many of the models in our ensemble do not explicitly account for anthropogenic (e.g., land use) sources of dust emission, which might account for ~10-25% of present-climate dust emissions (Mahowald et al., 2004; Tegen et al., 2004; Ginoux et al., 2012). However, the observationally constrained DAOD used here to scale dust emissions and loading does not distinguish between natural and anthropogenic dust and thus inherently includes both. Nonetheless, the omission of land use dust emissions from many of the models in our ensemble could

produce important errors close to anthropogenic dust sources, which might account for a substantial fraction of total emissions in Asia, Australia, Southern Africa, and the Americas (Ginoux et al., 2012). Another limitation is that our approach neglects feedback between dust and meteorology, which could be important for certain regions or seasons (Cakmur et al., 2004; Miller et al., 2004; Perez et al., 2006; Ahn et al., 2007; Heinold et al., 2007; Colarco et al., 2014; Randles et al., 2017).

Finally, the conclusion that our methodology yields an improved representation of the global dust cycle depends on the quality of the independent data used to evaluate the inverse model results. However, these data have a few limitations. First, some of the measurements might have large, unquantified errors. This appears to be the case especially for deposition flux measurements, which show a much larger spread than surface concentration measurements, even for proximal locations. Second, the concentration and deposition data used to evaluate the

inverse model results do not coincide in time with the simulations, which could affect the comparisons (see Sect. 3 and further discussions in, e.g., Huneeus et al., 2011). Finally, some aspects of our representation of the global dust cycle were not explicitly tested against measurements. Future work could further investigate the accuracy of the inverse model results through comparisons against additional data, such as visibility data (Mahowald et al., 2007; Shao et al., 2013), dust vertical profile data (Yu et al., 2010; Kim et al., 2014), and remote sensing retrievals of the

Angstrom exponent (Huneeus et al., 2011). In addition to these limitations with the data, it is also possible that the inverse model better reproduces independent measurements because of canceling errors, for instance between model underestimates in long-range transport of coarse dust and overestimates in emissions from source regions closer to observational sites.

## 5.2.    Improving model representations of the global dust cycle

The results in Figs. 6-11 show that our methodology of integrating observational constraints on dust properties and abundance reduces model errors in simulating the global dust cycle. This finding is particularly clear from the results of the six "improved models". Each of these models shows a substantial reduction of model error against measurements and observations of the NH dust cycle (Figs. 7d-f, 8a-d), with the average reduction of the errors in "improved models" equaling ~35% (Fig. 8e). These findings suggest several ways in which the representation of the

global dust cycle can be improved in global and regional models.

First, our results indicate that it is critical for models to account for the substantial asphericity of dust aerosols (Okada et al., 2001; Huang et al., 2020). Dust asphericity enhances the MEE by ~40% because aspherical dust particles extinguish more radiation than volume-equivalent spherical particles (Kalashnikova and Sokolik, 2004; Potenza et al., 2016; Kok et al., 2017). As such, not accounting for dust asphericity causes an overestimation of the

dust loading needed to match DAOD constraints by ~40%, and thus can produce a corresponding bias against concentration and deposition flux measurements. This is illustrated by the MERRA-2 results, which are in good

agreement with DAOD constraints (Figs. 7d, 7e, 8e), but overestimate NH dust deposition flux measurements by ~25% and surface concentration measurements by ~50% (Figs. 8a, b and Figs. S9 and S10). MERRA-2 uses dust optics from Colarco et al. (2014) based on spheroids, which underestimate dust asphericity (Huang et al., 2020) and yielded a ~25% enhancement of dust extinction. Accounting for the full extinction enhancement of ~40% due to dust asphericity would thus reduce the biases of the MERRA-2 dust product against surface concentration and deposition flux measurements. Since most current models either do not account for dust asphericity or substantially underestimate its effect on extinction efficiency (Huang et al., 2020) we recommend that models account for the full enhancement of extinction by dust asphericity, for instance by implementing the constraints on the extinction efficiency of aspherical dust from Kok et al. (2017).

Second, models can be improved by correcting the current substantial underestimation of coarse dust loading. In this study, we integrated a joint observational-modeling constraint on the globally averaged dust size distribution in order to account for the ~17 Tg of coarse dust ($5 \leq D \leq 20$ μm) that observations indicate is present in the atmosphere (Ryder et al., 2019; Adebiyi and Kok, 2020). The finding that our methodology almost eliminates biases against NH measurements (Figs. 6-8) suggests that this constraint on the globally averaged dust size distribution is relatively accurate. This further supports the conclusion from several studies that models substantially underestimate coarse dust loading (Ansmann et al., 2017; van der Does et al., 2018; Ryder et al., 2019; Adebiyi and Kok, 2020; Gliss et al., 2021). Models can thus be improved by eliminating the current underestimation of coarse dust. This could be done either by adjusting the size distribution of emitted dust aerosols such that the size-resolved global dust loading matches the constraints on the globally averaged size distribution (Adebiyi and Kok, 2020) or, preferably, by improving the relevant model physics. Specifically, recent studies indicate that the underestimation of coarse dust is due to both an underestimation of the emission of coarse dust (Huang et al., 2021) and an underestimation of the lifetime of the emitted coarse dust (Maring et al., 2003; Weinzierl et al., 2017). Measurements of the emitted dust size distribution show a much larger flux of dust with $D \geq 10$ μm than current parameterizations, including brittle fragmentation theory (Kok, 2011b, a), account for (Huang et al., 2021). The fact that models need to use a fractional contribution of emitted super coarse dust ($10 \leq D \leq 20$ μm) that is even larger than found by measurements (Fig. 5a; Sow et al., 2009; Rosenberg et al., 2014; Huang et al., 2019; Huang et al., 2021) suggests that models underestimate the lifetime of (super) coarse dust. This finding further supports the inference from several lines of evidence that models underestimate the lifetime of (super) coarse dust (Maring et al., 2003; Weinzierl et al., 2017; van der Does et al., 2018). As such, models require improved parameterizations of both the emitted dust size distribution and dry deposition processes to properly account for the abundance of (super) coarse dust in our atmosphere. Improved parameterizations of the emitted dust size distribution that better account for the large contribution of (super) coarse dust are under development (Huang et al., 2021). To improve size-resolved dry deposition, we recommend that models account for the ~20% slowing of the gravitational settling speed due to dust asphericity (Huang et al., 2020). Further improvements in dust deposition parameterizations are likely needed, including accounting for the strong enhancement of upward vertical transport of emitted (super) coarse dust by topography (Rosenberg et al., 2014; Heisel et al., in review) and possible reductions of gravitational settling due to electrification and turbulence in dust layers (Ulanowski et al., 2007; Gasteiger et al., 2017; van der Does et al., 2018).

Finally, our results indicate that model accuracy can be substantially improved by correcting biases in the dust loading generated by each main source region (Figs. 3, 8e). These biases could be reduced in two ways. First, models could emulate the procedure developed here and scale emission of dust from each region to match the observed regional DAOD obtained in Ridley et al. (2016). A second approach would be to scale the simulated (size-resolved) emissions or loading per source region and season to that obtained in our companion paper (Kok et al., 2021). These improvements would be most effective for simulations of the present-day dust cycle by regional and global models, as well as short range, medium range and seasonal forecasts of dustiness by numerical weather models. Ultimately, parameterizations of dust emission should be improved to eliminate the need for adjustment of model simulations in this manner. This is critical because without identifying and correcting the problematic model physics, we cannot know how these processes change with climate, for example under global warming or over glacial cycles. Together with uncertainties due to future land use changes, this problem limits the ability of models to predict future changes in the global dust cycle and its effect on climate and the Earth system (Evan et al., 2016; Kok et al., 2018).

Although we found that the integration of observational constraints on dust properties and abundance is effective in reducing model errors in the representation of the NH dust cycle, we found only slight improvements for the SH dust cycle (Fig. 11e). There are two likely reasons for this finding. First, whereas the inverse model is informed by accurate observational constraints on regional DAOD in the NH, such constraints are less accurate for the less dusty

SH (Ridley et al., 2016). And second, the dust cycle simulations used in our ensemble are less accurate for the SH dust cycle than for the NH dust cycle, as indicated by substantially larger root mean-squared errors relative to measurements for the SH (Figs. 11c, d) than for the NH (Figs. 8c, d). These larger model errors for the SH likely occur because a large fraction of SH dust emission originates from regions containing sparse vegetation (Ito and Kok, 2017), the effects of which on dust emission is difficult for models to represent accurately (King et al., 2005; Okin, 2008). Additionally, there are less data available in the SH from ground-based measurements such as dust surface concentration measurements. And whereas many measurements close to dust source regions are available for the NH, most measurements for the SH are at sites remote from the main dust source regions (Figs. 2c, d), where they are less effective at constraining the main features of the SH dust cycle. There are also fewer satellite retrievals available to constrain simulations of the SH dust cycle. For instance, dust sources such as Patagonia are shrouded by clouds for a larger fraction of the year than for most NH sources (Ginoux et al., 2012), which limits constraints on dust emissions and DAOD from satellite retrievals (Gasso and Stein, 2007). Additionally, the errors in satellite retrievals tend to be larger for the SH than for the NH because the relative error decreases with AOD (Kahn et al., 2005; Remer et al., 2005). Considering the important role that the SH dust cycle plays in biogeochemistry, the carbon cycle, and the climate system (Lambert et al., 2008; Hamilton et al., 2020), our results underscore a critical need for more observations to constrain the SH dust cycle.

### 5.3. Utility of DustCOMM data set in understanding the role of dust in the Earth system

In addition to identifying mechanisms to improve individual model simulations, this study obtained an improved representation of the global dust cycle that can be used to improve our understanding and quantification of the impact of dust on the Earth system. This addition to the DustCOMM data set (Adebiyi et al., 2020) contains dust loading, DAOD, (surface) concentration, and (wet and dry) deposition flux fields that are resolved by space, particle size, and season (data are available at https://dustcomm.atmos.ucla.edu/data/K21a/). Our results in Sect. 4.3 indicate that this data set is more accurate than both a large number of climate and chemical transport model simulations and the MERRA-2 dust product. Moreover, whereas MERRA-2 is internally inconsistent because dust loading is adjusted after emission by assimilating AOD measurements (Randles et al., 2017; Wu et al., 2020), our method for integrating observational constraints yields a self-consistent representation of the global dust cycle. Our companion article (Kok et al., 2021) will supplement this data set by partitioning all these fields by the originating source region. This data set representing the seasonally resolved and size-resolved global dust cycle can be used to more accurately quantify dust impacts on the Earth system, such as on climate, weather, the hydrological cycle, biogeochemistry, and human health.

Our data set of an improved representation of the global dust cycle has an additional strength that amplifies its use: our data set quantifies and propagates a range of observational and modeling uncertainties (see Sect. 2.5). Comparisons against independent data sets indicate that the propagated error is realistic for the NH and might slightly underestimate the true errors in the SH (Figs. 7 and 10). The availability of realistic errors allows for the propagation of uncertainty into dust impacts constrained using our data set, such as in the quantification of direct radiative effects and indirect cloud and biogeochemistry effects (Mahowald, 2011). With a few exceptions (Kok et al., 2017; Regayre et al., 2018; Di Biagio et al., 2020), the quantification of the uncertainty of (dust) aerosol direct and indirect radiative effects is uncommon, yet is critical to robustly constraining (dust) aerosol impacts on the Earth system (Carslaw et al., 2010; Mahowald et al., 2011b). Moreover, the quantification of uncertainties on aerosol effects in both the present-day and pre-industrial climates is crucial to constraining climate sensitivity (Carslaw et al., 2013; Carslaw et al., 2018).

A second strength of our data set representing the global dust cycle is that it uses an analytical framework that could be improved and expanded. The framework could be improved by using more accurate observational constraints of dust properties and dust abundance as inputs (see Fig. 1), for instance from several recent DAOD climatologies (Pu and Ginoux, 2018; Voss and Evan, 2020; Gkikas et al., 2021), or by adding additional types of observational constraints, such as on the dust vertical profile (Song et al., 2021). The framework could be expanded by adding calculations of additional dust properties and impacts, such as dust mineralogy and radiative effects. The framework could also be expanded to cover different time periods than the 2004-2008 time period we used here, or to constrain the historical variability of the global dust cycle, for instance using time-resolved DAOD climatologies (Voss and Evan, 2020; Gkikas et al., 2021; Song et al., 2021). As such, our approach has potential to continually improve the representation of the global dust cycle and its impacts on the Earth system.

## 6. Conclusions

We have obtained an improved representation of the global dust cycle by developing an analytical framework that uses inverse modeling to integrate observational constraints on the dust size distribution, extinction efficiency, and regional DAOD with an ensemble of global dust cycle simulations (Fig. 1). This new approach mitigates two critical challenges that models face in representing the global dust cycle, namely (i) that capturing the magnitude and spatial distribution of dust emissions is a fundamental challenge for large-scale models because of the large mismatch between the resolved scales (~10-100 km) and the physically relevant scales (~1 m to several km) over which dust emissions vary, and (ii) that models have difficulty representing uncertainties in dust microphysical properties and often use values that are not consistent with up-to-date observational and experimental constraints.

Comparisons against independent measurements indicate that this new framework of integrating observational constraints with model simulations produces an improved representation of the present-day (2004-2008) global dust cycle. Our inverse model reproduces NH measurements of dust surface concentration well within the experimental and modeling uncertainties and with a factor of 1.5-5 less error than both individual model simulations and the MERRA-2 dust product (Figs. 8c, d). This large improvement is due to reduced errors in capturing the seasonal cycle (Fig. 6) and the spatial variability of dust surface concentration (Figs. 7d, e), and because of the near elimination of biases against measurements in the NH (Figs. 8a, b). Overall, the inverse model results show a reduction of errors against measurements and observations of the NH dust cycle measurements of approximately a factor of two (Fig. 8e). These improvement are noteworthy as previous studies have had difficulty simultaneously reproducing dust AOD, surface concentration, and deposition flux (Cakmur et al., 2006; Mahowald et al., 2006; Albani et al., 2014).

The elimination of bias against independent data suggests several ways in which dust models can be improved. First, models should account for the enhancement of the MEE by dust asphericity (Kalashnikova and Sokolik, 2004; Kok et al., 2017). Otherwise, a ~40% greater dust loading would be needed to match DAOD constraints, resulting in a corresponding overestimation of NH dust surface concentration and deposition fluxes. Our results further indicate that models can be improved by correcting the current underestimation of coarse dust loading (Adebiyi and Kok, 2020) and by adjusting source-resolved emissions to match regional DAOD constraints (Ridley et al., 2016).

Although the integration of observational constraints thus improves the representation of the NH dust cycle, we found less improvement in the SH dust cycle. This is likely due to both the lower quality of constraints on regional DAOD in the SH and because of the difficulty models have in reproducing the dust cycle in the less dusty SH.

We also find that the emission flux of dust with geometric diameter up to 20 μm ($PM_{20}$) is approximately 5,000 Tg/year (one standard error range of 3400 to 8900 Tg/year; Table 3), which is greater than most models account for. This greater global emission rate is partially driven by a larger emission flux of (super) coarse dust with $D \geq 5$ μm, which we find accounts for ~80% of the global $PM_{20}$ emission flux (Fig. 5a). This large flux of coarse dust is needed to generate the ~17 Tg of atmospheric (super) coarse dust loading that *in situ* measurements indicate resides in the atmosphere (Adebiyi and Kok, 2020). Accounting for this substantial loading of coarse dust is important because these particles account for a substantial fraction of absorption of shortwave radiation and both absorption and scattering of longwave radiation (Tegen and Lacis, 1996; Ryder et al., 2018; Ryder et al., 2019; Fig. 5), and also can account for a large fraction of nutrients delivered to ecosystems by dust.

The improved representation of the global dust cycle presented here is publicly available as part of the DustCOMM data set (Adebiyi and Kok, 2020; Adebiyi et al., 2020). These data include gridded dust emission, loading, (surface) concentration, wet and dry deposition, and DAOD fields that are resolved by season and particle size, including by particle bin and for $PM_{2.5}$, $PM_{10}$, and $PM_{20}$ dust. Additional strengths of this data set are that it includes uncertainty estimates and that the data can be readily updated as improved constraints on dust properties and abundance become available. As such, our improved representation of the global dust cycle can facilitate more accurate constraints on the various critical impacts of dust on the Earth system.

## Glossary

$\alpha_k$         Dimensionless global scaling factor by which a unit dust loading in a global model simulation's particle size bin $k$ is multiplied in order to bring the annually averaged global

| | |
|---|---|
| | dust loading generated from all source regions in agreement with the constraint on the globally averaged dust size distribution ($\frac{d\bar{V}_{\text{atm}}(D)}{dD}$). |
| $\bar{\epsilon}_k$ | Mass extinction efficiency (m²/kg) of a global model simulation's particle size bin $k$, obtained by integrating the constraint on the globally averaged extinction efficiency $\bar{Q}_{\text{ext}}(D)$ over the particle bin's size range. |
| $\theta$ | Longitude. |
| $\bar{\rho}_d$ | Density of dust aerosols, which is taken as $(2.5\pm0.2)\times10^3$ kg m$^{-3}$. |
| $\bar{\sigma}_s^p$ | Total uncertainty on the area-averaged observed DAOD of region $p$ for season $s$. |
| $\bar{\tau}_s^p$ | Area-averaged observed DAOD for region $p$ and season $s$. |
| $\check{\tau}_{r,s}(\theta,\phi)$ | Inverse model seasonally averaged DAOD produced by dust emitted from source region $r$, averaged over season $s$. |
| $\check{\tau}_{r,s}^p$ | Inverse model seasonally averaged DAOD produced by dust emitted from source region $r$, averaged over season $s$ and observed region $p$. |
| $\phi$ | Latitude. |
| $\chi_\tau^2$ | Summed squared deviation between the observed DAOD in the fifteen regions and that obtained from our analysis. |
| $A_p$ | Area of the region $p$ defined in Table 2 (m²). |
| $c_{\text{PM10}}$ | Global constant denoting the fractional contribution to the PM$_{10}$ deposition flux of a model particle size bin that straddles 10 μm. |
| $\tilde{C}_{r,s,k}(\theta,\phi,P)$ | Model-simulated 3D dust concentration (m$^{-3}$) produced by a unit of dust loading (1 Tg) in particle size bin $k$ emitted from source region $r$, averaged over season $s$. |
| $\check{C}_s(\theta,\phi,P)$ | Inverse model 3D bulk dust concentration (kg m$^{-3}$), averaged over season $s$. |
| $\check{C}_{s,k}(\theta,\phi,P)$ | Inverse model 3D dust concentration (kg m$^{-3}$) produced by dust in particle size bin $k$, averaged over season $s$. |
| $D$ | Geometric (or volume-equivalent) diameter (m). |
| $D_{k-}$ | Lower geometric diameter limit of a global model simulation's particle size bin $k$ (m). |
| $D_{k+}$ | Upper geometric diameter limit of a global model simulation's particle size bin $k$ (m). |
| $D_{\text{max}}$ | Maximum dust aerosol geometric diameter considered in this study, namely $D_{\text{max}} = 20$ μm. |
| $\tilde{D}_{r,s,k}(\theta,\phi)$ | Model-simulated spatial distribution of dust deposition flux (m$^{-2}$ yr$^{-1}$) produced by a unit of dust loading (1 Tg) in particle size bin $k$ emitted from source region $r$, averaged over season $s$. |
| $\check{D}_{s,k}(\theta,\phi)$ | Inverse model spatial distribution of deposition flux (kg m$^{-2}$ yr$^{-1}$) of dust in particle bin $k$, averaged over season $s$. |
| $\check{D}_s(\theta,\phi)$ | Inverse model spatial distribution of bulk dust deposition flux (kg m$^{-2}$ yr$^{-1}$), averaged over season $s$. |
| $\tilde{f}_{r,s,k}$ | Model-simulated seasonally averaged fraction of global dust loading emitted from source region $r$ that is contained in particle size bin $k$. |
| $\check{f}_{r,s,k}$ | Inverse model fraction of seasonally averaged global dust loading emitted from source region $r$ that is contained in particle size bin $k$. |
| $\tilde{F}_{r,s,k}(\theta,\phi)$ | Model-simulated spatial distribution of dust emission flux (m$^{-2}$ yr$^{-1}$) needed to generate a unit (1 Tg) of dust loading in particle size bin $k$ emitted from source region $r$, and averaged over season $s$. |
| $\check{F}_{s,k}(\theta,\phi)$ | Inverse model spatial distribution of dust emission flux (kg m$^{-2}$ yr$^{-1}$) of dust in particle size bin $k$, averaged over season $s$. |
| $\check{F}_s(\theta,\phi)$ | Inverse model spatial distribution of bulk dust deposition flux (kg m$^{-2}$ yr$^{-1}$), averaged over season $s$. |
| $J_{r,s}(\theta,\phi)$ | Spatial distribution of the Jacobian matrix (Tg$^{-1}$) of $\check{\tau}_{r,s}$ with respect to $\check{L}_{r,s}$, which equals the DAOD produced per unit of bulk dust loading from source region $r$, averaged over season $s$. |
| $J_{r,s}^p$ | The Jacobian matrix of $\bar{\tau}_s^p$ with respect to $\check{L}_{r,s}$ (Tg$^{-1}$), which equals the seasonally averaged DAOD produced per unit of dust loading originating from source region $r$ in season $s$ and averaged over the observed region $p$. |
| k | Index that sums over the different particle size bins of a given global model. |

| | |
|---|---|
| $\tilde{l}_{r,s,k}(\theta,\phi)$ | Model-simulated spatial distribution of dust column loading produced by a unit of dust loading (1 Tg) in particle size bin $k$, emitted from source region $r$, averaged over season $s$ ($m^{-2}$). |
| $\check{l}_s(\theta,\phi)$ | Inverse model spatial distribution of dust bulk column loading, averaged over season $s$ (kg $m^{-2}$). |
| $\check{l}_{s,k}(\theta,\phi)$ | Inverse model spatial distribution of dust column loading produced by dust in particle size bin $k$, averaged over season $s$ (kg $m^{-2}$). |
| $\check{L}_r$ | Inverse model annually averaged global dust loading produced by source region $r$ (Tg). |
| $\check{L}_{r,s}$ | Inverse model global dust loading produced by source region $r$, averaged over season $s$ (Tg). |
| $N_{bins}$ | Number of dust particle size bins in a given global model simulation. |
| $N_{\tau,reg}$ | Number of regions with observationally constrained DAOD; $N_{\tau,reg} = 15$. |
| $N_{sreg}$ | Number of source regions; $N_{sreg} = 9$. |
| p | Index that sums over the fifteen regions with observationally constrained DAOD. |
| P | Vertical pressure level. |
| $\bar{Q}_{ext}(D)$ | Globally averaged size-resolved extinction efficiency (dimensionless) from Kok et al. (2017), which is defined as the extinction cross-section divided by the projected area of a sphere with diameter $D$ ($\pi D^2/4$). |
| r | Index that sums over the $N_{sreg} = 9$ source regions. |
| s | Index that sums over the four seasons. |
| $\dfrac{d\bar{V}_{atm}(D)}{dD}$ | The size-normalized (that is, $\int_0^{D\max} \frac{d\bar{V}_{atm}}{dD} dD = 1$) globally averaged volume size distribution of atmospheric dust from Adebiyi and Kok (2020). |

## Data availability

The data obtained in this paper are available at https://dustcomm.atmos.ucla.edu/data/K21a/. These data include gridded dust emission, loading, (surface) concentration, wet and dry deposition, and DAOD fields that are resolved by season and particle size, including by particle bin and for $PM_{2.5}$, $PM_{10}$, and $PM_{20}$ dust. All fields include one and two standard error uncertainty estimates.

## Author contributions

JFK designed the study, analyzed model data and wrote the manuscript. DSH, LL, NMM, and JSW performed CESM/CAM4 simulations; AI performed IMPACT simulations; RLM performed GISS ModelE2.1 simulations; PRC and ARL performed GEOS/GOCART simulations; MK, VO, and CPGP performed MONARCH simulations; and SA, YB and RCG performed INCA simulations. CAW and AAA analyzed dust surface concentrations. YH analyzed results from AeroCom Phase 1 models and MERRA-2. AAA provided observational DAOD constraints.
DML and MC provided valuable comments on study design. All authors edited and commented on the manuscript.

## Competing interests

The authors declare that they have no conflict of interest.

## Acknowledgements

This work was developed with support from the National Science Foundation (NSF) grants 1552519 and 1856389
and the Army Research Office under Cooperative Agreement Number W911NF-20-2-0150 awarded to J.F.K, from the University of California President's Postdoctoral Fellowship awarded to A.A.A., from the European Union's Horizon 2020 research and innovation programme under the Marie Skłodowska-Curie grant agreement No 708119 awarded to S.A. and No. 789630 awarded to M.K. R. C.-G. received funding from the European Union's Horizon 2020 research and innovation grant 641816 (CRESCENDO), and from JSPS KAKENHI Grant Number 20H04329

and Integrated Research Program for Advancing Climate Models (TOUGOU) Grant Number JPMXD0717935715 from the Ministry of Education, Culture, Sports, Science and Technology (MEXT), Japan to A.I.  P.R.C. and A.R.-L. acknowledge support from the NASA Atmospheric Composition: Modeling and Analysis Program (R. Eckman, program manager) and the NASA Center for Climate Simulation (NCCS) for computational resources, Y.H. acknowledges NASA grant 80NSSC19K1346, awarded under the Future Investigators in NASA Earth and Space
Science and Technology (FINESST) program, and R.L.M. acknowledges support from the NASA Modeling, Analysis and Prediction Program (NNG14HH42I). S.A. acknowledges funding from MIUR (Progetto Dipartimenti di Eccellenza 2018-2022). C.P.G.P. acknowledges support by the European Research Council (grant no. 773051, FRAGMENT), the EU H2020 project FORCES (grant no. 821205), the AXA Research Fund, and the Spanish Ministry of Science, Innovation and Universities (RYC-2015-18690 and CGL2017-88911-R). M.K. and C.P.G.P.
acknowledge PRACE for awarding access to MareNostrum at Barcelona Supercomputing Center to run MONARCH. L.L. acknowledges support from the NASA EMIT project and the Earth Venture – Instrument program (grant no. E678605). Y.B. and R.C.-G. benefited in this study from funding by the PolEASIA ANR project under the allocation ANR-15-CE04-0005. We also acknowledge high-performance computing support from Cheyenne (doi:10.5065/D6RX99HX) provided by NCAR's Computational and Information Systems Laboratory,
sponsored by the National Science Foundation. We further thank Anna Benedictow for assistance in accessing the AeroCom modeling data, the AeroCom modeling groups for making their simulations available, Joseph Prospero and Nicolas Huneeus for providing dust surface concentration data from in situ measurements from the University of Miami Ocean Aerosol Network, and the investigators of the Sahelian Dust Transect for making their dust concentration measurements available. The MERRA-2 data used in this study/project have been provided by the
Global Modeling and Assimilation Office (GMAO) at NASA Goddard Space Flight Center. The views and conclusions contained in this document are those of the authors and should not be interpreted as representing the official policies, either expressed or implied, of the Army Research Laboratory or the U.S. Government.

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
