# Peer review of "Improved representation of the global dust cycle using observational constraints on dust properties and abundance"

_Atmospheric Chemistry and Physics, 2020_

## Referee Comment (RC1) · Anonymous Referee #2 · 8 Dec 2020

The authors present a modelling approach based on an inversion model that includes a multi-model ensemble. With their approach the authors aim at improving the representation of the global dust cycle in global climate models.

The manuscript is well written; however, it is quite lengthy. I appreciate that all aspects are explained in detail, nevertheless, I think the manuscript would benefit from overall shortening.

Not all references are listed in the section References. Please complete.

General remarks I wish the authors to comment on:

[Figure]

(1) In order to apply the inversion approach, the authors separate the world (excluding high-latitudes) into 9 regions, across which the dust load is averaged. Although I understand that following such a procedure reduced the computational cost, I am wondering up to which degree this approach fosters obtaining the expected results (with regard to the observations) for the wrong reason. In particular as some atmospheric processes controlling dust emission and dispersion may cancel each other out.

(2) Can you elaborate in more depth in which way you see the general applicability and future potential of the presented modelling approach? I am wondering how feasible it is for a use in practise. The here presented study involves a 5-year period from more than 10 years ago (2004-2008). Not every model ensemble involved provides all necessary variables over the entire time period. Furthermore, 6 different model simulations are involved, not all of them freely accessible.

(3) Following on remark (2), how consistent is study if for some variables all model ensembles are used and for others not?

(4) Aren't the individual ensemble models as well as the MERRA2 data set somehow "tuned" towards similar observation data sets? In other words, how independent is your "independent" data set? This general comment refers in particular to section 4.1.

(5) Unfortunately, the methodology and results as presented in current form, seem to not contribute to a overall improvement of dust production models or their underlying conceptual understanding. Rather, the inversion model seems to reflect the consensus view across all models. What can the science community learn from this approach? I am sure there is something we can learn from this approach. Maybe this can be presented in a more prominent and obvious way?

(6) The physical consistency of dust emission flux calculations and dispersion with atmospheric conditions which then results into a dependent dust aerosol optical depth seems to be circumvented by this approach here. Can you comment on how physically consistent the overall representation of the dust cycle and its associated feedbacks is?

(7) Furthermore, as the individual model ensembles use different parameterisation schemes for various processes ultimately determining the dust cycle, I am wondering if this is an advantage here as diversity is reflected, or a disadvantage as it ignores physical consistency for the multi-model average. Please clarify.

(8) Where do you see the benefit of applying an inverse model over satellite data assimilation or a common ensemble mean?

Specific comments:

Line 26: Is this an "improved representation" of the global dust cycle or an averaged representation as it reflects the consensus view across all model ensembles? Please clarify.

Line 52-71: Across this paragraph, the need for an as accurate as possible representation of processes driving and controlling the dust cycle is illustrated. However, isn't this, the relevance of process-driven parameterisation for improving the representation of the dust cycle, ignored by the approach suggested here? In particular as averaging may result into cancelling out relevant processes.

Line 313-317: I am wondering if this is only applicable to coarse grid model simulations which would not be able to capture fine-scale dust plumes anyway. Please comment.

---

## Referee Comment (RC2) · Anonymous Referee #1 · 18 Jan 2021

**Review of Kok et al., Improved representation of the global dust cycle using observational constraints on dust properties and abundance, 2020, ACPD**

**Overview**

This article is well presented and well written, and fits well within the scope of ACP. The authors present a new methodology to constrain and improve dust models using an inverse modeling approach. This is evaluated compared to independent observations of surface dust concentration and dust deposition from around the globe. The authors show that their results deliver significant improvements in modeled parameters which have previously been extremely challenging for dust models. This work is a significant undertaking, and delivers important results to the field of global aerosol modeling. My comments are only minor clarifications and I recommend publication after they have been answered.

**General Comments**

The article is complex and somewhat lengthy – however the authors strive to carefully explain their methodology, which is a crucial part of the paper, and the description of the method is clear and appropriate. Therefore I advise retaining the current amount of detail and length (and not shortening the article, as suggested by reviewer 1).

**Specific Comments**

L57-58 – It is not just wind speed which is important to emission, but also the infrequent but strong wind occurrence which can contribute disproportionately strongly towards dust emission due to the non-linear scaling mentioned. Cowie et al. (2015) and Roberts et al. (2017) are relevant papers towards understanding and representing dust-generating winds in observations in models, with a focus on high-wind speed occurrence, which should be mentioned.

L86-87 – sentence is unclear.

Introduction – readers may find the scope of this article fairly similar to several previous articles by the lead author and research group (e.g. Adebiyi et al. 2020, Adebiyi & Kok 2020. Therefore, it would be helpful to guide the reader briefly through the steps in which this piece of work differs from others, either in the introduction or wherever the authors consider logical. Improvements here would contribute a large amount to increasing the significance of this paper.

L208-212 – it is not clear how the authors use the GEOS model to extend the size range for CESM and ModelE2 – please give more details (and also relate to comments for the supplement).

Section 2.3, p8 – take care when writing 'this study' and referring to Ridley et al. (2016) – it is not always clear that the authors mean Ridley et al. by 'this', as opposed to referring to the current article under review. 'That study' or a repeat of 'Ridley et al' would be clearer.

Section 2.3, 2nd & 3rd paragraphs – given that the reanalysis datasets also include aerosol data assimilation, why not just use them for DAOD rather than the Ridley dataset? What is the advantage of the Ridley dataset over the reanalysis datasets in general?

L412-413 – it would be useful to include the description of this dataset, the 'Saharan Dust Transect.'

L424-425 – this reasoning is unclear. "because most of the dust at this site originated locally from within and near the national park where the station was located" – does 'this site' refer to the Zimbabwe or Australian site?

Section 4.1 – Figs 3e and f are barely mentioned in the text. What does fig 3e tell us (seasonal DAOD) that fig 3f (annual) does not? What do their differences indicate?

L572 – O'Sullivan et al., 2020, is another recent publication finding that also found that coarse dust is deposited too quickly in models, and could be referenced here. Also L952.

L572-575 – what does a similar plot of either MEE or DAOD for dust longwave properties look like? (in terms of the panels shown in fig 5). Given that the authors point out that the longwave radiative effects of dust are also important, it would be useful to provide some information from this study regarding the LW properties.

Table 3 – include DAOD and MEE wavelength in caption or table header.

Section 4.3.1 – regarding figure 6 – it might be expected that if one took a multi-model mean the result might agree fairly well with the observations as well. Since the inverse model incorporates input from several models, can the authors explain and confirm why the inverse model can be taken as agreeing better with the observations that a multi-model mean might? I suppose this can be inferred from the size of the error bars compared to the range individual models, though it would be useful to include a discussion of this.

L738-9 – But isn't MERRA used for DAOD in the SH, rather than the Ridley dataset?

L953-954 – "This could be done either by similarly applying the constraints on the globally averaged size distribution (Adebiyi and Kok, 2020)" – can the authors be more specific about how this could be carried out in an online (climate) model?

Section 5.3 – various other datasets of global Dust AOD are now available, which may have similarities and/or differences to Ridley et al. (2016). For example, Pu and Ginoux (2018) based on MODIS data, datasets based on CALIOP satellite AOD where dust AOD is derived based on shape, and combinations of MODIS/MERRA data (e.g. MIDAS, Gkikas et al., 2021). A discussion of the possible implications of applying a different DAOD dataset might result in would be beneficial.

**Figures**

Fig 3 – caption "same as panel b" – should be panel e?

Figure 5 – what is meant by 'number load' – should this be number concentration?

Figure 6 – it is impossible to distinguish between the black and brown lines, and also very difficult to make out the letters identifying different models.

Fig 7 – what is meant by 'seasonal surface concentration' when given as one number? Same applies to fig 8, 10 and 11.

Fig 7 and fig 8 – could the mean of Aerocom models also be shown on the figures, since means of improved models and mean of ensemble models are shown, for clarity?

**Supplement**

CESM/CAM4 – A description of how the additional largest size bin was generated is missing.

L90-91 "Simulations for each of the nine source regions were obtained by tagging each particle size bin from the different source regions." – it is not clear what is meant by this statement – please expand.

L96-97 – "We did not use the largest bin because it exceeds the 20 μm maximum diameter used in the inverse model." – this statement appears to contradict the caption for table 1, main article, which indicates that the maximum size bin has been extended for this model.

L215 – "mm" – should be microns?

**References**

Cowie et al., The importance of rare, high-wind events for dust uplift in northern Africa, GRL, 2015, https://agupubs.onlinelibrary.wiley.com/doi/full/10.1002/2015GL065819

Gkikas, A., Proestakis, E., Amiridis, V., Kazadzis, S., Di Tomaso, E., Tsekeri, A., Marinou, E., Hatzianastassiou, N., and Pérez García-Pando, C.: ModIs Dust AeroSol (MIDAS): a global fine-resolution dust optical depth data set, Atmos. Meas. Tech., 14, 309–334, https://doi.org/10.5194/amt-14-309-2021, 2021.

O'Sullivan et al., Models transport Saharan dust too low in the atmosphere: a comparison of the MetUM and CAMS forecasts with observations, Atmos. Chem. Phys., 20, 12955–12982, https://doi.org/10.5194/acp-20-12955-2020, 2020.

Roberts et al., New Saharan wind observations reveal substantial biases in analysed dust-generating winds, Atmospheric Science Letters, 2017, https://rmets.onlinelibrary.wiley.com/doi/full/10.1002/asl.765

---

## Author Comment (AC1)

**Responses to reviewers' comments**
Comments in black and responses in blue

We thank both reviewers for their constructive comments, which has helped us to improve the paper. Below we include a point-by-point response to the referee comments, and describe the corresponding changes we have made to the manuscript.

In addition to addressing the reviewer comments, we have also improved the clarity of the description of the treatment of uncertainties in the paper. Specifically, we have included error bars on the scatter-plots in Figures 7 and 10, and we calculated and reported the reduced chi squared statistic (which should be of order 1 for a model that reproduces the data within the uncertainties). We added two paragraphs on these topics at the end of Section 3.1.

We believe these and other changes have addressed the reviewer comments and look forward to possible further suggestions and comments from the referees and editor.

**Referee #2**

The authors present a modelling approach based on an inversion model that includes a multi-model ensemble. With their approach the authors aim at improving the representation of the global dust cycle in global climate models.

The manuscript is well written; however, it is quite lengthy. I appreciate that all aspects are explained in detail, nevertheless, I think the manuscript would benefit from overall shortening.

Thank you for your careful reading of the paper and your helpful and positive comments.

We agree that the article is lengthy. However, we do agree with referee #1 that this length is necessary to adequately explain the methodology. We have not been able to identify any sections of the paper that could be cut or substantially shortened without affecting the clarity and main conclusions of the paper.

Not all references are listed in the section References. Please complete.

Thank you for pointing this out. We've gone over the text carefully and indeed found a few missing references. Do let us know in case we missed any other ones.

General remarks I wish the authors to comment on:

(1) In order to apply the inversion approach, the authors separate the world (excluding high-latitudes) into 9 regions, across which the dust load is averaged. Although I understand that following such a procedure reduced the computational cost, I am wondering up to which degree this approach fosters obtaining the expected results (with regard to the observations) for the wrong reason. In particular as some atmospheric processes controlling dust emission and dispersion may cancel each other out.

This is a great point. We agree that it is possible some of the agreement against observations is for the wrong reason (e.g., a systematic underestimation of long-range transport of coarse dust might be compensated for by overestimates in emission from source regions closer to observational sites) and we've added a sentence pointing this out at the end of Section 5.1.

(2) Can you elaborate in more depth in which way you see the general applicability and future potential of the presented modelling approach? I am wondering how feasible it is for a use in practise. The here presented study involves a 5-year period from more than 10 years ago (2004-2008). Not every model ensemble involved provides all necessary variables over the entire time period. Furthermore, 6 different model simulations are involved, not all of them freely accessible.

This is a good question. The reviewer is correct that this is an involved methodology that is not easy to apply by others. However, it could for instance be used in the framework of a multi-model collaboration, such as AeroCom.

We see the usefulness of this work in terms of applicability and future potential as (1) obtaining a gridded data set of the global dust cycle that is (1) more accurate than other available products, (2) is resolved by source region, (3) includes realistic errors, and (4) can be updated as more accurate constraints become available that could be used in our framework.

To address this comment, we have expanded the discussion at the end of Section 5.3 on this method's future potential by integrating additional and more accurate observational constraints, and by expanding the framework to constrain additional dust properties like mineralogy.

(3) Following on remark (2), how consistent is study if for some variables all model ensembles are used and for others not?

The modeling procedure presented here is internally consistent because each model ensemble member provides all the used fields (normalized emission, deposition, concentration, and loading fields per particle bin; see yellow boxed in Fig. 1) for the given years. We used the multi-year mean of the variables for each season, which we know clarified by adding a note for the "simulation period" to Table 1 that "A multi-year mean for each season was used."

We understand your concern about the differences in the simulation years. This concern is mitigated by the fact that we only use the normalized fields, which depend only on the spatial pattern of emission/deposition/loading and not on absolute amounts, and thus are less affected by interannual variability. We conducted sensitivity simulations using different years of output from different simulations and found only small differences in the inversion estimates (< 10%). We added a sentence to section 2.1 stating this ("Sensitivity tests indicated that using different years from each simulation resulted in differences of less than 10% in the inverse model results").

(4) Aren't the individual ensemble models as well as the MERRA2 data set somehow "tuned" towards similar observation data sets? In other words, how independent is your "independent" data set? This general comment refers in particular to section 4.1.

This is a great question. Since we only use each model's normalized fields (so per 1 Tg loading and for each particle bin), our results do not depend on how models tune their total emissions to match observations or even on how entire regions are tuned (as done in for instance Albani et al., 2014). Results do depend somewhat on tuning to describe emissions within each source region, as this affects the spatial pattern of normalized emission, loading, etc. We've added a sentence in section 2.1 to clarify that our results are (mostly) independent of model tuning, and also added a sentence to Section 4.1 pointing out that the AOD data used for the Ridley product is indeed assimilated by MERRA2 and was used in the development of many of the models in the ensemble.

(5) Unfortunately, the methodology and results as presented in current form, seem to not contribute to a overall improvement of dust production models or their underlying conceptual understanding. Rather, the inversion model seems to reflect the consensus view across all models.

This work indeed does not explicitly evaluate dust production models; it provides insights into the assumed microphysical dust properties and makes several recommendations to improve dust simulations (see Section 5.2).

What can the science community learn from this approach? I am sure there is something we can learn from this approach. Maybe this can be presented in a more prominent and obvious way?

We use our results to provide recommendations to improve both dust models (Section 5.2) and the calculation of dust impacts on the Earth system (Section 5.3). It's a good idea to make this more prominent and we have now done so by placing a summary of these recommendations in a separate paragraph in the conclusions.

(6) The physical consistency of dust emission flux calculations and dispersion with atmospheric conditions which then results into a dependent dust aerosol optical depth seems to be circumvented by this approach here. Can you comment on how physically consistent the overall representation of the dust cycle and its associated feedbacks is?

This might be partially a misunderstanding. The crux of our (admittedly quite involved) approach is that, rather than letting each model decide how much dust is produced by each region and what the size distribution and extinction efficiency of that dust is, we base those factors on observational constraints (the blue boxes in Fig.1). So the representation of the dust cycle is internally consistent. However, this approach in its current form does not allow feedbacks between dust and the driving meteorology. Although this is common for dust studies (e.g., Miller et al., 2004; Mahowald et al., 2010), this is indeed an important limitation that we now mention in the limitation section (5.1).

(7) Furthermore, as the individual model ensembles use different parameterization schemes for various processes ultimately determining the dust cycle, I am wondering if this is an advantage

here as diversity is reflected, or a disadvantage as it ignores physical consistency for the multi-model average. Please clarify.

Per the response to remark #3 above, we do consider our procedure internally consistent. So the reviewer makes a great point that drawing from several models captures (some of) the diversity in the different treatment of dust emission and deposition in models, which we do consider a strength.

(8) Where do you see the benefit of applying an inverse model over satellite data assimilation or a common ensemble mean?

That's a good question. There are several key benefits of the inverse model over satellite data assimilation. First, our approach enables the use of constraints that explicitly include uncertainty and that apply to dust only (like the Ridley et al. (2016) DAOD constraints), rather than satellite retrievals that inherently include all (or several) aerosol species, making results dependent on the fidelity of model simulations for those other species. Second, results from multi-model ensembles or satellite data assimilations are subject to possibly substantial biases due to a number of required assumptions, including regarding the optical properties and size distribution of dust. In our approach, dust properties are instead based on observational constraints for which the uncertainties have been propagated into our results (Kok et al., 2017; Adebiyi and Kok, 2020). A third key advantage of the constraints presented here and in our companion paper (Kok et al., 2021) is that the constraints are source region-resolved. This is particularly important for accounting for the effects of regional differences in soil mineralogy on dust impacts on radiation, clouds, and biogeochemistry, as well as for interpreting records of dust deposition from natural archives.

Specific comments:

Line 26: Is this an "improved representation" of the global dust cycle or an averaged representation as it reflects the consensus view across all model ensembles? Please clarify.

We consider it an improved representation because our product performs better against independent data than any of the models or the MERRA-2 dust product. We are not entirely sure what is meant by "consensus view across all model ensembles", but it's important to note that our product is not an average across models as it only uses the normalized (per unit loading) dust fields per particle size bin from the different models, and then adds those up in the way that best meets constraints on size distribution, extinction efficiency, and DAOD. As such, the product here is for instance independent of model choices regarding the global emission tuning constant, emitted size distribution, and dust mass extinction efficiency.

Line 52-71: Across this paragraph, the need for an as accurate as possible representation of processes driving and controlling the dust cycle is illustrated. However, isn't this, the relevance of process-driven parameterisation for improving the representation of the dust cycle, ignored by the approach suggested here? In particular as averaging may result into cancelling out relevant processes.

The reviewer is correct that this paper does not attempt to directly improve the parameterization of dust processes in models, although our results do provide insight into improved ways to parameterize the dust size distribution and mass extinction efficiency. The point this paragraph on lines 52-71 tries to make is that bottom-up modeling of the global dust cycle is made more difficult by issues such as the mismatch of scales between the model grid box and the relevant physics, and the scarcity of surface data at the needed resolution. We then note later in the Introduction that "The nature of these challenges in accurately representing the global dust cycle is such that they are difficult to overcome from advances in modeling alone. We therefore develop a new methodology to obtain an improved representation of the present-day global dust cycle." In other words, rather than improving the bottom-up modeling, this paper focuses on a new top-down approach to more accurately represent the global dust cycle, in order to get around the many limitations of effectively representing the global dust cycle from the ground up.

Line 313-317: I am wondering if this is only applicable to coarse grid model simulations which would not be able to capture fine-scale dust plumes anyway. Please comment.

The text on lines 313-7 is: "We then used an inverse modeling approach to determine the optimal combination of dust loadings from the nine source regions (denoted with subscript $r$) that minimizes the disagreement against the DAOD constraint of these 15 observed regions (denoted with subscript $p$) for each season. We thus need to account for the contribution of each of the nine source regions (Fig. 2a) to the DAOD in each of these 15 observed regions. The seasonally averaged DAOD over the observed region $p$ is [...]"

This does not mention fine-scale dust plumes so we are confused what the comment is referring to. Perhaps the comment referred to incorrect line numbers? In any case, we note that higher resolution will not by itself improve model estimates of regional emission if the physical processes responsible for emission remain incompletely understood. Given this limitation of understanding, the empirical constraints upon regional emission applied in this article will remain valuable. We have added this point to Section 5.1.

**Referee #1**

**Overview**

This article is well presented and well written, and fits well within the scope of ACP. The authors present a new methodology to constrain and improve dust models using an inverse modeling approach. This is evaluated compared to independent observations of surface dust concentration and dust deposition from around the globe. The authors show that their results deliver significant improvements in modeled parameters which have previously been extremely challenging for dust models. This work is a significant undertaking, and delivers important results to the field of global aerosol modeling. My comments are only minor clarifications and I recommend publication after they have been answered.

Thank you for your careful review of our paper and your positive comments.

**General Comments**

The article is complex and somewhat lengthy – however the authors strive to carefully explain their methodology, which is a crucial part of the paper, and the description of the method is clear and appropriate. Therefore I advise retaining the current amount of detail and length (and not shortening the article, as suggested by reviewer 1).

We do agree with this suggestion and have not made substantial cuts to the paper that could undermine the explanation of the methodology or conclusions.

**Specific Comments**

L57-58 – It is not just wind speed which is important to emission, but also the infrequent but strong wind occurrence which can contribute disproportionately strongly towards dust emission due to the non-linear scaling mentioned. Cowie et al. (2015) and Roberts et al. (2017) are relevant papers towards understanding and representing dust-generating winds in observations in models, with a focus on high-wind speed occurrence, which should be mentioned.

Thank you for the helpful suggestion. We've added a phrase to this paragraph pointing out that models poorly capture high wind events and cited those two papers.

L86-87 – sentence is unclear.

We've edited this sentence to improve clarity.

Introduction – readers may find the scope of this article fairly similar to several previous articles by the lead author and research group (e.g. Adebiyi et al. 2020, Adebiyi & Kok 2020. Therefore, it would be helpful to guide the reader briefly through the steps in which this piece of work differs from others, either in the introduction or wherever the authors consider logical. Improvements here would contribute a large amount to increasing the significance of this paper.

That's a good idea for clarity. We've added a sentence to the second-to-last paragraph in the Introduction that points out that this paper used constraints on dust properties and abundance from this past work from our research group and incorporates those into the inverse model. We hope this will indeed aid the reader in understanding the logical progression from Adebiyi et al. (2020) and Adebiyi & Kok (2020) to this present paper.

L208-212 – it is not clear how the authors use the GEOS model to extend the size range for CESM and ModelE2 – please give more details (and also relate to comments for the supplement).

Thank you for flagging this. We've added a more detailed description in the Supplement, which we now refer to in the main text.

Section 2.3, p8 – take care when writing 'this study' and referring to Ridley et al. (2016) – it is not always clear that the authors mean Ridley et al. by 'this', as opposed to referring to the current article under review. 'That study' or a repeat of 'Ridley et al' would be clearer.

Thank you for pointing this out and we have corrected that section as suggested.

Section 2.3, 2nd & 3rd paragraphs – given that the reanalysis datasets also include aerosol data assimilation, why not just use them for DAOD rather than the Ridley dataset? What is the advantage of the Ridley dataset over the reanalysis datasets in general?

That's a good question and the article would indeed benefit from articulating this better. We have now added a sentence to section 2.3 to point out the advantages of the Ridley product over reanalysis data more clearly.

The sentence we added is: "We also consider the Ridley et al. dataset more accurate than aerosol reanalysis products that assimilate similar AOD observations. This is because the Ridley et al. product includes a transparent quantification of errors and because the partitioning of assimilated AOD into different aerosol species in reanalysis products depends on the underlying aerosol models and is thus susceptible to the large biases in the prognostic aerosol schemes of these models (Adebiyi et al., 2020; Gliss et al., 2021)"

L412-413 – it would be useful to include the description of this dataset, the 'Saharan Dust Transect.'

That is a helpful suggestion and we have added some detail on this data set to Section 3.1.

L424-425 – this reasoning is unclear. "because most of the dust at this site originated locally from within and near the national park where the station was located" – does 'this site' refer to the Zimbabwe or Australian site?

We've edited this sentence to clarify that this statement refers to the Zimbabwe site.

Section 4.1 – Figs 3e and f are barely mentioned in the text. What does fig 3e tell us (seasonal DAOD) that fig 3f (annual) does not? What do their differences indicate?

Figs. 3e and 3f are indeed highly similar. We chose to show both because the seasonal DAOD is what the Ridley product provides and the annual DAOD is what is used in the error metric shown in Figs. 8e and 11e for consistency with the annual timescale of deposition and surface concentration. We have rephrased the relevant sentence in Section 4.1 to draw more attention to the inverse model reproducing DAOD very well on both the seasonal and the annual timescales.

L572 – O'Sullivan et al., 2020, is another recent publication finding that also found that coarse dust is deposited too quickly in models, and could be referenced here. Also L952.

Thank you for the suggestion. We've read this excellent article that shows that several models underestimate the height at which Saharan dust is transported across the Atlantic. However, this paper does not seem to directly infer that this causes models to deposit dust too quickly, so we have not added the reference at those locations. We did add a citation to this reference in section 5.1, to reinforce the statement that the vertical distribution of dust is known to be subject to important model errors.

L572-575 – what does a similar plot of either MEE or DAOD for dust longwave properties look like? (in terms of the panels shown in fig 5). Given that the authors point out that the longwave

radiative effects of dust are also important, it would be useful to provide some information from this study regarding the LW properties.

This is an excellent idea, and we have added a panel showing the size-resolved contribution to DAOD at a 10 um wavelength.

Table 3 – include DAOD and MEE wavelength in caption or table header.

Corrected as suggested.

Section 4.3.1 – regarding figure 6 – it might be expected that if one took a multi-model mean the result might agree fairly well with the observations as well. Since the inverse model incorporates input from several models, can the authors explain and confirm why the inverse model can be taken as agreeing better with the observations that a multi-model mean might? I suppose this can be inferred from the size of the error bars compared to the range individual models, though it would be useful to include a discussion of this.

This is a good question. Although we did not examine how the multi-model means perform against the independent measurement, we have now included the mean bias and RMSE of models in both ensembles in Figures 8 and 11. For the bias, the multi-model mean would be identical to that of the mean of the models. In all cases (deposition flux and seasonal and annual concentration), the inverse model outperforms the mean of both ensembles for the NH.

One can also see clear biases for the multi-model mean of the different ensembles in Fig. 6. For instance, all six models in our ensemble clearly overestimate the dust surface concentration for the African stations.

We think that the likely reason for why the inverse model does better against independent measurements is because it integrates observational constraints on the microphysical properties of dust (size distribution and mass extinction efficiency) and its abundance (regional DAOD). We discuss this further in Section 5.2.

L738-9 – But isn't MERRA used for DAOD in the SH, rather than the Ridley dataset?

Yes, indeed we use an ensemble of reanalysis products to constrain DAOD in the SH. We've rewritten this sentence for clarity.

L953-954 – "This could be done either by similarly applying the constraints on the globally averaged size distribution (Adebiyi and Kok, 2020)" – can the authors be more specific about how this could be carried out in an online (climate) model?

This indeed was not clear. We've revised this sentence to clarify that we propose to adjust the parameterized emitted dust size distribution such that the global dust loading matches the constraints on the globally averaged size distribution in Adebiyi and Kok (2020).

Section 5.3 – various other datasets of global Dust AOD are now available, which may have similarities and/or differences to Ridley et al. (2016). For example, Pu and Ginoux (2018) based on MODIS data, datasets based on CALIOP satellite AOD where dust AOD is derived based on shape, and combinations of MODIS/MERRA data (e.g. MIDAS, Gkikas et al., 2021). A

discussion of the possible implications of applying a different DAOD dataset might result in would be beneficial.

That's a great suggestion and we have added a brief discussion of this at the end of Section 5.3.

**Figures**

Fig 3 – caption "same as panel b" – should be panel e?

Corrected.

Figure 5 – what is meant by 'number load' – should this be number concentration?

We clarified this in the caption to "global dust loading in terms of number of particles per size bin".

Figure 6 – it is impossible to distinguish between the black and brown lines, and also very difficult to make out the letters identifying different models.

The individual models are indeed difficult to distinguish. This is partially because having 22 lines makes it very difficult to draw each line so that it can be easily distinguished, and partially because we want to draw the reader's eye not to individual models in the two ensembles as analysis of individual models is beyond the scope of this paper, but rather to the behavior of these models as a group. Nonetheless, it'd be preferable if individual models are more readily distinguishable and we have thickened the lines and enlarged the symbols for models in the two ensembles.

Fig 7 – what is meant by 'seasonal surface concentration' when given as one number? Same applies to fig 8, 10 and 11.

We have changed the wording in these captions to seasonally averaged and annually averaged surface concentration to make this clearer.

Fig 7 and fig 8 – could the mean of Aerocom models also be shown on the figures, since means of improved models and mean of ensemble models are shown, for clarity?

That's a good idea and we have done so (we assume the reviewer meant to refer here to figures 8 and 11, which shows means of improved models and the model ensemble).

**Supplement**

CESM/CAM4 – A description of how the additional largest size bin was generated is missing.

This has now been added, and also for the GISS model.

L90-91 "Simulations for each of the nine source regions were obtained by tagging each particle size bin from the different source regions." – it is not clear what is meant by this statement – please expand.

We clarified this to "To isolate the contribution of each of the specified source regions to the global dust loading, we tagged the dust originating from each source region."

L96-97 – "We did not use the largest bin because it exceeds the 20 μm maximum diameter used in the inverse model." – this statement appears to contradict the caption for table 1, main article, which indicates that the maximum size bin has been extended for this model.

Indeed, we did not use the largest (16-32 um) size bin and instead generated a 16-20 um bin. We now clarified this as "We did not use the largest bin (16-32 μm) because it exceeds the 20 μm maximum diameter used in the inverse model and instead generated a 16-20 μm bin based on the 8-16 μm bin and the GOCART 12-20 μm bin, as follows [..]"

L215 – "mm" – should be microns?

Corrected.

**References**

Adebiyi, A.A., Kok, J.F., 2020. Climate models miss most of the coarse dust in the atmosphere. Science Advances 6, eaaz9507.

Adebiyi, A.A., Kok, J.F., Wang, Y., Ito, A., Ridley, D.A., Nabat, P., Zhao, C., 2020. Dust Constraints from joint Observational-Modelling-experiMental analysis (DustCOMM): comparison with measurements and model simulations. Atmospheric Chemistry and Physics 20, 829-863.

Gliss, J., et al., 2021. AeroCom phase III multi-model evaluation of the aerosol life cycle and optical properties using ground- and space-based remote sensing as well as surface in situ observations. Atmospheric Chemistry and Physics 21, 87-128.

Kok, J.F., et al., 2021. Contribution of the world's main dust source regions to the global cycle of desert dust. Atmos. Chem. Phys. Discuss. [preprint] in review.

Kok, J.F., et al., 2017. Smaller desert dust cooling effect estimated from analysis of dust size and abundance. Nature Geoscience 10, 274-278.

Mahowald, N.M., et al., 2010. Observed 20th century desert dust variability: impact on climate and biogeochemistry. Atmos. Chem. Phys. 10, 10875-10893.

Miller, R.L., Tegen, I., Perlwitz, J., 2004. Surface radiative forcing by soil dust aerosols and the hydrologic cycle. Journal of Geophysical Research-Atmospheres 109, D04203.

---

## Referee Report (RR1)

This work presents an inverse modelling framework that improves the representation of the global dust cycle by using global model simulations and observational constrains dust aerosol optical depth, extinction efficiency and dust size distribution. The authors compare the inverse model results against independent observations of surface concentration and dust deposition and show the large improvements obtained in the northern hemisphere and the modest improvements in the southern hemisphere. Furthermore, the authors include interesting discussion on the limitations and implications of the results as well as the potential and future perspective of the developed methodology.

The paper is well written and well structured. Although the paper is long, the effort to present the methodology is appreciated and the authors do a good job in explaining it. Considering the extensive work done and numerous results obtained, the authors manage to focus on the main results and not get lost in the details. I only have some very minor comments after which I believe the paper can be published.

Line 98-104: I don't see the point of including results of the work at this point. This is not the place to present results, the methodology hasn't even been presented.

Line 113: I would suggest to replace "inform" with "force". The information (observations) in inverse modelling is not provided (inform) to the model but used in the model to produce (force) a change.

Line 113-114: What are these substantial differences?

Figure 3e-f: I couldn't find any reference to figures 3e and 3f in the text. Either include a reference to the figures in the text or remove the figure.

Lines 973-974: Could absorption AOD be an additional independent dataset to which compare the inverse model results?

---

## Author Response (AR2)

**Responses to reviewers' comments**
Comments in black and responses in blue

We thank the reviewers and Editor for their constructive and supportive comments, which we address below.

This work presents an inverse modelling framework that improves the representation of the global dust cycle by using global model simulations and observational constrains dust aerosol optical depth, extinction efficiency and dust size distribution. The authors compare the inverse model results against independent observations of surface concentration and dust deposition and show the large improvements obtained in the northern hemisphere and the modest improvements in the southern hemisphere.

Furthermore, the authors include interesting discussion on the limitations and implications of the results as well as the potential and future perspective of the developed methodology. The paper is well written and well structured. Although the paper is long, the effort to present the methodology is appreciated and the authors do a good job in explaining it. Considering the extensive work done and numerous results obtained, the authors manage to focus on the main results and not get lost in the details.

Thank you for these positive and helpful comments.

I only have some very minor comments after which I believe the paper can be published.

Line 98-104: I don't see the point of including results of the work at this point. This is not the place to present results, the methodology hasn't even been presented.

Since the end of the introduction is a point of emphasis, we think that summarizing our main results and conclusions here helps ensure that the reader will obtain the paper's main points.

Line 113: I would suggest to replace "inform"with "force". The information (observations) in inverse modelling is not provided (inform) to the model but used in the model to produce (force) a change.

Corrected as suggested.

Line 113-114: What are these substantial differences?

We've added "in that it integrates several different constraints on dust microphysical properties and uses a bootstrap to propagate and quantify uncertainties" to the end of this sentence to clarify the main differences here.

Figure 3e-f: I couldn't find any reference to figures 3e and 3f in the text. Either include a reference to the figures in the text or remove the figure.

Thank you for pointing this out. Somehow, the paragraph discussing this figures was accidentally deleted from the revised version. We have added it back in on lines 549-560.

Lines 973-974: Could absorption AOD be an additional independent dataset to which compare the inverse model results?

That's a good idea. Unfortunately, AAOD cannot be measured directly with current remote sensing assets – it's always retrieved – and existing retrievals by show large differences (e.g., Samset et al., 2018). In fact, our group is working on a paper that indicates substantial errors in AERONET retrievals of AAOD, apparently due to errors in retrievals of dust optical properties and size.